# FIRST-ORDER ANIL PROVABLY LEARNS REPRESENTATIONS DESPITE OVERPARAMETRIZATION

**Oğuz Kaan Yüksel**
TML Lab, EPFL
`oguz.yuksel@epfl.ch`

**Etienne Boursier** *
INRIA, Université Paris Saclay, LMO
`etienne.boursier@inria.fr`

**Nicolas Flammarion**
TML Lab, EPFL
`nicolas.flammarion@epfl.ch`

## ABSTRACT

Due to its empirical success in few-shot classification and reinforcement learning, meta-learning has recently received significant interest. Meta-learning methods leverage data from previous tasks to learn a new task in a sample-efficient manner. In particular, model-agnostic methods look for initialization points from which gradient descent quickly adapts to any new task. Although it has been empirically suggested that such methods perform well by learning shared representations during pretraining, there is limited theoretical evidence of such behavior. More importantly, it has not been shown that these methods still learn a shared structure, despite architectural misspecifications. In this direction, this work shows, in the limit of an infinite number of tasks, that first-order ANIL with a linear two-layer network architecture successfully learns linear shared representations. This result even holds with *overparametrization*; having a width larger than the dimension of the shared representations results in an asymptotically low-rank solution. The learned solution then yields a good adaptation performance on any new task after a single gradient step. Overall, this illustrates how well model-agnostic methods such as first-order ANIL can learn shared representations.

## 1 INTRODUCTION

Supervised learning usually requires a large amount of data. To overcome the limited number of available training samples for a single task, multi-task learning estimates a model across multiple tasks (Ando & Zhang, 2005; Cheng et al., 2011). The global performance can then be improved for individual tasks once structural similarities between these tasks are correctly learned and leveraged. Closely related, *meta-learning* aims to quickly adapt to any new task, by leveraging the knowledge gained from previous tasks, e.g., by learning a shared representation that enables fast adaptation.

Meta-learning has been mostly popularized by the success of the Model-Agnostic Meta-Learning (MAML) algorithm for few-shot image classification and reinforcement learning (Finn et al., 2017). MAML searches for an initialization point such that only a few task-specific gradient descent iterations yield good performance on any new task. It is *model-agnostic* in the sense that the objective is readily applicable to any architecture that is trained with a gradient descent procedure, without any modifications. Subsequently, many model-agnostic methods have been proposed (Nichol et al., 2018; Antoniou et al., 2019; Raghu et al., 2020; Hospedales et al., 2022). Raghu et al. (2020) empirically support that MAML implicitly learns a shared representation across the tasks, since its intermediate layers do not significantly change during task-specific finetuning. Consequently, they propose the Almost-No-Inner-Loop (ANIL) algorithm, which only updates the last layer during task-specific updates and performs similarly to MAML. However, to avoid heavy computations for second-order derivatives, practitioners generally use first-order approximations such as FO-MAML or FO-ANIL that achieve comparable performances at a cheaper cost (Nichol et al., 2018).

Despite the empirical success of model-agnostic methods, little is known about their behaviors in theory. To this end, our work considers the following question on the pretraining of FO-ANIL:

---
* This work was completed while E. Boursier was a member of TML Lab, EPFL.

*Do model-agnostic methods learn shared representations in few-shot settings?*

Proving positive optimization results on the pretraining of meta-learning models is out of reach in general, complex settings that may be encountered in practice. Indeed, research beyond linear models has mostly been confined to the finetuning phase (Ju et al., 2022; Chua et al., 2021). Hence, to allow a tractable analysis, we study FO-ANIL in the canonical multi-task model of a linear shared representation; and consider a linear two-layer network, which is the minimal architecture achieving non-trivial performance. Traditional multi-task learning methods such as Burer-Monteiro factorization (or matrix factorization) (Tripuraneni et al., 2021; Du et al., 2021; Thekumparampil et al., 2021) and nuclear norm regularization (Rohde & Tsybakov, 2011; Boursier et al., 2022) are known for correctly learning the shared representation. Besides being specific to this linear model, they rely on prior knowledge of the hidden dimension of the common structure that is unknown in practice.

For meta-learning in this canonical multi-task model, Saunshi et al. (2020) has shown the first result under overparametrization by considering a unidimensional shared representation, infinite samples per task, and an idealized algorithm. More recently, (Collins et al., 2022) has provided a multi-dimensional analysis for MAML and ANIL in which the hidden layer recovers the ground-truth low-dimensional subspace at an exponential rate. Similar to multi-task methods, the latter result relies on *well-specification* of the network width, i.e., it has to coincide with the hidden dimension of the shared structure. Moreover, it requires a weak alignment between the hidden layer and the ground truth at initialization, which is not satisfied in high-dimensional settings.

The power of MAML and ANIL, however, comes from their good performance despite mismatches between the architecture and the problem; and in *few-shot* settings, where the number of samples per task is limited but the number of tasks is not. In this direction, we prove a learning result under a framework that reflects the meta-learning regime. Specifically, we show that FO-ANIL successfully learns multidimensional linear shared structures with an overparametrised network width and without initial weak alignment. Our setting of finite samples and infinite tasks is better suited for practical scenarios and admits novel behaviors unobserved in previous works. In particular, FO-ANIL not only *learns* the low-dimensional subspace, but it also *unlearns* its orthogonal complement. This unlearning does not happen with infinite samples and is crucial during task-specific finetuning. In addition, we reveal a slowdown due to overparametrization, which has been also observed in supervised learning (Xu & Du, 2023). Overall, our result provides the first learning guarantee under misspecifications, and shows the benefits of model-agnostic meta-learning over multi-task learning.

**Contributions.** We study FO-ANIL in a linear shared representation model introduced in Section 2. In order to allow a tractable yet non-trivial analysis, we consider infinite tasks idealisation, which is more representative of meta-learning than the infinite samples idealisation considered in previous works. Section 3 presents our main result, stating that FO-ANIL asymptotically learns an accurate representation of the hidden problem structure despite a misspecification in the network width. When adapting this representation to a new task, FO-ANIL quickly achieves a test loss comparable to linear regression on the hidden low-dimensional subspace. Section 4 then discusses these results, their limitations, and compares them with the literature. Finally, Section 5 empirically illustrates the success of model-agnostic methods in learned representation and at test time.

## 2  PROBLEM SETTING

### 2.1  DATA DISTRIBUTION

In the following, tasks are indexed by $i \in \mathbb{N}$. Each task corresponds to a $d$-dimensional linear regression task with parameter $\theta_{\star,i} \in \mathbb{R}^d$ and $m$ observation samples. Mathematically, we have for each task $i$ observations $(X_i, y_i) \in \mathbb{R}^{m \times d} \times \mathbb{R}^m$ such that

$$y_i = X_i \theta_{\star,i} + z_i \quad \text{where } z_i \in \mathbb{R}^m \text{ is some random noise.}$$

Some shared structure is required between the tasks to meta-learn, i.e., to be able to speed up the learning of a new task. Similarly to the multi-task linear representation learning setting, we assume that the regression parameters $\theta_{\star,i}$ all lie in the same small $k$-dimensional linear subspace, with $k < d$. Equivalently, there is an orthogonal matrix $B_\star \in \mathbb{R}^{d \times k}$ and representation parameters $w_{\star,i} \in \mathbb{R}^k$ such that $\theta_{\star,i} = B_\star w_{\star,i}$ for any task $i$. To derive a proper analysis of this setting, we assume a random design of the different quantities of interest, summarized in Assumption 1. This assumption and how it could be relaxed is discussed in Section 4.

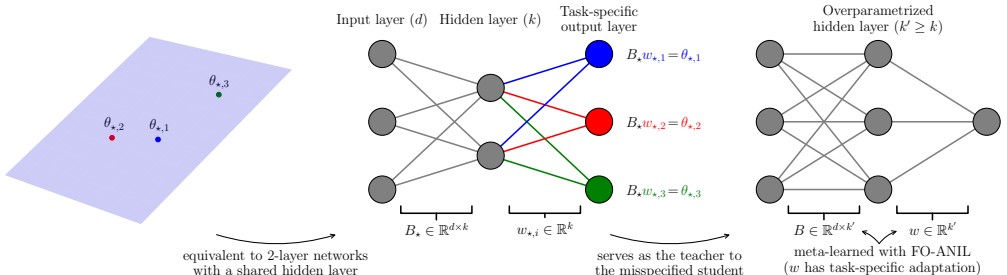

Figure 1: **Left:** Regression tasks with parameters $\theta_{\star,i} \in \mathbb{R}^d$ are confined in a lower dimensional subspace, equivalent to the column space of matrix $B_\star \in \mathbb{R}^{d \times k}$. **Center:** This is equivalent to having two-layer, linear, teacher networks where $B_\star$ is the shared hidden layer and the outputs layers $w_{\star,i} \in \mathbb{R}^k$ are task-specific. The meta-learning task is finding an initialization that allows fast adaptation to any such task with a few samples. **Right:** The student network has the same architecture but it is agnostic to the problem hidden dimension, i.e., $k' \geq k$, which is the main difficulty in our theoretical setting. On the contrary, previous works on model-agnostic and representation learning methods assume $k' = k$, i.e., the hidden dimension is *a priori* known to the learner.

**Assumption 1** (random design). *Each row of $X_i$ is drawn i.i.d. according to $\mathcal{N}(0, \mathbf{I}_d)$ and the coordinates of $z_i$ are i.i.d., centered random variables of variance $\sigma^2$. Moreover, the task parameters $w_{\star,i}$ are drawn i.i.d with $\mathbb{E}[w_{\star,i}] = 0$ and covariance matrix $\Sigma_\star := \mathbb{E}[w_{\star,i}w_{\star,i}^\top] = c \, \mathbf{I}_k$ with $c > 0$.*

## 2.2 FO-ANIL ALGORITHM

This section introduces FO-ANIL in the setting described above for $N \in \mathbb{N}$ tasks, as well as in the idealized setting of infinite tasks ($N = \infty$). The goal of model-agnostic methods is to learn parameters, for a given neural network architecture, that quickly adapt to a new task. This work focuses on a linear two-layer network architecture, parametrised by $\theta := (B, w) \in \mathbb{R}^{d \times k'} \times \mathbb{R}^{k'}$ with $k \leq k' \leq d$. The estimated function is then given by $f_\theta : x \mapsto x^\top B w$.

The ANIL algorithm aims at minimizing the test loss on a new task, after a small number of gradient steps on the last layer of the neural network. For the sake of simplicity, we here consider a single gradient step. ANIL then aims at minimizing over $\theta$ the quantity

$$\mathcal{L}_{\text{ANIL}}(\theta) := \mathbb{E}_{w_{\star,i}, X_i, y_i} \left[ \mathcal{L}_i \left( \theta - \alpha \nabla_w \hat{\mathcal{L}}_i(\theta; X_i, y_i) \right) \right], \tag{1}$$

where $\mathcal{L}_i$ is the (expected) test loss on the task $i$, which depends on $w_{\star,i}$; $\hat{\mathcal{L}}_i(\theta; X_i, y_i)$ is the empirical loss on the observations $(X_i, y_i)$; and $\alpha$ is the gradient step size. When the whole parameter is updated at test time, i.e., $\nabla_w$ is replaced by $\nabla_\theta$, this instead corresponds to the MAML algorithm.

For model-agnostic methods, it is important to split the data in two for inner and outer loops during training. Otherwise, the model would indeed overfit the training set and would learn a poor, full rank representation of the task parameters (Saunshi et al., 2021). For $m_{\text{in}} + m_{\text{out}} = m$ with $m_{\text{in}} < m$, we split the observations of each task as $(X_i^{\text{in}}, y_i^{\text{in}}) \in \mathbb{R}^{m_{\text{in}} \times d} \times \mathbb{R}^{m_{\text{in}}}$ the $m_{\text{in}}$ first rows of $(X_i, y_i)$; and $(X_i^{\text{out}}, y_i^{\text{out}}) \in \mathbb{R}^{m_{\text{out}} \times d} \times \mathbb{R}^{m_{\text{out}}}$ the $m_{\text{out}}$ last rows of $(X_i, y_i)$.

While training, ANIL alternates at each step $t \in \mathbb{N}$ between an inner and an outer loop to update the parameter $\theta_t$. In the inner loop, the last layer of the network is adapted to each task $i$ following

$$w_{t,i} \leftarrow w_t - \alpha \nabla_w \hat{\mathcal{L}}_i(\theta_t; X_i^{\text{in}}, y_i^{\text{in}}). \tag{2}$$

Again, updating the whole parameter $\theta_{t,i}$ with $\nabla_\theta$ would correspond to MAML algorithm. In the outer loop, ANIL then takes a gradient step (with learning rate $\beta$) on the validation loss obtained for the observations $(X_i^{\text{out}}, y_i^{\text{out}})$ after this inner loop. With $\theta_{t,i} := (B_t, w_{t,i})$, it updates

$$\theta_{t+1} \leftarrow \theta_t - \frac{\beta}{N} \sum_{i=1}^N \hat{H}_{t,i}(\theta_t) \nabla_\theta \hat{\mathcal{L}}_i(\theta_{t,i}; X_i^{\text{out}}, y_i^{\text{out}}), \tag{3}$$

where the matrix $\hat{H}_{t,i}$ accounts for the derivative of the function $\theta_t \mapsto \theta_{t,i}$. Computing the second-order derivatives appearing in $\hat{H}_{t,i}$ is often very costly. Practitioners instead prefer to use first-order approximations, since they are cheaper in computation and yield similar performances (Nichol et al., 2018). FO-ANIL then replaces $\hat{H}_{t,i}$ by the identity matrix in Equation (3).

### 2.2.1 DETAILED ITERATIONS

In our regression setting, the empirical squared error is given by $\hat{\mathcal{L}}_i((B, w); X_i, y_i) = \frac{1}{2m}\|y_i - X_i B w\|_2^2$. In that case, the FO-ANIL inner loop of Equation (2) gives in the setting of Section 2.1:

$$w_{t,i} = w_t - \frac{\alpha}{m_{\mathrm{in}}} B_t^\top (X_i^{\mathrm{in}})^\top X_i^{\mathrm{in}} (B_t w_t - B_\star w_{\star,i}) + \frac{\alpha}{m_{\mathrm{in}}} B_t^\top (X_i^{\mathrm{in}})^\top z_i^{\mathrm{in}}. \tag{4}$$

The multi-task learning literature often considers a large number of tasks (Thekumparampil et al., 2021; Boursier et al., 2022) to allow a tractable analysis. Similarly, we study FO-ANIL in the limit of an infinite number of tasks $N = \infty$ to simplify the outer loop updates. In this limit, iterations are given by the exact gradient of the ANIL loss defined in Equation (1), when ignoring the second-order derivatives. The first-order outer loop updates of Equation (3) then simplify with Assumption 1 to

$$w_{t+1} = w_t - \beta(\mathbf{I}_{k'} - \alpha B_t^\top B_t)B_t^\top B_t w_t, \tag{5}$$

$$B_{t+1} = B_t - \beta B_t \mathbb{E}[w_{t,i} w_{t,i}^\top] + \alpha\beta B_\star \Sigma_\star B_\star^\top B_t, \tag{6}$$

where $w_{t,i}$ is still given by Equation (4). Moreover, Lemma 15 in the Appendix allows with Assumption 1 to compute an exact expression of $\mathbb{E}[w_{t,i} w_{t,i}^\top]$ as

$$\begin{aligned}
\mathbb{E}[w_{t,i} w_{t,i}^\top] =&\ (\mathbf{I}_{k'} - \alpha B_t^\top B_t) w_t w_t^\top (\mathbf{I}_{k'} - \alpha B_t^\top B_t) + \alpha^2 B_t^\top B_\star \Sigma_\star B_\star^\top B_t \\
&+ \frac{\alpha^2}{m_{\mathrm{in}}} B_t^\top \left( B_t w_t w_t^\top B_t^\top + B_\star \Sigma_\star B_\star^\top + \left(\|B_t w_t\|^2 + \mathrm{Tr}(\Sigma_\star) + \sigma^2\right) \mathbf{I}_d \right) B_t.
\end{aligned} \tag{7}$$

The first line is the covariance obtained for an infinite number of samples. The second line comes from errors due to the finite number of samples and the label noise. As a comparison, MAML also updates matrices $B_{t,i}$ in the inner loop, which then intervene in the updates of $w_t$ and $B_t$. Because of this entanglement, the iterates of first-order MAML (and hence its analysis) are very cumbersome.

## 3 LEARNING A GOOD REPRESENTATION

Given the complexity of its iterates, FO-ANIL is very intricate to analyze even in the simplified setting of infinite tasks. The objective function is non-convex in its arguments and the iterations involve high-order terms in both $w_t$ and $B_t$, as seen in Equations (5) to (7). Theorem 1 yet characterizes convergence towards some fixed point (of the iterates) satisfying a number of conditions.

**Theorem 1.** *Let $B_0$ and $w_0$ be initialized such that $B_\star^\top B_0$ is full rank,*

$$\|B_0\|_2^2 = \mathcal{O}\left(\alpha^{-1} \min\left(\frac{1}{m_{\mathrm{in}}}, \frac{m_{\mathrm{in}}}{\bar{\sigma}^2}\right)\right), \quad \|w_0\|_2^2 = \mathcal{O}\left(\alpha \lambda_{\min}(\Sigma_\star)\right),$$

*where $\lambda_{\min}(\Sigma_\star)$ is the smallest eigenvalue of $\Sigma_\star$, $\bar{\sigma}^2 := \mathrm{Tr}(\Sigma_\star) + \sigma^2$ and the $\mathcal{O}$ notation hides universal constants. Let also the step sizes satisfy $\alpha \geq \beta$ and $\alpha = \mathcal{O}(1/\bar{\sigma})$.*

*Then under Assumption 1, FO-ANIL (given by Equations (5) and (6)) with initial parameters $B_0, w_0$, asymptotically satisfies the following*

$$\lim_{t\to\infty} B_{\star,\perp}^\top B_t = 0, \qquad\qquad \lim_{t\to\infty} B_t w_t = 0,$$

$$\lim_{t\to\infty} B_\star^\top B_t B_t^\top B_\star = \Lambda_\star := \frac{1}{\alpha}\frac{m_{\mathrm{in}}}{m_{\mathrm{in}}+1}\left(\mathbf{I}_k - \left(\frac{m_{\mathrm{in}}+1}{\bar{\sigma}^2}\Sigma_\star + \mathbf{I}_k\right)^{-1}\right), \tag{8}$$

*where $B_{\star,\perp} \in \mathbb{R}^{d\times(d-k)}$ is an orthogonal matrix spanning the orthogonal of $\mathrm{col}(B_\star)$, i.e.,*

$$B_{\star,\perp}^\top B_{\star,\perp} = \mathbf{I}_{d-k}, \quad \text{and} \quad B_\star^\top B_{\star,\perp} = 0.$$

An extended version of Theorem 1 and its proof are postponed to Appendix C. We conjecture that Theorem 1 holds with arbitrary task covariances $\Sigma_\star$ beyond the identity covariance in Assumption 1. Before discussing the implications of Theorem 1, we provide details on the proof strategy.

The proof is based on the monotonic decay of $\|B_{\star,\perp}^\top B_t\|_2$, $\|B_t w_t\|_2$ and the monotonic increase of $B_\star^\top B_t B_t^\top B_\star$ in the Loewner order sense. As these three quantities are interrelated, simultaneously controlling them is challenging. The initialization given in Theorem 1 achieves this by conditioning the dynamics to be bounded and well-behaved. The choice of $\alpha$ is also crucial as it guarantees the

decay of $\|B_t w_t\|_2$ after $\|B_{\star,\perp}^\top B_t\|_2$ decays. While these two quantities decay, $\Lambda_t := B_\star^\top B_t B_t^\top B_\star$ follows a recursion where $B_{\star,\perp}^\top B_t$ and $B_t w_t$ act as noise terms. The proof utilizes two associated recursions to respectively upper and lower bound $\Lambda_t$ (in Loewner order) and then show their monotonic convergence to $\Lambda_\star$. A more detailed sketch of the proof could be found in Appendix B.

Theorem 1 states that under mild assumptions on initialization and step sizes, the parameters learned by FO-ANIL verify three key properties after convergence:

1. $B_\infty$ is rank-deficient, i.e., FO-ANIL learns to ignore the entire $d-k$ dimensional orthogonal subspace given by $B_{\star,\perp}$, as expressed by the first limit in Equation (8).

2. The learned initialization yields the zero function, as given by the second limit in Equation (8). Note that $w_t$ does not necessarily converge to $0$; however, it converges to the null space of $B_\star$, thanks to the third property. Although intuitive, showing that $B_t w_t$ converges to the mean task parameter (assumed $0$ here) is very challenging when starting away from it, as discussed in Section 4. This property is crucial for fast adaptation on a new task.

3. $B_\star^\top B_\infty B_\infty^\top B_\star$ is proportional to identity. Along with the first property, this fact implies that the learned matrix $B_\infty$ exactly spans $\mathrm{col}(B_\star)$. Moreover, its squared singular values scale as $\alpha^{-1}$, allowing to perform rapid learning with a single gradient step of size $\alpha$.

These three properties allow to obtain a good performance on a new task after a single gradient descent step, as intended by the training objective of ANIL. The generalization error at test time is precisely quantified by Proposition 1 in Section 3.1. In addition, the limit points characterized by Theorem 1 are shown to be global minima of the ANIL objective in Equation (1) in Appendix F.

Interestingly, Theorem 1 holds for quite large step sizes $\alpha, \beta$ and the limit points only depend on these parameters by the $\alpha^{-1}$ scaling of $\Lambda_\star$. Also note that $\Lambda_\star \to \frac{1}{\alpha}\mathbf{I}_k$ when $m_{\mathrm{in}} \to \infty$. Yet, there is some shrinkage of $\Lambda_\star$ for finite number of samples, that is significant when $m_{\mathrm{in}}$ is of order of the inverse eigenvalues of $\frac{1}{\sigma^2}\Sigma_\star$. This shrinkage mitigates the variance of the estimator returned after a single gradient step, while this estimator is unbiased with no shrinkage ($m_{\mathrm{in}} = \infty$).

Although the limiting behavior of FO-ANIL holds for any finite $m_{\mathrm{in}}$, the convergence rate can be arbitrarily slow for large $m_{\mathrm{in}}$. In particular, FO-ANIL becomes very slow to unlearn the orthogonal complement of $\mathrm{col}(B_\star)$ when $m_{\mathrm{in}}$ is large, as highlighted by Equation (14) in Appendix B. At the limit of infinite samples $m_{\mathrm{in}} = \infty$, FO-ANIL thus does not unlearn the orthogonal complement and the first limit of Equation (8) in Theorem 1 does not hold anymore. This unlearning is yet crucial at test time, since it reduces the dependency of the excess risk from $k'$ to $k$ (see Proposition 1).

### 3.1 FAST ADAPTATION TO A NEW TASK

Thanks to Theorem 1, FO-ANIL learns the shared representation during pretraining. It is yet unclear how this result enhances the learning of new tasks, often referred as *finetuning* in the literature. Consider having learned parameters $(\hat{B}, \hat{w}) \in \mathbb{R}^{d \times k'} \times \mathbb{R}^{k'}$ following Theorem 1,

$$B_{\star,\perp}^\top \hat{B} = 0; \quad \hat{B}\hat{w} = 0; \quad B_\star^\top \hat{B}\hat{B}^\top B_\star = \Lambda_\star. \tag{9}$$

We then observe a new regression task with $m_{\mathrm{test}}$ observations $(X, y) \in \mathbb{R}^{m_{\mathrm{test}} \times d} \times \mathbb{R}^{m_{\mathrm{test}}}$ and parameter $w_\star \in \mathbb{R}^k$ such that

$$y = XB_\star w_\star + z, \tag{10}$$

where the entries of $z$ are i.i.d. centered $\sigma$ sub-Gaussian random variables and the entries of $X$ are i.i.d. standard Gaussian variables following Assumption 1. The learner then estimates the regression parameter of the new task doing one step of gradient descent:

$$w_{\mathrm{test}} = \hat{w} - \alpha\nabla_w \hat{\mathcal{L}}((\hat{B}, \hat{w}); X, y) = \hat{w} + \alpha\hat{B}^\top \Sigma_{\mathrm{test}} B_\star w_\star + \frac{\alpha}{m_{\mathrm{test}}}\hat{B}^\top X^\top z, \tag{11}$$

with $\Sigma_{\mathrm{test}} := \frac{1}{m_{\mathrm{test}}}X^\top X$. As in the inner loop of ANIL, a single gradient step is processed here. Note that it is unclear whether a single or more gradient steps should be run at test time. Notably, $(\hat{B}, \hat{w})$ has not exactly converged in practice, since we consider a finite training time: $\hat{B}$ is thus full rank. The least squares estimator of the linear regression with data $(X\hat{B}, y)$ might then lead to overfitting. Running just a few gradient steps can be helpful by preventing overfitting since it implicitly regularizes the norm of the estimated parameters (Yao et al., 2007; Neu & Rosasco,

2018). The best strategy (e.g. the number of gradient steps) to run while finetuning is an intricate problem, independently studied in the literature (see e.g. Chua et al., 2021; Ren et al., 2023) and is out of the scope of this work. Additional details are provided in Appendix I.

When estimating the regression parameter with $\hat{B}w_{\text{test}}$, the excess risk on this task is exactly $\|\hat{B}w_{\text{test}} - B_\star w_\star\|_2^2$. Proposition 1 below allows to bound the risk on any new observed task.

**Proposition 1.** *Let* $\hat{B}, w_{\text{test}}$ *satisfy Equations* (9) *and* (11) *for a new task defined by Equation* (10). *If* $m_{\text{test}} \geq k$, *then with probability at least* $1 - 4e^{-\frac{k}{2}}$,

$$\|\hat{B}w_{\text{test}} - B_\star w_\star\|_2 = \mathcal{O}\Big(\frac{1 + \overline{\sigma}^2/\lambda_{\min}(\Sigma_\star)}{m_{\text{in}}}\|w_\star\|_2 + \|w_\star\|_2\sqrt{\frac{k}{m_{\text{test}}}} + \sigma\sqrt{\frac{k}{m_{\text{test}}}}\Big).$$

A more general version of Proposition 1 and its proof are postponed to Appendix D. The proof relies on the exact expression of $w_{\text{test}}$ after a single gradient update. The idea is to decompose the difference $\hat{B}w_{\text{test}} - B_\star w_\star$ in three terms, which are then bounded using concentration inequalities.

The first two terms come from the error due to proceeding a single gradient step, instead of converging towards the ERM weights: the first one is the bias of this error, while the second one is due to its variance. The last term is the typical error of linear regression on a $k$ dimensional space. Note this bound does not depend on the feature dimension $d$ (nor $k'$), but only on the hidden dimension $k$.

When learning a new task without prior knowledge, e.g., with a simple linear regression on the $d$-dimensional space of the features, the error instead scales as $\sigma\sqrt{\frac{d}{m_{\text{test}}}}$ (Bartlett et al., 2020). FO-ANIL thus leads to improved estimations on new tasks, when it beforehand learned the shared representation. Such a learning is guaranteed thanks to Theorem 1. Surprisingly, FO-ANIL might only need a single gradient step to outperform linear regression on the $d$-dimensional feature space, as empirically confirmed in Section 5. As explained, this quick adaptation is made possible by the $\alpha^{-1}$ scaling of $\hat{B}$, which leads to considerable updates after a single gradient step.

## 4 DISCUSSION

**No prior structure knowledge.** Previous works on model-agnostic methods and matrix factorization consider a well-specified learning architecture, i.e., $k' = k$ (Tripuraneni et al., 2021; Thekumparampil et al., 2021; Collins et al., 2022). In practical settings, the true dimension $k$ is hidden, and estimating it is part of learning the representation. Theorem 1 instead states that FO-ANIL recovers this hidden true dimension $k$ asymptotically when misspecified ($k' > k$) and still learns good shared representation despite overparametrization (e.g., $k' = d$). Theorem 1 thus illustrates the adaptivity of model-agnostic methods, which we believe contributes to their empirical success.

Proving good convergence of FO-ANIL despite misspecification in network width is the main technical challenge of this work. When correctly specified, it is sufficient to prove that FO-ANIL learns the subspace spanned by $B_\star$, which is simply measured by the principal angle distance by Collins et al. (2022). When largely misspecified ($k' = d$), this measure is always 1 and poorly reflects how good is the learned representation. Instead of a single measure, two phenomena are quantified here. FO-ANIL indeed not only learns the low-dimensional subspace, but it also unlearns its orthogonal complement.[1] More precisely, misspecification sets additional difficulties in controlling simultaneously the variables $w_t$ and $B_t$ through iterations. When $k' = k$, this control is possible by lower bounding the singular values of $B_t$. A similar argument is however not possible when $k' > k$, as the matrix $B_t$ is now rank deficient (at least asymptotically). To overcome this challenge, we use a different initialization regime and analysis techniques with respect to Saunshi et al. (2020); Collins et al. (2022). These advanced techniques allow to prove convergence of FO-ANIL with different assumptions on both the model and the initialization regime, as explained below.

**Superiority of agnostic methods.** When correctly specified ($k' = k$), model-agnostic methods do not outperform traditional multi-task learning methods. For example, the Burer-Monteiro factorization minimizes the non-convex problem

$$\min_{B \in \mathbb{R}^{d \times k'},\, W \in \mathbb{R}^{k' \times N}} \frac{1}{2N} \sum_{i=1}^N \hat{\mathcal{L}}_i(BW^{(i)}; X_i, y_i), \tag{12}$$

---

[1] Although Saunshi et al. (2020) consider a misspecified setting, the orthogonal complement is not unlearned in their case, since they assume an infinite number of samples per task (see *Infinite tasks model* paragraph).

where $W^{(i)}$ stands for the $i$-th column of the matrix $W$. Tripuraneni et al. (2021) show that any local minimum of Equation (12) correctly learns the shared representation when $k' = k$. However when misspecified (e.g., taking $k' = d$), there is no such guarantee. In that case, the optimal $B$ need to be full rank (e.g., $B = \mathbf{I}_d$) to perfectly fit the training data of all tasks, when there is label noise. This setting then resembles running independent $d$-dimensional linear regressions for each task and directly leads to a suboptimal performance of Burer-Monteiro factorizations, as illustrated in Section 5. This is another argument in favor of model-agnostic methods in practice: while they provably work despite overparametrization, traditional multi-task methods *a priori* do not.

Although Burer-Monteiro performs worse than FO-ANIL in the experiments of Section 5, it still largely outperforms the single-task baseline. We believe this good performance despite over-parametrization might be due to the implicit bias of matrix factorization towards low-rank solutions. This phenomenon remains largely misunderstood in theory, even after being extensively studied (Gunasekar et al., 2017; Arora et al., 2019; Razin & Cohen, 2020; Li et al., 2021). Explaining the surprisingly good performance of Burer-Monteiro thus remains a major open problem.

**Infinite tasks model.** A main assumption in Theorem 1 is the infinite tasks model, where updates are given by the exact (first-order) gradient of the objective function in Equation (1). Theoretical works often assume a large number of tasks to allow a tractable analysis (Thekumparampil et al., 2021; Boursier et al., 2022). The infinite tasks model idealises this type of assumption and leads to simplified parameters' updates. Note these updates, given by Equations (5) and (6), remain intricate to analyze. Saunshi et al. (2020); Collins et al. (2022) instead consider an infinite number of samples per task, i.e., $m_{\text{in}} = \infty$. This assumption leads to even simpler updates, and their analysis extends to the misspecified setting with some extra work, as explained in Appendix G. Collins et al. (2022) also extend their result to a finite number of samples in finite-time horizon, using concentration bounds on the updates to their infinite samples counterparts when sufficiently many samples are available.

More importantly, the infinite samples idealisation is not representative of few-shot settings and some phenomena are not observed in this setting. First, the superiority of model-agnostic methods is not apparent with an infinite number of samples per task. In that case, matrices $B$ only spanning $\text{col}(B_\star)$ also minimise the problem of Equation (12), potentially making Burer-Monteiro optimal despite misspecification. Second, a finite number of samples is required to unlearn the orthogonal of $\text{col}(B_\star)$. When $m_{\text{in}} = \infty$, FO-ANIL does not unlearn this subspace, which hurts the performance at test time for large $k'$, as observed in Section 5. Indeed, there is no risk of overfitting (and hence no need to unlearn the orthogonal space) with an infinite number of samples. On the contrary with a finite number of samples, FO-ANIL tends to overfit during its inner loop. This overfitting is yet penalized by the outer loss and leads to unlearning the orthogonal space.

Extending Theorem 1 to a finite number of tasks is left open for future work. Section 5 empirically supports that a similar result holds. A finite tasks and sample analysis similar to Collins et al. (2022) is not desirable, as mimicking the infinite samples case through concentration would omit the unlearning part, as explained above. With misspecification, we believe that extending Theorem 1 to a finite number of tasks is directly linked to relaxing Assumption 1. Indeed, the empirical task mean and covariance are not exactly $0$ and the identity matrix in that case. Obtaining a convergence result with general task mean and covariance would then help in understanding the finite tasks case.

**Limitations.** Assumption 1 assumes zero mean task parameters, $\mu_\star := \mathbb{E}[w_{\star,i}] = 0$. Considering non-zero task mean adds two difficulties to the existing analysis. First, controlling the dynamics of $w_t$ is much harder, as there is an extra term $\mu_\star$ in its update, but also $B_t w_t$ converges to $B_\star \mu_\star \neq 0$ instead. Moreover, updates of $B_t$ have an extra asymmetric rank 1 term depending on $\mu_\star$. Experiments in Appendix I yet support that both FO-ANIL and FO-MAML succeed when $\mu_\star \neq 0$.

In addition, we assume that the task covariance $\Sigma_\star$ is identity. The condition number of $\Sigma_\star$ is related to the *task diversity* and the problem hardness (Tripuraneni et al., 2020; Thekumparampil et al., 2021; Collins et al., 2022). Under Assumption 1, the task diversity is perfect (i.e., the condition number is 1), which simplifies the problem. The main challenge in dealing with general task covariances is that the updates involve non-commutative terms. Consequently, the main update rule of $B_\star^\top B_t B_t^\top B_\star$ no longer preserves the monotonicity used to derive upper and lower bounds on its iterates. However, experimental results in Section 5 suggest that Theorem 1 still holds with any diagonal covariance. Hence, we believe our analysis can be extended to any diagonal task covariance. The matrix $\Sigma_\star$ being diagonal is not restrictive, as it is always the case for a properly chosen $B_\star$.

Lastly, the features $X_i$ follow a standard Gaussian distribution here. It is needed to derive an exact expression of $\mathbb{E}[w_{t,i} w_{t,i}^\top]$ with Lemma 15, which can be easily extended to any spherically symmetric distribution. Whether Theorem 1 holds for general feature distributions yet remains open.

**Additional technical discussion.** For space reasons, we leave the technical details on Theorem 1 to Appendix A. In particular, we remark that our initialization only requires full-rank initialization without any initial alignment and describe how to derive a rate for the first limit in Theorem 1 which shows a slowdown due to overparametrization similar to the previous work by Xu & Du (2023).

## 5 EXPERIMENTS

This section empirically studies the behavior of model-agnostic methods on a toy example. We consider a setup with a large but finite number of tasks $N = 5000$, feature dimension $d = 50$, a limited number of samples per task $m = 30$, small hidden dimension $k = 5$ and Gaussian label noise with variance $\sigma^2 = 4$. We study a largely misspecified problem where $k' = d$. To demonstrate that Theorem 1 holds more generally, we consider a non-identity covariance $\Sigma_\star$ proportional to $\mathrm{diag}(1,\cdots,k)$. Further experimental details, along with additional experiments involving two-layer and three-layer ReLU networks, can be found in Appendix I.

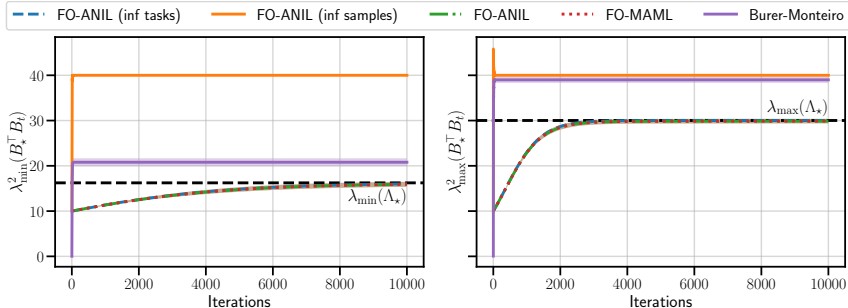

Figure 2: Evolution of smallest (*left*) and largest (*right*) squared singular value of $B_\star^\top B_t$ during training. The shaded area represents the standard deviation observed over 10 runs.

To observe the differences between the idealized models and the true algorithm, FO-ANIL with finite samples and tasks is compared with both its infinite tasks and infinite samples versions. It is also compared with FO-MAML and Burer-Monteiro factorization.

Figure 2 first illustrates how the different methods learn the ground truth subspace given by $B_\star$. More precisely, it shows the evolution of the largest and smallest squared singular value of $B_\star^\top B_t$. On the other hand, Figure 3 illustrates how different methods unlearn the orthogonal complement of $\mathrm{col}(B_\star)$, by showing the evolution of the largest and averaged squared singular value of $B_{\star,\perp}^\top B_t$.

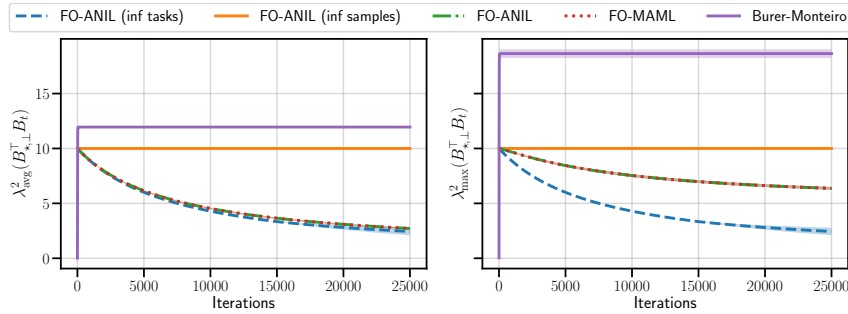

Figure 3: Evolution of average (*left*) and largest (*right*) squared singular value of $B_{\star,\perp}^\top B_t$ during training. The shaded area represents the standard deviation observed over 10 runs.

Finally, Table 1 compares the excess risks achieved by these methods on a new task with both 20 and 30 samples. The parameter is estimated by a ridge regression on $(X\hat{B}, y)$, where $\hat{B}$ is the representation learned while training. Additionally, we report the loss obtained for model-agnostic methods after a single gradient descent update. These methods are also compared with the single-task baseline that performs ridge regression on the $d$-dimensional feature space, and the oracle baseline that

directly performs ridge regression on the ground truth $k$-dimensional parameter space. Ridge regression is used for all methods, since regularizing the objective largely improves the test loss here. For each method, the regularization parameter is tuned using a grid-search over multiple values.

Table 1: Excess risk evaluated on 1000 testing tasks. The number after $\pm$ is the standard deviation over 10 independent training runs. For model-agnostic methods, `1-GD` refers to a single gradient descent step at test time; `Ridge` refers to ridge estimator with respect to the learned representation.

| | $m_{\text{test}} = 20$ | | $m_{\text{test}} = 30$ | |
| --- | --- | --- | --- | --- |
| Single-task ridge | $1.84 \pm 0.03$ | | $1.63 \pm 0.02$ | |
| Oracle ridge | $0.50 \pm 0.01$ | | $0.34 \pm 0.01$ | |
| Burer-Monteiro | $1.23 \pm 0.03$ | | $1.03 \pm 0.02$ | |
| | `1-GD` | `Ridge` | `1-GD` | `Ridge` |
| FO-ANIL | $0.81 \pm 0.01$ | $0.73 \pm 0.03$ | $0.64 \pm 0.01$ | $0.57 \pm 0.02$ |
| FO-MAML | $0.81 \pm 0.01$ | $0.73 \pm 0.04$ | $0.63 \pm 0.01$ | $0.58 \pm 0.01$ |
| FO-ANIL infinite tasks | $0.77 \pm 0.01$ | $0.67 \pm 0.03$ | $0.60 \pm 0.01$ | $0.52 \pm 0.01$ |
| FO-ANIL infinite samples | $1.78 \pm 0.02$ | $1.04 \pm 0.02$ | $1.19 \pm 0.01$ | $0.84 \pm 0.02$ |

As predicted by Theorem 1, FO-ANIL with infinite tasks exactly converges to $\Lambda_\star$. More precisely, it quickly learns the ground truth subspace and unlearns its orthogonal complement as the singular values of $B_{\star,\perp}^\top B_t$ decrease to 0, at the slow rate given in Appendix H. FO-ANIL and FO-MAML with a finite number of tasks, almost coincide. Although very close to infinite tasks FO-ANIL, they seem to unlearn the orthogonal space of $\text{col}(B_\star)$ even more slowly. In particular, there are a few directions (given by the maximal singular value) that are unlearned either very slowly or up to a small error. However on average, the unlearning happens at a comparable rate, and the effect of the few extreme directions is negligible. These methods thus learn a good representation and reach an excess risk approaching the oracle baseline with either ridge regression or just a single gradient step.

On the other hand, as predicted in Section 4, FO-ANIL with an infinite number of samples quickly learns $\text{col}(B_\star)$, but it does not unlearn the orthogonal complement. The singular values along the orthogonal complement stay constant. A similar behavior is observed for Burer-Monteiro factorization: the ground truth subspace is quickly learned, but the orthogonal complement is not unlearned. Actually, the singular values along the orthogonal complement even increase during the first steps of training. For both methods, the inability of unlearning the orthogonal complement significantly hurts the performance at test time. Note however that they still outperform the single-task baseline. The singular values along $\text{col}(B_\star)$ are indeed larger than along its orthogonal complement. More weight is then put on the ground truth subspace when estimating a new task.

These experiments confirm the phenomena described in Sections 3 and 4. Model-agnostic methods not only learn the good subspace, but also unlearn its orthogonal complement. This unlearning yet happens slowly and many iterations are required to completely ignore the orthogonal space.

## 6 Conclusion

This work studies first-order ANIL in the shared linear representation model with a linear two-layer architecture. Under infinite tasks idealisation, FO-ANIL successfully learns the shared, low-dimensional representation despite overparametrization in the hidden layer. More crucially for performance during task-specific finetuning, the iterates of FO-ANIL not only learn the low-dimensional subspace but also forget its orthogonal complement. Consequently, a single-step gradient descent initialized on the learned parameters achieves a small excess risk on any given new task. Numerical experiments confirm these results and suggest they hold in more general setups, e.g., with uncentered, anisotropic task parameters and a finite number of tasks. As a consequence, our work suggests that model-agnostic methods are also *model-agnostic* in the sense that they successfully learn the shared representation, although their architecture is not adapted to the problem parameters. Extending our theoretical results to these more general settings or more intricate methods, such as MAML, remains open for future work. Lastly, our work presents a provable shared representation learning result for the pretraining of meta-learning algorithms. Thus, it connects to the literature on representation learning with pretraining; in particular, it demonstrates a slowdown due to overparametrization that has been recently demonstrated in supervised learning.

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

## A  ADDITIONAL DISCUSSION

**Initialization regime.**    Theorem 1 requires a bounded initialization to ensure the dynamics of FO-ANIL stay bounded. Roughly, we need the squared norm of $B_{\star,\perp}^\top B_0$ to be $\mathcal{O}\left((\alpha m_{\text{in}})^{-1}\right)$ to guarantee $\|B_t\|_2 \leq \alpha^{-1}$ for any $t$. We believe the $m_{\text{in}}$ dependency is an artifact of the analysis and it is empirically not needed. Additionally, we bound $w_0$ to control the scale of $\mathbb{E}[w_{t,i} w_{t,i}^\top]$ that appears in the update of $B_t$. A similar inductive condition is used by Collins et al. (2022).

More importantly, our analysis only needs a full rank $B_\star^\top B_0$, which holds almost surely for usual initializations. Collins et al. (2022) instead require that the smallest eigenvalue of $B_\star^\top B_0$ is bounded strictly away from $0$, which does not hold when $d \gg k'$. This indicates that their analysis covers only the tail end of training and not the initial alignment phase.

**Rate of convergence.**    In contrast with the convergence result of Collins et al. (2022), Theorem 1 does not provide any convergence rate for FO-ANIL but only states asymptotic results. Appendix H provides an analogous rate for the first limit of Theorem 1: $\|B_{\star,\perp}^\top B_t\|_2^2 = \mathcal{O}\left(\frac{m_{\text{in}}}{\alpha^2 \beta \overline{\sigma}^2 t}\right)$. Due to misspecification, this rate is slower than the one by Collins et al. (2022) (exponential vs. polynomial). A similar slow down due to overparametrization has been recently shown when learning a single ReLU neuron (Xu & Du, 2023). In our setting, rates are more difficult to obtain for the second and third limits, as the decay of quantities of interest depends on other terms in complex ways. Remark that rates for these two limits are not studied by Collins et al. (2022). In the infinite samples limit, a rate for the third limit can yet be derived when $k = k'$.

**Relaxation to a finite number of tasks.**    In the limit of infinite tasks, Theorem 1 proves that the asymptotic solution is low-rank and the complement $B_{\star,\perp}$ is unlearned. Figure 3 shows this behavior with a relatively big initialization. For pretraining with a finite number of tasks, the decay is not towards $0$ but to a small residual value. The scale of this residual depends on the number of tasks (decreasing to $0$ when the number of tasks goes to infinity) and possibly on other problem parameters such as $k$ and $\alpha$. Most notably, when a very small initialization is chosen, the singular values in the complement space can increase until this small scale.

This indicates that Theorem 1 relaxed to a finite number of tasks will include a non-zero but small component in the orthogonal space. In our experiments, these residuals do not hurt the finetuning performance. Note that these residuals are also present for simulations of FO-ANIL with infinite tasks due to the finite time horizon.

**Theoretical analyzes of model-agnostic meta-learning.**    Other theoretical works on model-agnostic meta-learning have focused on convergence guarentees (Fallah et al., 2020; Ji et al., 2022) or generalization (Fallah et al., 2021). In contrast, this work focuses on pretraining of model-agnostic meta-learning and learning of shared representations under the canonical model of multi-task learning.

# B  SKETCH OF PROOF

The challenging part of Theorem 1 is that $B_t \in \mathbb{R}^{d \times k'}$ involves two separate components with different dynamics:

$$B_t = B_\star B_\star^\top B_t + B_{\star,\perp} B_{\star,\perp}^\top B_t.$$

The first term $B_\star^\top B_t$ eventually scales in $\alpha^{-1/2}$ whereas the second term $B_{\star,\perp}^\top B_t$ converges to 0, resulting in a nearly rank-deficient $B_t$. The dynamics of these two terms and $w_t$ are interdependent, which makes it challenging to bound any of them.

**Regularity conditions.**   The first part of the proof consists in bounding all the quantities of interest. Precisely, we show by induction that the three following properties hold for any $t$,

    1. $\|B_\star^\top B_t\|_2^2 \le \|\Lambda_\star\|_2$,          2. $\|w_t\|_2 \le \|w_0\|_2$,         3. $\|B_{\star,\perp}^\top B_t\|_2 \le \|B_{\star,\perp}^\top B_0\|_2$.   (13)

Importantly, the first and third conditions, along with the initialization conditions, imply $\|B_t\|_2^2 \le \alpha^{-1}$. The monotonicity of the function $f^U$ described below leads to $\|B_\star^\top B_{t+1}\|_2^2 \le \|\Lambda_\star\|_2$. Also, using the inductive assumptions with the update equations for $B_{\star,\perp}^\top B_t$ and $w_t$ allows us to show that both the second and third properties hold at time $t+1$.

Now that the three different quantities of interest have been properly bounded, we can show the three limiting results of Theorem 1.

**Unlearning the orthogonal complement.**   We first show that $\lim_{t\to\infty} B_{\star,\perp}^\top B_t = 0$. Equation (6) directly yields $B_{\star,\perp}^\top B_{t+1} = B_{\star,\perp}^\top B_t \left(\mathbf{I}_{k'} - \beta \mathbb{E}[w_{t,i} w_{t,i}^\top]\right)$. The previous bounding conditions guarantee for a well chosen $\beta$ that $\|\mathbb{E}[w_{t,i} w_{t,i}^\top]\|_2 \le \beta^{-1}$. Moreover thanks to Equation (7), $\mathbb{E}[w_{t,i} w_{t,i}^\top] \succeq \alpha^2 \frac{\bar{\sigma}^2}{m_{\text{in}}} B_{\star,\perp}^\top B_t B_t^\top B_{\star,\perp}$, which finally yields

$$\|B_{\star,\perp}^\top B_{t+1}\|_2^2 \le \left(1 - \alpha^2 \beta \frac{\bar{\sigma}^2}{m_{\text{in}}} \|B_{\star,\perp}^\top B_t\|_2^2\right) \|B_{\star,\perp}^\top B_t\|_2^2. \tag{14}$$

**Learning the task mean.**   We can now proceed to the second limit in Theorem 1. $B_t w_t$ can be decomposed into two parts, giving $\|B_t w_t\|_2 \le \|B_\star^\top B_t w_t\|_2 + \|B_{\star,\perp}^\top B_t w_t\|_2$. As $\|w_t\|_2$ is bounded and $\|B_{\star,\perp}^\top B_t\|_2$ converges to 0, the second term vanishes. A detailed analysis on the updates of $B_\star^\top B_t w_t$ gives

$$\|B_\star^\top B_{t+1} w_{t+1}\|_2 \le \left(1 - \frac{\beta}{4\alpha} + \alpha\beta \|\Sigma_\star\|_2\right) \|B_\star^\top B_t w_t\|_2 + \mathcal{O}\left(\|B_{\star,\perp}^\top B_t\|_2^2 \|w_t\|_2\right),$$

which implies that $\lim_{t\to\infty} B_t w_t = 0$ for properly chosen $\alpha, \beta$.

**Feature learning.**   We now focus on the limit of the matrix $\Lambda_t := B_\star^\top B_t B_t^\top B_\star \in \mathbb{R}^{k \times k}$. The recursion on $\Lambda_t$ induced by Equations (5) and (6) is as follows,

$$\begin{aligned}
\Lambda_{t+1} = {} & \left(\mathbf{I}_k + \alpha\beta R_t(\Lambda_t)\right) \Lambda_t \left(\mathbf{I}_k + \alpha\beta R_t(\Lambda_t)\right) \\
& - 2\beta \text{Sym}\left(\left(\mathbf{I}_k + \alpha\beta R_t(\Lambda_t)\right) B_\star^\top B_t U_t B_t^\top B_\star\right) + \beta^2 B_\star^\top B_t U_t^2 B_t^\top B_\star.
\end{aligned} \tag{15}$$

where $\text{Sym}(A) := \frac{1}{2}\left(A + A^\top\right)$, $R_t(\Lambda_t) := \left(\mathbf{I}_k - \frac{\alpha(m_{\text{in}}+1)}{m_{\text{in}}}\Lambda_t\right)\Sigma_\star - \frac{\alpha}{m_{\text{in}}}\left(\bar{\sigma}^2 + \|B_t w_t\|_2^2\right)$ and $U_t$ is some noise term defined in Appendix C. From there, we can define functions $f_t^L$ and $f^U$ approximating the updates given in Equation (15) such that

$$f_t^L(\Lambda_t) \preceq \Lambda_{t+1} \preceq f^U(\Lambda_t).$$

Moreover, these functions preserve the Loewner matrix order for commuting matrices of interest. Thanks to that, we can construct bounding sequences of matrices $(\Lambda_t^L), (\Lambda_t^U)$ such that

    1. $\Lambda_{t+1}^L = f_t^L(\Lambda_t^L)$,         2. $\Lambda_{t+1}^U = f^U(\Lambda_t^U)$,         3. $\Lambda_t^L \preceq \Lambda_t \preceq \Lambda_t^U$.

Using the first two points, we can then show that both sequences $\Lambda_t^L, \Lambda_t^U$ are non-decreasing and converge to $\Lambda_\star$ under the conditions of Theorem 1. The third point then concludes the proof.

## C    PROOF OF THEOREM 1

The full version of Theorem 1 is given by Theorem 2. In particular, it gives more precise conditions on the required initialization and step sizes.

**Theorem 2.** *Assume that $c_1 < 1, c_2$ are small enough positive constants verifying*

$$c_2 \frac{m_{\text{in}} + 1}{m_{\text{in}}} + \frac{c_1 \left(c_2 + \bar{\sigma}^2\right)}{2m_{\text{in}}(m_{\text{in}} + 1)} < \lambda_{\min}(\Sigma_\star),$$

*and $\alpha, \beta$ are selected such that the following conditions hold:*

1. $\beta \leq \alpha$,

2. $\dfrac{1}{\alpha^2} \geq 4\|\Sigma_\star\|_2$,

3. $\dfrac{1}{\alpha\beta} \geq \left(c_2 \dfrac{m_{\text{in}} + 2}{m_{\text{in}}} + \dfrac{c_1 c_2}{2m_{\text{in}}(m_{\text{in}} + 1)} + \dfrac{2\bar{\sigma}^2}{m_{\text{in}}} + \dfrac{m_{\text{in}} + 1}{m_{\text{in}}}\|\Sigma_\star\|_2 + \dfrac{4}{3} \dfrac{m_{\text{in}}}{(m_{\text{in}} + 1)^2}\right)$,

4. $\dfrac{1}{\alpha\beta} \geq 6\left(\|\Sigma_\star\|_2 + \dfrac{c_2 + \bar{\sigma}^2}{m_{\text{in}} + 1}\right)$.

*Furthermore, suppose that parameters $B_0$ and $w_0$ are initialized such that the following three conditions hold:*

1. $B_\star^\top B_0$ *is full rank,*

2. $\|B_0\|_2^2 \leq \dfrac{1}{\alpha} \dfrac{c_1}{m_{\text{in}} + 1}$,

3. $\|w_0\|_2^2 \leq \alpha c_2$.

*Then, FO-ANIL (given by Equations (5) and (6)) with initial parameters $B_0, w_0$, inner step size $\alpha$, outer step size $\beta$, asymptotically satisfies the following*

$$\lim_{t \to \infty} B_{\star,\perp}^\top B_t = 0, \tag{16}$$

$$\lim_{t \to \infty} B_t w_t = 0, \tag{17}$$

$$\lim_{t \to \infty} B_\star^\top B_t B_t^\top B_\star = \Lambda_\star = \frac{1}{\alpha} \frac{m_{\text{in}}}{m_{\text{in}} + 1} \left(\mathbf{I}_k - \left(\frac{m_{\text{in}} + 1}{\bar{\sigma}^2}\Sigma_\star + \mathbf{I}_k\right)^{-1}\right). \tag{18}$$

The main tools for the proof are presented and discussed in the following subsections. Section C.1 proves monotonic decay in noise terms provided that $B_t$ is bounded by above. Section C.2 provides bounds for iterates and describes the monotonicity between updates. Section C.3 constructs sequences that bound the iterates from above and below. Section C.4 presents the full proof using the tools developed in previous sections. In the following, common recursions on relevant objects are derived.

The recursion on $B_t$ defined in Equation (6) leads to the following recursions on $C_t := B_\star^\top B_t \in \mathbb{R}^{k \times k'}$ and $D_t := B_{\star,\perp}^\top B_t \in \mathbb{R}^{(d-k) \times k'}$,

$$
C_{t+1} = \left( \mathbf{I}_k + \alpha\beta \left( \mathbf{I}_k - \frac{\alpha(m_{\text{in}}+1)}{m_{\text{in}}} C_t C_t^\top \right) \Sigma_\star - \frac{\alpha^2 \beta}{m_{\text{in}}} \left( \|B_t w_t\|^2 + \text{Tr}(\Sigma_\star) + \sigma^2 \right) C_t C_t^\top \right) C_t
$$
$$
- \beta C_t \left[ \left( \mathbf{I}_{k'} - \alpha B_t^\top B_t \right) w_t w_t^\top \left( \mathbf{I}_{k'} - \alpha B_t^\top B_t \right) + \frac{\alpha^2}{m_{\text{in}}} B_t^\top B_t w_t w_t^\top B_t^\top B_t \right.
$$
$$
\left. + \frac{\alpha^2}{m_{\text{in}}} \left( \|B_t w_t\|^2 + \text{Tr}(\Sigma_\star) + \sigma^2 \right) D_t^\top D_t \right], \tag{19}
$$

$$
D_{t+1} = D_t \left[ \mathbf{I}_{k'} - \beta \left( \mathbf{I}_{k'} - \alpha B_t^\top B_t \right) w_t w_t^\top \left( \mathbf{I}_{k'} - \alpha B_t^\top B_t \right) - \frac{\alpha^2 \beta}{m_{\text{in}}} B_t^\top B_t w_t w_t^\top B_t^\top B_t \right.
$$
$$
\left. - \frac{\alpha^2 \beta (m_{\text{in}}+1)}{m_{\text{in}}} C_t^\top \Sigma_\star C_t - \frac{\alpha^2 \beta}{m_{\text{in}}} \left( \|B_t w_t\|^2 + \text{Tr}(\Sigma_\star) + \sigma^2 \right) B_t^\top B_t \right]. \tag{20}
$$

For ease of notation, let $\bar{\sigma}^2 := \text{Tr}(\Sigma_\star) + \sigma^2$, $\delta_t := \|B_t w_t\|_2^2 + \bar{\sigma}^2$ and define the following objects,

$$
R(\Lambda, \tau) := \left( \mathbf{I}_k - \frac{\alpha(m_{\text{in}}+1)}{m_{\text{in}}} \Lambda \right) \Sigma_\star - \frac{\alpha}{m_{\text{in}}} \left( \bar{\sigma}^2 + \tau \right) \Lambda, \quad R_t(\Lambda) := R(\Lambda, \|B_t w_t\|_2^2),
$$

$$
W_t := \left( \mathbf{I}_{k'} - \alpha B_t^\top B_t \right) w_t w_t^\top \left( \mathbf{I}_{k'} - \alpha B_t^\top B_t \right) + \frac{\alpha^2}{m_{\text{in}}} B_t^\top B_t w_t w_t^\top B_t^\top B_t,
$$

$$
U_t := W_t + \frac{\alpha^2}{m_{\text{in}}} \delta_t D_t^\top D_t,
$$

$$
V_t := W_t + \frac{\alpha^2(m_{\text{in}}+1)}{m_{\text{in}}} C_t^\top \Sigma_\star C_t + \frac{\alpha^2}{m_{\text{in}}} \delta_t B_t^\top B_t. \tag{21}
$$

Then, the recursion for $\Lambda_t := C_t C_t^\top$ is

$$
\Lambda_{t+1} = \left( \mathbf{I}_k + \alpha\beta R_t(\Lambda_t) \right) \Lambda_t \left( \mathbf{I}_k + \alpha\beta R_t(\Lambda_t) \right)^\top + \beta^2 C_t U_t^2 C_t^\top
$$
$$
- \beta \left( \mathbf{I}_k + \alpha\beta R_t(\Lambda_t) \right) C_t U_t C_t^\top - \beta C_t U_t C_t^\top \left( \mathbf{I}_k + \alpha\beta R_t(\Lambda_t) \right)^\top. \tag{22}
$$

### C.1 REGULARITY CONDITIONS

Lemmas 1 and 2 control $\|w_t\|_2$ and $\|D_t\|_2$ across iterations, respectively. Lemma 3 shows that $\|C_t w_t\|_2$ is decaying with a noise term that vanishes as $\|D_t\|_2$ gets small. Corollary 1 combines all three results and yields the first two claims of Theorem 1,

$$
\lim_{t \to \infty} B_{\star,\perp} B_t = 0, \quad \lim_{t \to \infty} B_t w_t = 0,
$$

under the assumption that conditions of Lemmas 2 and 3 are satisfied for all $t$. Lemmas 4 and 5 bound $\|U_t\|_2$ and $\|W_t\|_2$, ensuring that the recursions of $\Lambda_t$ are well-behaved in later sections.

**Lemma 1.** *Assume that*

$$
c_0 \mathbf{I}_{k'} \preceq B_t^\top B_t \preceq \frac{1}{\alpha} \frac{m_{\text{in}} + c_1}{m_{\text{in}} + 1} \mathbf{I}_{k'},
$$

*for constants $0 \le c_0, 0 < c_1 < 1$ such that $\beta c_0(1 - c_1) \le m_{\text{in}} + 1$. Then,*

$$
\|w_{t+1}\|_2 \le \left( 1 - \beta \frac{c_0(1 - c_1)}{m_{\text{in}} + 1} \right) \|w_t\|_2.
$$

**Proof.** From the assumption,

$$
\frac{1 - c_1}{m_{\text{in}} + 1} \mathbf{I}_{k'} \preceq \mathbf{I}_{k'} - \alpha B_t^\top B_t \preceq (1 - \alpha c_0) \mathbf{I}_{k'},
$$

and

$$
B_t^\top B_t (\mathbf{I}_{k'} - \alpha B_t^\top B_t) \succeq \frac{c_0(1 - c_1)}{m_{\text{in}} + 1} \mathbf{I}_{k'}.
$$

Recalling the recursion for $w_t$ defined in Equation (5),

$$\|w_{t+1}\|_2 \leq \left(1 - \beta \frac{c_0(1 - c_1)}{m_{\text{in}} + 1}\right) \|w_t\|_2.$$

**Lemma 2.** *Assume that*

$$\|B_t\|_2^2 \leq \frac{1}{\alpha}, \quad \|w_t\|_2^2 \leq c\alpha,$$

*for a constant $c \geq 0$ and $\alpha, \beta$ satisfy*

$$\frac{1}{\alpha\beta} \geq \frac{m_{\text{in}} + 2}{m_{\text{in}}} c + \frac{1}{m_{\text{in}}} \left((m_{\text{in}} + 1)\|\Sigma_\star\|_2 + \bar{\sigma}^2\right), \tag{23}$$

$$\frac{1}{\alpha\beta} \geq \frac{2\bar{\sigma}^2}{m_{\text{in}}}. \tag{24}$$

*Then,*

$$\|D_{t+1} D_{t+1}^\top\|_2 \leq \left(1 - \frac{\alpha^2 \beta}{m_{\text{in}}} \bar{\sigma}^2 \|D_t D_t^\top\|_2\right) \|D_t D_t^\top\|_2.$$

**Proof.** The recursion on $D_t D_t^\top$ is given by

$$D_{t+1} D_{t+1}^\top = D_t (\mathbf{I}_{k'} - \beta V_t)^2 D_t^\top,$$

where we recall $V_t$ is defined in Equation (21). First step is to show $\mathbf{I}_{k'} - \beta V_t \succeq 0$ by proving $\|V_t\|_2 \leq \frac{1}{\beta}$. By the definition of $V_t$,

$$\|V_t\|_2 \leq \underbrace{\|W_t\|_2}_{(A)} + \underbrace{\frac{\alpha^2(m_{\text{in}} + 1)}{m_{\text{in}}} \|C_t^\top \Sigma_\star C_t\|_2}_{(B)} + \underbrace{\frac{\alpha^2}{m_{\text{in}}} \delta_t \|B_t^\top B_t\|_2}_{(C)}.$$

Term (A) is bounded by Lemma 5. For the term (B), using $\|C_t\|_2 = \|B_\star^\top B_t\|_2 \leq \|B_t\|_2$,

$$\|C_t^\top \Sigma_\star C_t\|_2 \leq \frac{1}{\alpha} \|\Sigma_\star\|_2.$$

Term (C) is bounded as

$$\delta_t = \|B_t w_t\|_2^2 + \bar{\sigma}^2 \leq \frac{1}{\alpha} \|w_t\|_2^2 + \bar{\sigma}^2 \leq c + \bar{\sigma}^2, \quad \|B_t^\top B_t\|_2 \leq \|B_t\|_2^2 \leq \frac{1}{\alpha}.$$

Combining three bounds and using the condition in Equation (23),

$$\|V_t\|_2 \leq \frac{m_{\text{in}} + 1}{m_{\text{in}}} \alpha c + \frac{\alpha}{m_{\text{in}}} \left((m_{\text{in}} + 1)\|\Sigma_\star\|_2 + \bar{\sigma}^2\right) + \alpha c \leq \frac{1}{\beta}.$$

Therefore, it is possible to upper bound $D_{t+1} D_{t+1}^\top$ as follows,

$$\begin{aligned}
D_{t+1} D_{t+1}^\top &= D_t (\mathbf{I}_{k'} - \beta V_t)^2 D_t^\top \\
&\preceq D_t (\mathbf{I}_{k'} - \beta V_t) D_t^\top \\
&\preceq D_t \left[\mathbf{I}_{k'} - \frac{\alpha^2 \beta}{m_{\text{in}}} \bar{\sigma}^2 D_t^\top D_t\right] D_t^\top \\
&= \left[\mathbf{I}_{k'} - \frac{\alpha^2 \beta}{m_{\text{in}}} \bar{\sigma}^2 D_t D_t^\top\right] D_t D_t^\top.
\end{aligned}$$

Let $D_t D_t^\top = \Omega_t S_t \Omega_t^\top$ be the SVD decomposition of $D_t D_t^\top$ in this proof. Then,

$$D_{t+1} D_{t+1}^\top \preceq \Omega_t \left(S_t - \frac{\alpha^2 \beta}{m_{\text{in}}} \bar{\sigma}^2 S_t^2\right) \Omega_t^\top.$$

Note that $\frac{1}{\alpha} < \frac{m_{\text{in}}}{2\alpha^2 \beta \bar{\sigma}^2}$ by Equation (24) and for any $s_1 \leq s_2 < \frac{1}{\alpha} < \frac{m_{\text{in}}}{2\alpha^2 \beta \bar{\sigma}^2}$,

$$s_2 \left(1 - \frac{\alpha^2 \beta}{m_{\text{in}}} \bar{\sigma}^2 s_2\right) \geq s_1 \left(1 - \frac{\alpha^2 \beta}{m_{\text{in}}} \bar{\sigma}^2 s_1\right),$$

by monotonicity of $x \mapsto x(1 - \frac{\alpha^2 \beta}{m_{\text{in}}} \bar{\sigma}^2 x)$. Hence, if $s$ is the largest eigenvalue of $S_t$, $s(1 - \frac{\alpha^2 \beta}{m_{\text{in}}} \bar{\sigma}^2 s)$ is the largest eigenvalue of $(S_t - \frac{\alpha^2 \beta}{m_{\text{in}}} \bar{\sigma}^2 S_t^2)$ and

$$\|D_{t+1} D_{t+1}^\top\|_2 \leq \left(1 - \frac{\alpha^2 \beta}{m_{\text{in}}} \bar{\sigma}^2 \|D_t D_t^\top\|_2\right) \|D_t D_t^\top\|_2.$$

**Lemma 3.** *Suppose that $\beta \leq \alpha$ and the following conditions hold,*

$$\|B_t\|_2^2 \leq \frac{1}{\alpha}, \|\Lambda_t\|_2^2 \leq \frac{1}{\alpha} \frac{m_{\text{in}}}{m_{\text{in}} + 1}, \quad \|w_t\|_2^2 \leq \alpha c,$$

*where $c \geq 0$ is a constant such that*

$$\frac{\left(1 - \frac{\beta}{4\alpha}\right)}{\alpha \beta} \geq \frac{\bar{\sigma}^2}{m_{\text{in}} + 1} + c \frac{m_{\text{in}} + 2}{m_{\text{in}} + 1} + \frac{m_{\text{in}}}{(m_{\text{in}} + 1)^2}.$$

*Then,*

$$\|C_{t+1} w_{t+1}\|_2 \leq \left(1 - \frac{\beta}{4\alpha} + \alpha \beta \|\Sigma_\star\|_2\right) \|C_t w_t\|_2 + M \|D_t\|_2^2 \|w_t\|_2,$$

*for a constant $M$ depending only on $\alpha$.*

**Proof.** Let $\Omega_t := C_t^\top C_t$. Expanding the recursion for $w_{t+1}$,

$$C_{t+1} w_{t+1} = \underbrace{C_{t+1} \left(\mathbf{I}_{k'} - \beta \Omega_t + \alpha \beta \Omega_t^2\right) w_t}_{(A)}$$
$$+ \alpha \beta \underbrace{C_{t+1} \left(D_t^\top D_t - 2\alpha D_t^\top D_t - \alpha^2 D_t^\top D_t \Omega_t - \alpha^2 \Omega_t D_t^\top D_t - \alpha^2 D_t^\top D_t D_t^\top D_t\right) w_t}_{(B)}.$$

Since $\|B_t\|_2^2 \leq \frac{1}{\alpha}$, there is some constant $M_B$ depending only on $\alpha$ such that

$$M_B \|D_t\|_2^2 \|w_t\|_2 \geq \|(B)\|_2.$$

Expanding term (A),

$$C_{t+1} \left(\mathbf{I}_{k'} - \beta \Omega_t + \alpha \beta \Omega_t^2\right) w_t = \alpha \beta \underbrace{\left(I - \alpha \frac{m_{\text{in}} + 1}{m_{\text{in}}} \Lambda_t\right) \Sigma_\star C_t \left(\mathbf{I}_{k'} - \beta \Omega_t + \alpha \beta \Omega_t^2\right) w_t}_{(C)}$$
$$- \underbrace{C_t \left(\mathbf{I}_{k'} - \frac{\alpha^2 \beta}{m_{\text{in}}} \delta_t \Omega_t - \beta (\mathbf{I}_{k'} - \alpha \Omega_t) w_t w_t^\top (\mathbf{I}_{k'} - \alpha \Omega_t) - \frac{\alpha^2 \beta}{m_{\text{in}}} \Omega_t w_t w_t^\top \Omega_t\right) \left(\mathbf{I}_{k'} - \beta \Omega_t + \alpha \beta \Omega_t^2\right) w_t}_{(D)}$$
$$+ \alpha \beta \underbrace{C_t \left(D_t^\top D_t w_t w_t^\top (\mathbf{I}_{k'} - \bar{\alpha} \Omega_t) + (\mathbf{I}_{k'} - \bar{\alpha} \Omega_t) w_t w_t^\top D_t^\top D_t - \bar{\alpha} D_t^\top D_t w_t w_t^\top D_t^\top D_t\right)}_{(E)},$$

where $\bar{\alpha} := \alpha \frac{m_{\text{in}} + 1}{m_{\text{in}}}$. Similarly to term (B), there is a constant $M_E$ depending only on $\alpha$ such that

$$M_E \|D_t\|_2^2 \|w_t\|_2^2 \geq \|(E)\|_2.$$

Bounding term (C),

$$\|(C)\|_2 = \left\| \left(I - \alpha \frac{m_{\text{in}} + 1}{m_{\text{in}}} \Lambda_t\right) \Sigma_\star \left(\mathbf{I}_k - \beta \Lambda_t + \alpha \beta \Lambda_t^2\right) C_t w_t \right\|_2$$
$$\leq \|I - \alpha \frac{m_{\text{in}} + 1}{m_{\text{in}}} \Lambda_t\|_2 \|\Sigma_\star\|_2 \|\mathbf{I}_k - \beta \Lambda_t + \alpha \beta \Lambda_t^2\|_2 \|C_t w_t\|_2$$
$$= \|\Sigma_\star\|_2 \left(1 - \alpha \frac{m_{\text{in}} + 1}{m_{\text{in}}} \lambda_k(\Lambda_t)\right) \left(1 - \beta \lambda_k(\Lambda_t) + \alpha \beta \lambda_k(\Lambda_t)^2\right) \|C_t w_t\|_2.$$

Re-writing term (D),

$$(D) = \underbrace{\left(\left(\mathbf{I}_k - \frac{\alpha^2\beta}{m_{\text{in}}}\delta_t\Lambda_t\right)\left(\mathbf{I}_k - \beta\Lambda_t + \alpha\beta\Lambda_t^2\right) - \beta d_1\left(\mathbf{I}_{k'} - \alpha\Lambda_t\right) - \frac{\alpha^2\beta}{m_{\text{in}}}d_2\Lambda_t\right)}_{(F)}C_t w_t$$

where $d_1$ and $d_2$ are defined as

$$d_1 := \big\langle\left(\mathbf{I}_{k'} - \alpha\Omega_t\right)w_t, \left(\mathbf{I}_{k'} - \beta\Omega_t + \alpha\beta\Omega_t^2\right)w_t\big\rangle, \quad d_2 := \big\langle\Omega_t w_t, \left(\mathbf{I}_k - \beta\Omega_t + \alpha\beta\Omega_t^2\right)w_t\big\rangle.$$

As all eigenvalues of $\Omega_t$ are in $\left[0, \frac{1}{\alpha}\frac{m_{\text{in}}}{m_{\text{in}}+1}\right]$,

$$\left(1 - \frac{\beta}{4\alpha}\right)\mathbf{I}_{k'} \preceq \mathbf{I}_{k'} - \beta\Omega_t + \alpha\beta\Omega_t^2 \preceq \mathbf{I}_{k'},$$

and

$$\frac{1}{m_{\text{in}}+1}\left(1 - \frac{\beta}{4\alpha}\right) \preceq \left(\mathbf{I}_{k'} - \alpha\Omega_t\right)\left(\mathbf{I}_{k'} - \beta\Omega_t + \alpha\beta\Omega_t^2\right) \preceq \mathbf{I}_{k'},$$

$$0 \preceq \Omega_t\left(\mathbf{I}_k - \beta\Omega_t + \alpha\beta\Omega_t^2\right) \preceq \frac{1}{\alpha}\frac{m_{\text{in}}}{m_{\text{in}}+1}.$$

Therefore, $d_1$ and $d_2$ are non-negative and bounded from above as follows,

$$\frac{\alpha c\left(1 - \frac{\beta}{4\alpha}\right)}{m_{\text{in}}+1} \le d_1 \le \alpha c, \quad 0 \le d_2 \le \frac{c m_{\text{in}}}{m_{\text{in}}+1}.$$

By assumptions,

$$\frac{\alpha^2\beta}{m_{\text{in}}}\delta_t\big\|\Lambda_t\left(\mathbf{I}_k - \beta\Lambda_t + \alpha\beta\Lambda_t^2\right)\big\|_2 \le \frac{\alpha\beta}{m_{\text{in}}+1}\delta_t \le \frac{\alpha\beta\left(c + \bar{\sigma}^2\right)}{m_{\text{in}}+1},$$

and combining all the negative terms in (F),

$$\frac{\alpha^2\beta}{m_{\text{in}}}\delta_t\Lambda_t\left(\mathbf{I}_k - \beta\Lambda_t + \alpha\beta\Lambda_t^2\right) + \beta d_1\left(\mathbf{I}_{k'} - \alpha\Lambda_t\right) + \frac{\alpha^2\beta}{m_{\text{in}}}d_2\Lambda_t \preceq \alpha\beta\left(\frac{c + \bar{\sigma}^2}{m_{\text{in}}+1} + c + \frac{m_{\text{in}}}{(m_{\text{in}}+1)^2}\right)\mathbf{I}_{k'}.$$

Hence, (F) is bounded by below and above,

$$0 \preceq (F) \preceq \left(\mathbf{I}_k - \beta\Lambda_t + \alpha\beta\Lambda_t^2\right).$$

Thus, the norm of (F) is bounded by above,

$$\|(F)\|_2 \le \left(1 - \frac{\beta}{4\alpha}\right).$$

Combining all the bounds,

$$\|C_{t+1}w_{t+1}\|_2 \le \left(1 - \frac{\beta}{4\alpha} + \alpha\beta\|\Sigma_\star\|_2\right)\|C_t w_t\|_2 + M\|D_t\|_2^2\|w_t\|_2,$$

where $M$ is a constant depending only on $\alpha$.

**Corollary 1.** *Assume that conditions of Lemma 2 are satisfied for a fixed $c > 0$ for all times $t$. Then, Lemma 2 directly implies that*

$$\lim_{t\to\infty} B_{\star,\perp}^\top B_t = \lim_{t\to\infty} D_t = 0.$$

*Further, assume that conditions of Lemma 3 is satisfied for all times $t$ and*

$$\frac{1}{\alpha^2} \ge 4\|\Sigma_\star\|_2. \tag{25}$$

*Then, Lemmas 2 and 3 together imply that*

$$\lim_{t\to\infty} \|B_t w_t\|_2 = 0.$$

**Proof.** The first result directly follows as by Lemma 2,

$$\lim_{t\to\infty} \|D_t\|_2 = 0.$$

Hence, for any $\epsilon > 0$, there exist a $t_\epsilon$ such that

$$\forall t > t_\epsilon, \quad \|D_t\|_2 < \frac{\epsilon}{\sqrt{c\alpha}}.$$

Observe that for any $t$,

$$\|B_{t+1}w_{t+1}\|_2 \leq \|B_\star C_{t+1}w_{t+1}\|_2 + \|B_{\star,\perp}D_{t+1}w_{t+1}\|_2 = \|C_{t+1}w_{t+1}\|_2 + \|D_{t+1}w_{t+1}\|_2.$$

Therefore, by Lemma 3, for any $t > t_\epsilon$,

$$\|B_{t+1}w_{t+1}\|_2 \leq \left(1 - \frac{\beta}{4\alpha} + \alpha\beta\|\Sigma_\star\|_2\right)\|C_t w_t\|_2 + \epsilon^2\frac{M}{\sqrt{c\alpha}} + \epsilon$$

$$\leq \left(1 - \frac{\beta}{4\alpha} + \alpha\beta\|\Sigma_\star\|_2\right)\|B_t w_t\|_2 + \epsilon^2\frac{M}{\sqrt{c\alpha}} + \epsilon.$$

By Equation (25),

$$\left(1 - \frac{\beta}{4\alpha} + \alpha\beta\|\Sigma_\star\|_2\right) < 1,$$

and $\|B_t w_t\|_2$ is decaying for $t > t_\epsilon$ as long as

$$\|B_t w_t\|_2 \geq \frac{\epsilon\left(1 + \epsilon\frac{M}{\sqrt{c\alpha}}\right)}{\alpha\beta\left(\frac{1}{4\alpha^2} - \|\Sigma_\star\|_2\right)}.$$

Hence, for any $\epsilon' > 0$, it is possible to find $t_{\epsilon'} > t_\epsilon$ such that for all $t > t_{\epsilon'}$,

$$\|B_t w_t\|_2 \leq \frac{\epsilon\left(1 + \epsilon\frac{M}{\sqrt{c\alpha}}\right)}{\alpha\beta\left(\frac{1}{4\alpha^2} - \|\Sigma_\star\|_2\right)} + \epsilon'.$$

As $\epsilon$ and $\epsilon'$ are arbitrary,

$$\lim_{t\to\infty} \|B_t w_t\|_2 = 0.$$

**Lemma 4.** *Assume that* $\|D_t\|_2^2 \leq \frac{1}{\alpha}\frac{c_1}{2(m_{\mathrm{in}}+1)}$, $\|B_t\|_2^2 \leq \frac{1}{\alpha}$, $\|w_t\|_2^2 \leq \alpha c_2$ *for constants* $c_1, c_2 \in \mathbb{R}_+$. *Then,*

$$\|U_t\|_2 \leq \alpha\left(c_2\frac{m_{\mathrm{in}}+1}{m_{\mathrm{in}}} + \frac{c_1\left(c_2+\bar{\sigma}^2\right)}{2m_{\mathrm{in}}(m_{\mathrm{in}}+1)}\right).$$

**Proof.** By definition of $U_t$,

$$\|U_t\|_2 \leq \underbrace{\|W_t\|_2}_{(A)} + \underbrace{\frac{\alpha^2}{m_{\mathrm{in}}}\delta_t\|D_t^\top D_t\|_2}_{(B)}.$$

Term (A) is bounded by Lemma 5. For the term (B), bounding $\delta_t$ by conditions on $B_t$ and $w_t$,

$$\delta_t = \|B_t w_t\|_2^2 + \bar{\sigma}^2 \leq c_2 + \bar{\sigma}^2,$$

one has the following bound

$$\frac{\alpha^2}{m_{\mathrm{in}}}\delta_t\|D_t^\top D_t\|_2 \leq \frac{\alpha c_1\left(c_2+\bar{\sigma}^2\right)}{2m_{\mathrm{in}}\left(m_{\mathrm{in}}+1\right)}.$$

Combining the two bounds yields the result,

$$\|U_t\|_2 \leq \alpha\left(c_2\frac{m_{\mathrm{in}}+1}{m_{\mathrm{in}}} + \frac{c_1\left(c_2+\bar{\sigma}^2\right)}{2m_{\mathrm{in}}(m_{\mathrm{in}}+1)}\right).$$

**Lemma 5.** *Assume that $\|B_t\|_2^2 \le \frac{1}{\alpha}$ and $\|w_t\|_2^2 \le \alpha c$ for a constant $c \in \mathbb{R}_+$. Then,*

$$\|W_t\|_2 \le \alpha c \frac{m_{\text{in}} + 1}{m_{\text{in}}}.$$

**Proof.** By using $0 \preceq B_t^\top B_t \preceq \frac{1}{\alpha} \mathbf{I}_{k'}$,

$$\|(\mathbf{I}_{k'} - \alpha B_t^\top B_t) w_t w_t^\top (\mathbf{I}_{k'} - \alpha B_t^\top B_t)\|_2 = \|(\mathbf{I}_{k'} - \alpha B_t^\top B_t) w_t\|_2^2 \le \|w_t\|_2^2 \le \alpha c,$$

$$\|B_t^\top B_t w_t w_t^\top B_t^\top B_t\|_2 = \|B_t^\top B_t w_t\|_2^2 \le \frac{1}{\alpha^2} \|w_t\|_2^2 \le \frac{c}{\alpha},$$

and the result follows by

$$\|W_t\|_2 \le \|(\mathbf{I}_{k'} - \alpha B_t^\top B_t) w_t w_t^\top (\mathbf{I}_{k'} - \alpha B_t^\top B_t)\|_2 + \frac{\alpha^2}{m_{\text{in}}} \|B_t^\top B_t w_t w_t^\top B_t^\top B_t\|_2 \le \alpha c \frac{m_{\text{in}} + 1}{m_{\text{in}}}.$$

## C.2 BOUNDS ON ITERATES AND MONOTONICITY

The recursion for $\Lambda_t$ given in Equation (22) has the following main term:

$$(\mathbf{I}_k + \alpha\beta R_t(\Lambda_t)) \Lambda_t (\mathbf{I}_k + \alpha\beta R_t(\Lambda_t))^\top.$$

Lemma 6 bounds $\Lambda_{t+1}$ from above by this term, i.e., terms involving $U_t$ are negative. On the other hand, Lemma 7 bounds $\Lambda_{t+1}$ from below with the expression

$$(\mathbf{I}_k + \alpha\beta R_t(\Lambda_t) - \alpha\beta\gamma_t \mathbf{I}_k) \Lambda_t (\mathbf{I}_k + \alpha\beta R_t(\Lambda_t) - \alpha\beta\gamma_t \mathbf{I}_k)^\top, \tag{26}$$

where $\gamma_t \in \mathbb{R}_+$ is a scalar such that $\|U_t\|_2 \le \alpha\gamma_t$. Lastly, Lemma 9 shows that updates of the form of Equation (26) enjoy a monotonicity property which allows the control of $\Lambda_t$ over time from above and below by constructing sequences of matrices, as described in Appendix C.3.

**Lemma 6.** *Suppose that $\|U_t\|_2 \le \frac{1}{\beta}$. Then,*

$$\Lambda_{t+1} \preceq (\mathbf{I}_k + \alpha\beta R_t(\Lambda_t)) \Lambda_t (\mathbf{I}_k + \alpha\beta R_t(\Lambda_t))^\top.$$

**Proof.** As $\|U_t\|_2 \le \frac{1}{\beta}$,

$$C_t U_t C_t^\top - \beta C_t U_t^2 C_t^\top = C_t (U_t - \beta U_t^2) C_t^\top \succeq 0.$$

Using Appendix C.2,

$$\Lambda_{t+1} = (\mathbf{I}_k + \alpha\beta R_t(\Lambda_t)) \left(\Lambda_t - \beta C_t U_t C_t^\top\right) (\mathbf{I}_k + \alpha\beta R_t(\Lambda_t))^\top - \beta C_t U_t C_t^\top + \beta^2 C_t U_t^2 C_t^\top$$

$$\preceq (\mathbf{I}_k + \alpha\beta R_t(\Lambda_t)) \left(\Lambda_t - \beta C_t U_t C_t^\top\right) (\mathbf{I}_k + \alpha\beta R_t(\Lambda_t))^\top$$

$$\preceq (\mathbf{I}_k + \alpha\beta R_t(\Lambda_t)) \Lambda_t (\mathbf{I}_k + \alpha\beta R_t(\Lambda_t))^\top.$$

**Lemma 7.** *Let $\gamma_t$ be a scalar such that $\|U_t\|_2 \le \alpha\gamma_t \le \frac{1}{2\beta}$. Then,*

$$(\mathbf{I}_k + \alpha\beta R_t(\Lambda_t) - \alpha\beta\gamma_t \mathbf{I}_k) \Lambda_t (\mathbf{I}_k + \alpha\beta R_t(\Lambda_t) - \alpha\beta\gamma_t \mathbf{I}_k)^\top \preceq \Lambda_{t+1}.$$

**Proof.** By using $\|U_t\|_2 \le \alpha\gamma_t$,

$$\alpha\gamma_t \Lambda_t - C_t U_t C_t^\top = C_t (\alpha\gamma_t \mathbf{I}_k - U_t) C_t^\top \succeq 0.$$

Moreover, as

$$x \mapsto x - \beta x^2,$$

is an increasing function in $[0, \frac{1}{2\beta}]$, the maximal eigenvalue of

$$U_t - \beta U_t^2$$

is $s - \beta s^2 \le \alpha\gamma_t - \alpha^2\beta\gamma_t^2$ where $s$ is the maximal eigenvalue $U_t$. Hence,

$$\left(\alpha\gamma_t - \alpha^2\beta\gamma_t^2\right) \mathbf{I}_{k'} - \left(U_t - \beta U_t^2\right) \succeq 0.$$

Therefore, the following expression is positive semi-definite,

$$2\mathrm{Sym}\left((\mathbf{I}_k + \alpha\beta R_t(\Lambda_t))\left(\alpha\gamma_t\Lambda_t - C_tU_tC_t^\top\right)\right) - \beta(\alpha^2\gamma_t^2\Lambda_t - C_tU_t^2C_t^\top)$$
$$= (\mathbf{I}_k + \alpha\beta R_t(\Lambda_t))\left(\alpha\gamma_t\Lambda_t - C_tU_tC_t^\top\right)(\mathbf{I}_k + \alpha\beta R_t(\Lambda_t))^\top$$
$$+ \left(\left(\alpha\gamma_t\Lambda_t - C_tU_tC_t^\top\right) - \beta\left(\alpha^2\gamma_t^2\Lambda_t - C_tU_t^2C_t^\top\right)\right)$$
$$\succeq C_t\left[\left(\alpha\gamma_t - \alpha^2\beta\gamma_t^2\right)\mathbf{I}_{k'} - \left(U_t - \beta U_t^2\right)\right]C_t^\top.$$

The result follows by

$$\Lambda_{t+1} = (\mathbf{I}_k + \alpha\beta R_t(\Lambda_t))\Lambda_t(\mathbf{I}_k + \alpha\beta R_t(\Lambda_t))^\top - 2\beta\mathrm{Sym}\left((\mathbf{I}_k + \alpha\beta R_t(\Lambda_t))C_tU_tC_t^\top\right) - \beta^2 C_tU_t^2C_t^\top$$
$$\succeq (\mathbf{I}_k + \alpha\beta R_t(\Lambda_t))\Lambda_t(\mathbf{I}_k + \alpha\beta R_t(\Lambda_t))^\top - 2\alpha\beta\gamma_t\mathrm{Sym}\left((\mathbf{I}_k + \alpha\beta R_t(\Lambda_t))\Lambda_t\right) - \alpha^2\beta^2\gamma_t^2\Lambda_t$$
$$= (\mathbf{I}_k + \alpha\beta R_t(\Lambda_t) - \alpha\beta\gamma_t\mathbf{I}_k)\Lambda_t(\mathbf{I}_k + \alpha\beta R_t(\Lambda_t) - \alpha\beta\gamma_t\mathbf{I}_k)^\top.$$

**Lemma 8.** *Let $C_t = \Psi_t S_t \Gamma_t^\top$ be the (thin) SVD decomposition of $C_t$ and let $\gamma_t$ be a scalar such that $\|\Gamma_t^\top U_t \Gamma_t\|_2 \le \alpha\gamma_t \le \frac{1}{2\beta}$. Then,*

$$(\mathbf{I}_k + \alpha\beta R_t(\Lambda_t) - \alpha\beta\gamma_t\mathbf{I}_k)\Lambda_t(\mathbf{I}_k + \alpha\beta R_t(\Lambda_t) - \alpha\beta\gamma_t\mathbf{I}_k)^\top \preceq \Lambda_{t+1}.$$

**Proof.** It is sufficient to observe that

$$\alpha\gamma_t\Lambda_t - C_tU_tC_t^\top = \Psi_t S_t\left(\alpha\gamma_t\mathbf{I}_{k'} - \Gamma_t^\top U_t\Gamma_t\right)S_t\Psi_t^\top \succeq 0,$$

and use the same argument as in the proof of Lemma 7.

**Lemma 9.** *For non-negative scalars $\tau, \gamma$, let $f(\cdot; \tau, \gamma) : \mathrm{Sym}_k(\mathbb{R}) \to \mathrm{Sym}_k(\mathbb{R})$ be defined as follows,*

$$f(\Lambda; \tau, \gamma) \coloneqq (\mathbf{I}_k + \alpha\beta R(\Lambda, \tau) - \alpha\beta\gamma\mathbf{I}_k)\Lambda(\mathbf{I}_k + \alpha\beta R(\Lambda, \tau) - \alpha\beta\gamma\mathbf{I}_k)^\top.$$

*Then, $f(\cdot; \tau, \gamma)$ preserves the partial order between any $\Lambda, \Lambda'$ that commutes with each other and $\Sigma_\star$, i.e.,*

$$\frac{1}{\alpha}\frac{m_{\mathrm{in}}}{m_{\mathrm{in}} + 1}\mathbf{I}_k \succeq \Lambda \succeq \Lambda' \succeq 0 \implies f(\Lambda; \tau, \gamma) \succeq f(\Lambda'; \tau, \gamma),$$

*when the following condition holds,*

$$1 - \alpha\beta\gamma \ge 5\alpha\beta(\|\Sigma_\star\|_2 + \frac{\bar\sigma^2 + \tau}{m_{\mathrm{in}} + 1}).$$

**Proof.** The result follows if and only if

$$(1 - \alpha\beta\gamma)^2(\Lambda - \Lambda') \succeq \alpha\beta(1 - \alpha\beta\gamma)\underbrace{[R(\Lambda', \tau)\Lambda' - R(\Lambda, \tau)\Lambda]}_{(A)}$$
$$+ \alpha\beta(1 - \alpha\beta\gamma)\underbrace{[\Lambda'R(\Lambda', \tau) - \Lambda R(\Lambda, \tau)]}_{(B)}$$
$$+ \alpha^2\beta^2\underbrace{[R(\Lambda', \tau)\Lambda'R(\Lambda', \tau) - R(\Lambda, \tau)\Lambda R(\Lambda, \tau)]}_{(C)}. \tag{27}$$

By Lemma 16,

$$\Lambda^2 - \Lambda'^2 = \frac{1}{2}(\Lambda - \Lambda')(\Lambda + \Lambda') + \frac{1}{2}(\Lambda + \Lambda')(\Lambda - \Lambda')$$
$$\preceq \|\Lambda + \Lambda'\|_2(\Lambda - \Lambda') \preceq 2\|\Lambda_\star\|_2(\Lambda - \Lambda').$$

Bounding term (A) by using commutativity of $\Lambda, \Lambda'$ with $\Sigma_\star$ and $\Lambda, \Lambda' \preceq \frac{1}{\alpha}\frac{m_{\mathrm{in}}}{m_{\mathrm{in}}+1}$,

$$R(\Lambda', \tau)\Lambda' - R(\Lambda, \tau)\Lambda = \frac{\alpha(m_{\mathrm{in}} + 1)}{m_{\mathrm{in}}}\Lambda\left[\Sigma_\star + \frac{\bar\sigma^2 + \tau}{m_{\mathrm{in}} + 1}\mathbf{I}_k\right]\Lambda$$
$$- \frac{\alpha(m_{\mathrm{in}} + 1)}{m_{\mathrm{in}}}\Lambda'\left[\Sigma_\star + \frac{\bar\sigma^2 + \tau}{m_{\mathrm{in}} + 1}\mathbf{I}_k\right]\Lambda' - \Sigma_\star(\Lambda - \Lambda')$$
$$\preceq \frac{\alpha(m_{\mathrm{in}} + 1)}{m_{\mathrm{in}}}\left[\|\Sigma_\star\|_2 + \frac{\bar\sigma^2 + \tau}{m_{\mathrm{in}} + 1}\right](\Lambda^2 - \Lambda'^2).$$

The term (B) is equal to the term (A) and thus bounded by the same expression. By Lemma 17,

$$\Lambda^3 - \Lambda'^3 \succeq 0.$$

Bounding term (C), using the commutativity of $\Lambda, \Lambda'$ with $\Sigma_\star$ and $\Lambda, \Lambda' \preceq \frac{1}{\alpha} \frac{m_{\mathrm{in}}}{m_{\mathrm{in}}+1}$,

$$R(\Lambda', \tau)\Lambda' R(\Lambda', \tau) - R(\Lambda, \tau)\Lambda R(\Lambda, \tau) = 2 \frac{\alpha(m_{\mathrm{in}}+1)}{m_{\mathrm{in}}} \left[ \Sigma_\star^2 + \frac{\bar{\sigma}^2 + \tau}{m_{\mathrm{in}}+1} \Sigma_\star \right] (\Lambda^2 - \Lambda'^2)$$

$$- \Sigma_\star(\Lambda - \Lambda')\Sigma_\star - \left( \frac{\alpha(m_{\mathrm{in}}+1)}{m_{\mathrm{in}}} \right)^2 \left[ \Sigma_\star + \frac{\bar{\sigma}^2 + \tau}{m_{\mathrm{in}}+1} \mathbf{I}_k \right]^2 (\Lambda^3 - \Lambda'^3)$$

$$\preceq 4 \|\Sigma_\star\|_2 \left[ \|\Sigma_\star\|_2 + \frac{\bar{\sigma}^2 + \tau}{m_{\mathrm{in}}+1} \right] (\Lambda - \Lambda').$$

Therefore, Equation (27) is satisfied if

$$(1 - \alpha\beta\gamma)^2 \geq 4\alpha\beta(1 - \alpha\beta\gamma) \left[ \|\Sigma_\star\|_2 + \frac{\bar{\sigma}^2 + \tau}{m_{\mathrm{in}}+1} \right] + 4\alpha^2\beta^2 \|\Sigma_\star\|_2 \left[ \|\Sigma_\star\|_2 + \frac{\bar{\sigma}^2 + \tau}{m_{\mathrm{in}}+1} \right],$$

which holds by the given condition.

**Remark 1.** *Let $\tau, \gamma$ be scalars such that $0 < \tau$ and $0 < \gamma < \lambda_{\min}(\Sigma_\star)$. Define $\Lambda_\star(\tau, \gamma)$ as follows,*

$$\Lambda_\star(\tau, \gamma) := \frac{1}{\alpha} \frac{m_{\mathrm{in}}}{m_{\mathrm{in}}+1} \left( \mathbf{I}_k - \left( \frac{\bar{\sigma}^2 + \tau}{m_{\mathrm{in}}+1} + \gamma \right) \left( \Sigma_\star + \frac{\bar{\sigma}^2 + \tau}{m_{\mathrm{in}}+1} \mathbf{I}_k \right)^{-1} \right). \tag{28}$$

*$(\Lambda_\star, \tau, \gamma)$ is a fixed point of the function $f$ as*

$$R\left(\Lambda_\star(\tau, \gamma)\right) = \gamma \mathbf{I}_k.$$

**Corollary 2.** *Let $\Lambda$ be a symmetric p.s.d. matrix which commutes with $\Sigma_\star$ and satisfy*

$$\Lambda \preceq \Lambda_\star(\tau, \gamma),$$

*for some scalars $0 < \tau$ and $0 < \gamma < \lambda_{\min}(\Sigma_\star)$. Then, assuming that conditions of Lemma 9 are satisfied,*

$$\Lambda \preceq f(\Lambda; \tau, \gamma) \preceq \Lambda_\star(\tau, \gamma).$$

**Proof.** For the left-hand side, note that

$$R(\Lambda, \tau, \gamma) \succeq \gamma \mathbf{I}_k \iff \Lambda \preceq \Lambda_\star(\tau, \gamma).$$

Hence, by the given assumption and commutativity,

$$\Lambda \preceq \left( \mathbf{I}_k + \alpha\beta R(\Lambda, \tau) - \alpha\beta\gamma \mathbf{I}_k \right) \Lambda \left( \mathbf{I}_k + \alpha\beta R(\Lambda, \tau) - \alpha\beta\gamma \mathbf{I}_k \right)^\top = f(\Lambda; \tau, \gamma).$$

For the right-hand side, note that by Lemma 9

$$f(\Lambda; \tau, \gamma) \preceq f(\Lambda_\star(\tau, \gamma); \tau, \gamma) = \Lambda_\star(\tau, \gamma).$$

**Lemma 10.** *Let $\tau_t$ and $\gamma_t$ be non-negative, non-increasing scalar sequences such that $\gamma_0 < \lambda_{\min}(\Sigma_\star)$, and $\Lambda$ be a symmetric p.s.d. matrix that commutes with $\Sigma_\star$ such that*

$$\Lambda \preceq \Lambda_\star(\tau_0, \gamma_0),$$

*where $\Lambda_\star(\tau, \gamma)$ is defined in Equation (28). Furthermore, suppose that $\alpha$ and $\beta$ satisfy*

$$\frac{1}{\alpha\beta} \geq \left( s_\star + \frac{\bar{\sigma}^2 + \tau_0}{m_{\mathrm{in}}+1} \right).$$

*Then, the sequence of matrices that are defined recursively as*

$$\Lambda^{(0)} := \Lambda, \quad \Lambda^{(t+1)} := f(\Lambda^{(t)}; \tau_t, \gamma_t),$$

*satisfy*

$$\lim_{t \to \infty} \Lambda^{(t)} = \Lambda_\star\left( \lim_{t \to \infty} \tau_t, \lim_{t \to \infty} \gamma_t \right).$$

**Proof.** By the monotone convergence theorem, $\tau_t$ and $\gamma_t$ are convergent. Let $\tau_\infty$ and $\gamma_\infty$ denote the limits, i.e.,

$$\tau_\infty := \lim_{t \to \infty} \tau_t, \quad \gamma_\infty := \liminf_{t \to \infty} \gamma_t.$$

As $\Lambda^{(0)}$ and $\Sigma_\star$ are commuting normal matrices, they are simultaneously diagonalizable, i.e., there exists an orthogonal matrix $Q \in \mathbb{R}^{k \times k}$ and diagonal matrices with positive entries $D^{(0)}, D_\star$ such that

$$\Lambda^{(0)} = Q D^{(0)} Q^\top, \quad \Sigma_\star = Q D_\star Q^\top.$$

Then, applying $f$ to any matrix of from $\Lambda = QDQ^\top$, where $D$ is a diagonal matrix with positive entries, yields

$$f(\Lambda; \tau, \gamma) = Q \left( \mathbf{I}_k + \alpha\beta D_\star - \alpha^2 \beta \frac{m_{\text{in}} + 1}{m_{\text{in}}} D \left( D_\star + \frac{\bar{\sigma}^2 + \tau}{m_{\text{in}} + 1} \mathbf{I}_k \right) - \alpha\beta\gamma \right)^2 D Q^\top.$$

Observe that $f$ operates entry-wise on diagonal elements of $D$, i.e., for any diagonal element $s$ of $D$, the output in the corresponding entry of $f$ is given by the following map $g(\cdot, s_\star, \tau, \gamma) : \mathbb{R} \to \mathbb{R}$,

$$g(s; s_\star, \tau, \gamma) := \left( 1 + \alpha\beta s_\star - \alpha^2\beta \frac{m_{\text{in}} + 1}{m_{\text{in}}} s(s_\star + \frac{\bar{\sigma}^2 + \tau}{m_{\text{in}} + 1}) - \alpha\beta\gamma \right)^2 s,$$

where $s_\star$ is the corresponding diagonal entry of $D_\star$. Hence, Lemma 10 holds if

$$\lim_{t \to \infty} s_t = s_\infty(\tau_\infty, \gamma_\infty),$$

where $s_t$ is defined recursively from an initial value $s_0$ for any $t \geq 1$ as follows,

$$s_{t+1} := g(s_t; s_\star, \tau_t, \gamma_t),$$

and $s_\infty(\tau, \gamma)$ is defined as

$$s_\infty(\tau, \gamma) := \frac{1}{\alpha} \frac{m_{\text{in}}}{m_{\text{in}} + 1} \left( 1 - \left( \gamma + \frac{\bar{\sigma}^2 + \tau}{m_{\text{in}} + 1} \right) \left( s_\star + \frac{\bar{\sigma}^2 + \tau}{m_{\text{in}} + 1} \right)^{-1} \right).$$

Observe that

$$s_\infty(\tau, \gamma) \left( s_\star + \frac{\bar{\sigma}^2 + \tau}{m_{\text{in}} + 1} \right) = \frac{1}{\alpha} \frac{m_{\text{in}}}{m_{\text{in}} + 1} (s_\star - \gamma),$$

and

$$g(s_t; s_\star, \tau_t, \gamma_t) = \left( 1 + \alpha\beta \left( s_\infty(\tau, \gamma) - s_t \right) \left( s_\star + \frac{\bar{\sigma}^2 + \tau}{m_{\text{in}} + 1} \right)^{-1} \right) s_t.$$

Hence,

$$s_\infty(\tau_t, \gamma_t) - s_{t+1} = (s_\infty(\tau_t, \gamma_t) - s_t) \left( 1 - \alpha\beta \left( s_\star + \frac{\bar{\sigma}^2 + \tau}{m_{\text{in}} + 1} \right)^{-1} \right),$$

and in each iteration $s_t$ takes a step towards $s_\infty(\tau_t, \gamma_t)$. By assumptions $s_0 \leq s_\infty(\tau_0, \gamma_0)$ and as

$$\frac{1}{\alpha\beta} \geq \left( s_\star + \frac{\bar{\sigma}^2 + \tau_t}{m_{\text{in}} + 1} \right),$$

for all $t$, $s_{t+1}$ never overshoots $s_\infty(\tau_t, \gamma_t)$, i.e.,

$$s_t \leq s_{t+1} \leq s_\infty(\tau_t, \gamma_t) \leq s_\infty(\tau_{t+1}, \gamma_{t+1}).$$

Therefore, $s_t$ is an increasing sequence bounded above by $s_\infty(\tau_\infty, \gamma_\infty)$ and by invoking the monotone convergence theorem, $s_t$ is convergent. Assume that $s_t$ convergences to a $s'_\infty < s_\infty(\tau_\infty, \gamma_\infty)$. Then, there exist a $t_\epsilon$ such that $s_\infty(\tau_{t_\epsilon}, \gamma_{t_\epsilon}) > s'_\infty + \epsilon$. By analyzing the sequence,

$$s'_{t_\epsilon} = s_{t_\epsilon}, \quad s'_{t_\epsilon + s} = g(s'_{t_\epsilon + s - 1}, s_\star, \tau_{t_\epsilon}, \gamma_{t_\epsilon}),$$

it is easy to show that

$$s_{t_\epsilon + s} \geq s'_{t_\epsilon + s}, \quad \text{and} \quad \lim_{s \to \infty} s'_{t_\epsilon + s} = s_\infty(\tau_{t_\epsilon}, \gamma_{t_\epsilon}) > s'_\infty,$$

which leads to a contradiction. Hence, $\lim_{t \to \infty} s_t = s_\infty(\tau_\infty, \gamma_\infty)$.

**Remark 2.** *Assume the setup of Lemma 10 and that the sequences $\tau_t$ and $\gamma_t$ converge to 0. Then, as $t \to \infty$, $\Lambda_t$ convergences to $\Lambda_\star$,*

$$\lim_{t \to \infty} \Lambda_t = \Lambda_\star.$$

## C.3 SEQUENCE OF BOUNDS

Lemma 11 constructs a sequence of matrices $\Lambda_t^U$ that upper bounds iterates of $\Lambda_t$. The idea is to use the monotonicity property described in Lemma 9, together with the upper bound in Lemma 6, to control $\Lambda_t$ from above. Lemma 10 with Remark 2 then allow to conclude $\lim_{t\to\infty} \Lambda_t^U = \Lambda_\star$. For this purpose, Lemma 11 assume a sufficiently small initialization that leads to a dynamics where $\|B_t\|_2 \leq \alpha^{-1/2}$ and $\|w_t\|_2, \|D_t\|_2$ are monotonically decreasing.

In a similar spirit, Lemma 12 construct a sequence of lower bound matrices $\Lambda_t^L$ given that it is possible to select two scalar sequences $\tau_t$ and $\gamma_t$. At each step, the lower bounds $\Lambda_t^L$ takes a step towards $\Lambda_\star(\tau_t, \gamma_t)$ described by Remark 1. For ensuring that $\Lambda_t$ does not decay, the sequences $\tau_t$ and $\gamma_t$ are chosen to be non-increasing, which results in increasing $\Lambda_\star(\tau_t, \gamma_t)$ and $\Lambda_t^L$. In the limit $t \to \infty$, $\Lambda_t^L$ convergences to the fixed-point $\Lambda_\star(\lim_{t\to\infty} \tau_t, \lim_{t\to\infty} \gamma_t)$, which serves as the asymptotic lower bound. Finally, Corollary 3 shows that it is possible to construct these sequences with the limit 0 under some conditions.

**Lemma 11.** *Assume that $B_0$ and $w_0$ are initialized such that*

$$\|B_0\|_2^2 \preceq \frac{c_1}{\alpha} \frac{1}{m_{\text{in}} + 1}, \quad \|w_0\|_2^2 \leq \alpha c_2,$$

*for constants $0 < c_1 < 1, 0 < c_2$ and $\alpha, \beta$ satisfy the following conditions:*

1. $\dfrac{1}{\alpha\beta} \geq \max\left( c_2 \dfrac{m_{\text{in}} + 2}{m_{\text{in}}} + \dfrac{1}{m_{\text{in}}} \left( (m_{\text{in}} + 1)\|\Sigma_\star\|_2 + \bar{\sigma}^2 \right), \dfrac{2\bar{\sigma}^2}{m_{\text{in}}} \right)$,

2. $\dfrac{1}{\alpha\beta} \geq 2 \left( c_2 \dfrac{m_{\text{in}} + 1}{m_{\text{in}}} + \dfrac{c_1 \left( c_2 + \bar{\sigma}^2 \right)}{2m_{\text{in}}(m_{\text{in}} + 1)} \right)$,

3. $\dfrac{1}{\alpha\beta} \geq 5(\|\Sigma_\star\|_2 + \dfrac{\bar{\sigma}^2}{m_{\text{in}} + 1})$,

4. $\beta \leq \alpha$.

*The series $\Lambda_t^U$ defined recursively as*

$$\Lambda_0^U := \|\Lambda_0\|_2 \mathbf{I}_k,$$
$$\Lambda_{t+1}^U := \|(\mathbf{I}_k + \alpha\beta R(\Lambda_t^U))\Lambda_t^U(\mathbf{I}_k + \alpha\beta R(\Lambda_t^U))\|_2 \mathbf{I}_k,$$

*upper bounds the iterates $\Lambda_t$, i.e., for all $t$, $\Lambda_t^U \succeq \Lambda_t$. Moreover, $\Lambda_\star \succeq \Lambda_t^U$ for all $t$.*

**Proof.** The result follows by induction. It is easy to check that the given assumptions satisfy the conditions of Lemmas 2 and 9 for all time steps. Assume that for time $t$, the following assumptions hold.

1. $\|D_s D_s^\top\|_2$ is a non-increasing sequence for $s \leq t$.

2. $\|w_s\|_2$ is a non-increasing sequence for $s \leq t$.

3. $\Lambda_s \preceq \Lambda_s^U \preceq \Lambda_\star$ for all $s \leq t$.

Then, for time $t + 1$, the following conditions holds:

1. By using $\Lambda_t \preceq \Lambda_\star \preceq \frac{1}{\alpha} \frac{m_{\text{in}}}{(m_{\text{in}}+1)}$ and $D_t D_t^\top \preceq B_0 B_0^\top \preceq \frac{c_1}{\alpha} \frac{1}{m_{\text{in}}+1}$,

$$B_t B_t^\top \preceq \frac{1}{\alpha} \frac{m_{\text{in}} + c_1}{m_{\text{in}} + 1}, \quad \|B_t\|_2 \leq \frac{1}{\alpha}.$$

Therefore, by Lemma 1, and Lemma 2,

$$\|w_{t+1}\|_2 \leq \|w_t\|_2, \quad \|D_{t+1} D_{t+1}^\top\|_2 \leq \|D_t D_t^\top\|_2.$$

2. By applying Lemma 4,

$$\|U_t\|_2 \leq \alpha \left( c_2 \frac{m_{\text{in}} + 1}{m_{\text{in}}} + \frac{c_1 \left( c_2 + \bar{\sigma}^2 \right)}{2m_{\text{in}}(m_{\text{in}} + 1)} \right) \leq \frac{1}{2\beta}.$$

Therefore, by Lemma 6,

$$\Lambda_{t+1} \preceq (\mathbf{I}_k + \alpha\beta R_t)\Lambda_t(\mathbf{I}_k + \alpha\beta R_t)^\top.$$

3. By applying Lemma 9 with $\Lambda := \Lambda_t^U$ and $\Lambda' := \Lambda_t$,

$$(\mathbf{I}_k + \alpha\beta R_t)\Lambda_t(\mathbf{I}_k + \alpha\beta R_t)^\top = f(\Lambda_t; 0, 0) \preceq f(\Lambda_t^U; 0, 0) \preceq \Lambda_{t+1}^U.$$

4. By applying Lemma 9 with $\Lambda := \Lambda_\star$ and $\Lambda' := \Lambda_t^U$,

$$f(\Lambda_t^U; 0, 0) \preceq f(\Lambda_\star; 0, 0) = \Lambda_\star.$$

Therefore, $\Lambda_{t+1}^U \preceq \|\Lambda_\star\|_2 \mathbf{I}_k = \Lambda_\star$.

5. Combining all the results,

$$\Lambda_{t+1} \preceq \Lambda_{t+1}^U \preceq \Lambda_\star.$$

**Lemma 12.** *Let $\tau_t$ and $\gamma_t$ be non-increasing scalar sequences such that*

$$\|B_t w_t\|_2^2 \leq \tau_t, \quad \|U_t\|_2 \leq \alpha\gamma_t \leq \frac{1}{2\beta},$$

*and $\tau_0 \leq c_2, \gamma_0 < \lambda_{\min}(\Sigma_\star)$. Assume that all the assumptions of Lemma 11 hold with constants $c_1$ and $c_2$. and $\alpha, \beta$ satisfy the following extra conditions*

$$\frac{1}{\alpha\beta} \geq 5(\|\Sigma_\star\|_2 + \frac{c_2 + \bar{\sigma}^2}{m_{\mathrm{in}} + 1}) + \lambda_{\min}(\Sigma_\star).$$

*Then, the series $\Lambda_t^L$ defined as follows*

$$\Lambda_0^L = \min\left(\lambda_{\min}(\Lambda_0), \lambda_{\min}(\Lambda_\star(\tau_0, \gamma_0))\right)\mathbf{I}_k,$$
$$\Lambda_{t+1}^L = \lambda_{\min}\left((\mathbf{I}_k + \alpha\beta R(\Lambda_t^L, \tau_t) - \alpha\beta\gamma_t\mathbf{I}_k)\Lambda_t^L(\mathbf{I}_k + \alpha\beta R(\Lambda_t^L, \tau_t) - \alpha\beta\gamma_t\mathbf{I}_k)^\top\right)\mathbf{I}_k,$$

*lower bounds the iterates $\Lambda_t$, i.e., for all $t$, $\Lambda_t^L \preceq \Lambda_t$. Moreover, $\Lambda_t^L \preceq \Lambda_{t+1}^L$ for all $t$.*

**Proof.** The result follows by induction. It is easy to check that given assumptions satisfy the conditions of Lemmas 2 and 9 for all time steps. Suppose that for all time $s \leq t$,

$$\Lambda_s^L \preceq \Lambda_s \preceq \Lambda_\star, \quad \Lambda_s^L \preceq \Lambda_\star(\tau_t, \gamma_t).$$

Then, for time $t + 1$, the following conditions hold:

1. By Lemma 7,

$$\Lambda_{t+1} \succeq (\mathbf{I}_k + \alpha\beta R_t(\Lambda_t) - \alpha\beta\gamma_t\mathbf{I}_k)\Lambda_t(\mathbf{I}_k + \alpha\beta R_t(\Lambda_t) - \alpha\beta\gamma_t\mathbf{I}_k)^\top.$$

2. By Lemma 9,

$$(\mathbf{I}_k + \alpha\beta R_t(\Lambda_t) - \alpha\beta\gamma_t\mathbf{I}_k)\Lambda_t(\mathbf{I}_k + \alpha\beta R_t(\Lambda_t) - \alpha\beta\gamma_t\mathbf{I}_k)^\top$$
$$\succeq (\mathbf{I}_k + \alpha\beta R_t(\Lambda_t^L) - \alpha\beta\gamma_t\mathbf{I}_k)\Lambda_t^L(\mathbf{I}_k + \alpha\beta R_t(\Lambda_t^L) - \alpha\beta\gamma_t\mathbf{I}_k)^\top.$$

3. Using commutativity of $\Sigma_\star$ and $\Lambda_t^L$,

$$(\mathbf{I}_k + \alpha\beta R_t(\Lambda_t^L) - \alpha\beta\gamma_t\mathbf{I}_k)\Lambda_t^L(\mathbf{I}_k + \alpha\beta R_t(\Lambda_t^L) - \alpha\beta\gamma_t\mathbf{I}_k)^\top$$
$$\succeq (\mathbf{I}_k + \alpha\beta R(\Lambda_t^L, \tau_t) - \alpha\beta\gamma_t\mathbf{I}_k)\Lambda_t^L(\mathbf{I}_k + \alpha\beta R(\Lambda_t^L, \tau_t) - \alpha\beta\gamma_t\mathbf{I}_k)^\top$$
$$\succeq \Lambda_{t+1}^L.$$

4. By Corollary 2,

$$\Lambda_t^L \preceq \Lambda_{t+1}^L \preceq \Lambda_\star(\tau_t, \gamma_t).$$

As $\tau_{t+1} \leq \tau_t$ and $\gamma_{t+1} \leq \gamma_t$,

$$\Lambda_{t+1}^L \preceq \Lambda_\star(\tau_t, \gamma_t) \preceq \Lambda_\star(\tau_{t+1}, \gamma_{t+1}).$$

5. Combining all the results,

$$\Lambda_{t+1}^L \preceq \Lambda_{t+1}, \quad \Lambda_{t+1}^L \preceq \Lambda_\star(\tau_t, \gamma_t) \preceq \Lambda_\star(\tau_{t+1}, \gamma_{t+1}).$$

**Remark 3.** *The condition on $\gamma_t$ in Lemma 12 can be relaxed by the condition used in Lemma 8.*

**Corollary 3.** *Assume that Lemma 12 holds with constants $c_1$, $c_2$, and constant sequences*

$$\tau := c_2, \quad \gamma := \left( c_2 \frac{m_{\mathrm{in}} + 1}{m_{\mathrm{in}}} + \frac{c_1 \left( c_2 + \bar{\sigma}^2 \right)}{2 m_{\mathrm{in}} (m_{\mathrm{in}} + 1)} \right) < \lambda_{\min}(\Sigma_\star).$$

*Furthermore, suppose $\alpha, \beta$ satisfy the following extra properties,*

$$\frac{1}{\alpha^2} \geq 4 \|\Sigma_\star\|_2,$$

$$\frac{\left( 1 - \frac{\beta}{4\alpha} \right)}{\alpha \beta} \geq \frac{\bar{\sigma}^2}{m_{\mathrm{in}} + 1} + c_2 \frac{m_{\mathrm{in}} + 2}{m_{\mathrm{in}} + 1} + \frac{m_{\mathrm{in}}}{(m_{\mathrm{in}} + 1)^2}.$$

*Let $C_t = \Psi_t S_t \Gamma_t^\top$ be the (thin) SVD decomposition of $C_t$. Then, there exist non-increasing scalar sequences $\tau_t$ and $\gamma_t$ such that*

$$\|B_t w_t\|_2^2 \leq \tau_t \leq c_2, \quad \|\Gamma_t U_t \Gamma_t^\top\|_2 \leq \alpha \gamma_t \leq \frac{1}{2\beta},$$

*with the limit*

$$\lim_{t \to \infty} \tau_t = 0, \quad \lim_{t \to \infty} \gamma_t = 0.$$

**Proof.** All the assumptions of Corollary 1 are satisfied with constant $c := c_2$. Hence,

$$\lim_{t \to \infty} \|D_t\|_2 = 0, \quad \lim_{t \to \infty} \|B_t w_t\|_2 = 0. \tag{29}$$

Moreover, the sequence $\|B_t w_t\|_2$ is upper bounded above,

$$\|B_t w_t\|_2 \leq \|B_t\|_2 \|w_t\|_2 \leq c_2.$$

Take any sequence $0 \leq \tau_t' \leq c_2$ that monotonically decays to 0. Set $\tau_0 = \tau_0'$ and $s_t = 0$. Recursively define $\tau_t$ as follows: for each $t > 0$, find the smallest $s_t$ such that

$$\|B_s w_s\|_2 \leq \tau_t',$$

for all $s \geq s_t$. Then, set $\tau_{s_t} = \tau_t'$ and for all $s_{t-1} \leq s < s_t$, set $\tau_s = \tau_{t-1}'$. It is easy to check that this procedure yields a non-increasing scalar sequence $\tau_t$ with the desired limit.

By Lemma 12 with $\gamma_t := \gamma$, $\Lambda_t$ is non-decaying, and its lowest eigenvalue is bounded from below. Using the limits in Equation (29),

$$\lim_{t \to \infty} C_t U_t C_t^\top = 0,$$

which implies that $\lim_{t \to \infty} \|\Gamma_t U_t \Gamma_t^\top\|_2 = 0$. A similar argument yields a non-increasing scalar sequence $\gamma_t$ with the desired limit.

### C.4 PROOF OF THEOREM 2

By Lemma 11, $\Lambda_t \preceq \Lambda_t^U \preceq \Lambda_\star$ and $\|D_{t+1}\|_2 \leq \|D_t\|_2$ for all $t$. Using the initialization condition,

$$\|B_t\|_2^2 = \|C_t\|_2^2 + \|D_t\|_2^2 \leq \|\Lambda_t\|_2 + \|D_0\|_2^2 \leq \|\Lambda_\star\|_2 + \|B_0\|_2^2 \leq \frac{1}{\alpha}.$$

Now, the conditions of Corollary 1 are satisfied with $c := c_2$. By Corollary 1,

$$\lim_{t \to \infty} D_t = 0, \quad \lim_{t \to \infty} B_t w_t = 0.$$

Moreover, by Corollary 3, there exist non-increasing sequences $\tau_t$ and $\gamma_t$ that are decaying. By Lemma 12 with these sequences yield $\Lambda_t^L \preceq \Lambda_t$, for all $t$. Finally, by Lemma 10,

$$\lim_{t \to \infty} \Lambda_t^L \to \Lambda_\star \quad \text{and} \quad \lim_{t \to \infty} \Lambda_t^U \to \Lambda_\star,$$

which concludes Theorem 2.

## D   PROOF OF PROPOSITION 1

Proposition 2 below gives a more complete version of Proposition 1, stating an upper bound holding with probability at least $1 - \delta$ for any $\delta > 0$.

**Proposition 2.** *Let* $\hat{B}, w_{\text{test}}$ *satisfy Equations* (9) *and* (11) *for a new task defined by Equation* (10). *For any* $\delta > 0$ *with probability at least* $1 - \delta$,

$$\|\hat{B}w_{\text{test}} - B_\star w_\star\|_2 = \mathcal{O}\left( \frac{1 + \bar{\sigma}^2/\lambda_{\min}(\Sigma_\star)}{m_{\text{in}}} \|w_\star\| + \max\left( \frac{\sqrt{k} + \sqrt{\log(\frac{4}{\delta})}}{\sqrt{m_{\text{test}}}}, \frac{k + \log(\frac{4}{\delta})}{m_{\text{test}}} \right) \|w_\star\| \right.$$

$$\left. + \sigma \sqrt{\frac{k}{m_{\text{test}}}} \left(1 + \sqrt{\frac{\log(\frac{4}{\delta})}{k}}\right)\left(1 + \sqrt{\frac{\log(\frac{4}{\delta})}{m_{\text{test}}}}\right) \right),$$

*where we recall* $\bar{\sigma}^2 = \text{Tr}(\Sigma_\star) + \sigma^2$.

Using Equation (11), it comes

$$\hat{B}w_{\text{test}} - B_\star w_\star = \left(\alpha \hat{B}\hat{B}^\top \Sigma_{\text{test}} B_\star - B_\star\right) w_\star + \frac{\alpha}{m_{\text{test}}} \hat{B}\hat{B}^\top X^\top z$$

$$= \underbrace{B_\star \left(\alpha \Lambda_\star - \mathbf{I}_k\right) w_\star}_{(A)} + \underbrace{\alpha B_\star \Lambda_\star \left(B_\star^\top \Sigma_{\text{test}} B_\star - \mathbf{I}_k\right) w_\star}_{(B)} + \underbrace{\frac{\alpha}{m_{\text{test}}} B_\star \Lambda_\star B_\star^\top X^\top z}_{(C)}.$$

The rest of the proof aims at individually bounding the norms of the terms (A), (B) and (C). First note that by definition of $\Lambda_\star$,

$$\alpha \Lambda_\star - \mathbf{I}_k = -\frac{1}{m_{\text{in}} + 1} \mathbf{I}_k - \frac{m_{\text{in}}\bar{\sigma}^2}{(m_{\text{in}} + 1)^2} \left[\Sigma_\star + \frac{\bar{\sigma}^2}{m_{\text{in}} + 1} \mathbf{I}_k\right]^{-1}.$$

This directly implies that

$$\|\alpha \Lambda_\star - \mathbf{I}_k\|_2 = \frac{1}{m_{\text{in}} + 1} + \frac{m_{\text{in}}\bar{\sigma}^2}{(m_{\text{in}} + 1)^2} \cdot \frac{1}{\lambda_{\min}(\Sigma_\star) + \frac{\bar{\sigma}^2}{m_{\text{in}}+1}}$$

$$\leq \frac{1 + \frac{\bar{\sigma}^2}{\lambda_{\min}(\Sigma_\star) + \bar{\sigma}^2/m_{\text{in}}}}{m_{\text{in}} + 1}. \tag{30}$$

Moreover, the concentration inequalities of Lemmas 13 and 14 claim that with probability at least $1 - \delta$:

$$\|B_\star^\top \Sigma_{\text{test}} B_\star - \mathbf{I}_k\|_2 \leq 3\max\left( \frac{\sqrt{k} + \sqrt{2\log(\frac{4}{\delta})}}{\sqrt{m_{\text{test}}}}, \frac{\left(\sqrt{k} + \sqrt{2\log(\frac{4}{\delta})}\right)^2}{m_{\text{test}}} \right),$$

$$\|B_\star^\top X^\top z\|_2 \leq 16\sigma\sqrt{m_{\text{test}}k}\,(1 + \sqrt{\frac{\log(\frac{4}{\delta})}{2k}})(1 + \sqrt{\frac{\log(\frac{4}{\delta})}{2m_{\text{test}}}}).$$

These two bounds along with Equation (30) then allow to bound the terms (A), (B) and (C) as follows

$$\|(A)\|_2 \leq \frac{1 + \frac{\bar{\sigma}^2}{\lambda_{\min}(\Sigma_\star) + \bar{\sigma}^2/m_{\text{in}}}}{m_{\text{in}} + 1} \|w_\star\|$$

$$\|(B)\|_2 \leq 3\left(1 - \frac{1 + \frac{\bar{\sigma}^2}{\|\Sigma_\star\|_2 + \bar{\sigma}^2/m_{\text{in}}}}{m_{\text{in}} + 1}\right) \max\left( \frac{\sqrt{k} + \sqrt{2\log(\frac{4}{\delta})}}{\sqrt{m_{\text{test}}}}, \frac{\left(\sqrt{k} + \sqrt{2\log(\frac{4}{\delta})}\right)^2}{m_{\text{test}}} \right) \|w_\star\|$$

$$\|(C)\|_2 \leq 16\left(1 - \frac{1 + \frac{\bar{\sigma}^2}{\|\Sigma_\star\|_2 + \bar{\sigma}^2/m_{\text{in}}}}{m_{\text{in}} + 1}\right) \sigma\sqrt{\frac{k}{m_{\text{test}}}}\,(1 + \sqrt{\frac{\log(\frac{4}{\delta})}{2k}})(1 + \sqrt{\frac{\log(\frac{4}{\delta})}{2m_{\text{test}}}}),$$

where we used in the two last bounds that $\alpha\|\Lambda_\star\|_2 \leq 1 - \frac{1+\frac{\overline{\sigma}^2}{\|\Sigma_\star\|_2+\overline{\sigma}^2/m_{\mathrm{in}}}}{m_{\mathrm{in}}+1}$. Summing these three bounds finally yields Proposition 2, and Proposition 1 with the particular choice $\delta = 4e^{-\frac{k}{2}}$. □

**Lemma 13.** *For any $\delta > 0$, with probability at least $1 - \frac{\delta}{2}$,*

$$\|B_\star^\top \Sigma_{\mathrm{test}} B_\star - \mathbf{I}_k\|_2 \leq 3\max\left(\frac{\sqrt{k}+\sqrt{2\log(\frac{4}{\delta})}}{\sqrt{m_{\mathrm{test}}}}, \frac{\left(\sqrt{k}+\sqrt{2\log(\frac{4}{\delta})}\right)^2}{m_{\mathrm{test}}}\right)$$

*and $\|B_\star^\top X^\top\|_2 \leq \sqrt{m_{\mathrm{test}}}\left(1 + \frac{\sqrt{k}+\sqrt{2\log(\frac{1}{4\delta})}}{\sqrt{m_{\mathrm{test}}}}\right)$.*

**Proof.** Note that $B_\star^\top X^\top$ is a matrix in $\mathbb{R}^{k \times m_{\mathrm{test}}}$ whose entries are independent standard Gaussian variables. From there, applying Corollary 5.35 and Lemma 5.36 from Vershynin (2012) with $t = \sqrt{2\log(\frac{1}{4\delta})}$ directly leads to Lemma 13.

**Lemma 14.**

$$\mathbb{P}\left(\|B_\star^\top X^\top z\|_2 \geq 16\sigma\sqrt{m_{\mathrm{test}}k}\,(1+\sqrt{\frac{\log(\frac{4}{\delta})}{2k}})(1+\sqrt{\frac{\log(\frac{4}{\delta})}{2m_{\mathrm{test}}}})\right) \leq \frac{\delta}{2}.$$

**Proof.** Let $A = B_\star^\top X^\top$ in this proof. Recall that $A$ has independent entries following a standard normal distribution. $A$ and $z$ are independent, which implies that $A\frac{z}{\|z\|} \sim \mathcal{N}(0, \mathbf{I}_k)$. Typical bounds on Gaussian variables then give (see e.g. Rigollet & Hütter, 2023, Theorem 1.19)

$$\mathbb{P}\left(\frac{\|Az\|}{\|z\|} \geq 4\sqrt{k}\,(1+\sqrt{\frac{\log(\frac{4}{\delta})}{2k}})\right) \leq \frac{\delta}{4}.$$

A similar bound holds on the $\sigma$ sub-Gaussian vector $z$, which is of dimension $m_{\mathrm{test}}$:

$$\mathbb{P}\left(\|z\| \geq 4\sqrt{m_{\mathrm{test}}}\,(1+\sqrt{\frac{\log(\frac{4}{\delta})}{2m_{\mathrm{test}}}})\right) \leq e^{-\frac{m_{\mathrm{test}}}{2}}.$$

Combining these two bounds then yields Lemma 14.

## E  TECHNICAL LEMMAS

**Lemma 15.** *Let $\Sigma = \frac{1}{n}X^\top X$ where $X \in \mathbb{R}^{n\times d}$ is such that each row is composed of i.i.d. samples $x \sim N(0, \mathbf{I}_d)$. For any unit vector $v$,*

$$\mathbb{E}[\Sigma vv^\top \Sigma] = \frac{1}{n}\mathbf{I}_d + \frac{n+1}{n}vv^\top.$$

**Proof.** Let $x, x' \sim N(0, \mathbf{I}_d)$. By expanding covariance $\Sigma$ and i.i.d. assumption,

$$\mathbb{E}[\Sigma vv^\top \Sigma] = \underbrace{\frac{1}{n}\mathbb{E}\left[\langle x, v\rangle^2 xx^\top\right]}_{(A)} + \underbrace{\frac{n-1}{n}\mathbb{E}\left[\langle x, v\rangle\langle x', v\rangle xx'^\top\right]}_{(B)}.$$

For the term (A),

$$\mathbb{E}\left[\langle x, v\rangle^2 xx^\top\right]_{jk} = \mathbb{E}\left[(\sum_{i=1}^d x_i v_i)^2 x_j x_k\right].$$

Any term with an odd-order power cancels out as the data is symmetric around the origin, and

$$\mathbb{E}\left[\langle x, v\rangle^2 xx^\top\right] = 2vv^\top + \mathbf{I}_d,$$

by the following computations,

$$\mathbb{E}\left[\langle x, v\rangle^2 xx^\top\right]_{jj} = v_j^2 \mathbb{E}[x_j^4] + \sum_{i\neq j} v_i^2 \mathbb{E}[x_i^2 x_j^2] = 3v_j^2 + \sum_{i\neq j} v_i^2 = 2v_j^2 + 1,$$

$$\mathbb{E}\left[\langle x, v\rangle^2 xx^\top\right]_{jk} = 2v_j v_k \mathbb{E}[x_j^2 x_k^2] = 2v_j v_k.$$

For the term (B), by i.i.d. assumption,

$$\mathbb{E}\left[\langle x, v\rangle\langle x', v\rangle xx'^\top\right] = \mathbb{E}[\langle x, v\rangle x]\mathbb{E}[\langle x, v\rangle x]^\top.$$

With a similar argument, it is easy to see

$$\mathbb{E}[\langle x, v\rangle x]_i = \mathbb{E}[x_i^2 v_i] = v_i, \quad \text{and} \quad \mathbb{E}[\langle x, v\rangle x] = v.$$

Combining the two terms yields Lemma 15.

**Lemma 16.** *Let $A$ and $B$ be positive semi-definite symmetric matrices of shape $k \times k$ and $AB = BA$. Then,*

$$AB \preceq \|A\|_2 B.$$

**Proof.** As $A$ and $B$ are normal matrices that commute, there exist an orthogonal $Q$ such that $A = Q\Lambda_A Q^\top$ and $B = Q\Lambda_B Q^\top$ where $\Lambda_A$ and $\Lambda_B$ are diagonal. Then,

$$AB = Q\Lambda_A \Lambda_B Q^\top \preceq \|A\|Q\Lambda_B Q^\top,$$

as for any vector $v \in \mathbb{R}^k$,

$$v^\top ABv = \sum_{i=1}^k (\Lambda_A)_{ii}(\Lambda_B)_{ii}(Qv_i)^2 \leq \|A\|_2 \sum_{i=1}^k (\Lambda_B)_{ii}(Qv_i)^2 = \|A\|_2 B.$$

**Lemma 17.** *Let $A$ and $B$ be positive semi-definite symmetric matrices of shape $k \times k$ such that $AB = BA$ and $A \preceq B$. Then, for any $k \in \mathbb{N}$,*

$$A^k \preceq B^k. \tag{31}$$

**Proof.** As $A$ and $B$ are normal matrices that commute, there exist an orthogonal $Q$ such that $A = Q\Lambda_A Q^\top$ and $B = Q\Lambda_B Q^\top$ where $\Lambda_A$ and $\Lambda_B$ are diagonal. Then,

$$B^k - A^k = Q(\Lambda_B^k - \Lambda_A^k)Q^\top \succeq 0,$$

as $B \succeq A$ implies $\Lambda_B \succeq \Lambda_A$.

## F    FIXED POINTS CHARACTERIZED BY THEOREM 1 ARE GLOBAL MINIMA

The ANIL loss with $m$ samples in the inner loop reads,

$$\mathcal{L}_{\text{ANIL}}(B, w; m) = \frac{1}{2}\mathbb{E}_{w_{\star,i}, X_i, y_i}\left[\left\|B\tilde{w}(w; X_i, y_i) - B_\star w_{\star,i}\right\|^2\right], \tag{32}$$

where is the updated head after a step of gradient descent, i.e.,

$$\tilde{w}(w; X_i, y_i) := \left(w - \frac{\alpha}{m}B^\top X_i^\top(X_i B w - y_i)\right). \tag{33}$$

Whenever the context is clear, we will write $\tilde{w}$ or $\tilde{w}(w)$ instead of $\tilde{w}(w; X_i, y_i)$ for brevity. Theorem 1 proves that minimizing objective in Equation (32) with FO-ANIL algorithm asymptotically convergences to a set of fixed points, under some conditions. In Proposition 3, we show that these points are global minima of the Equation (32).

**Proposition 3.** *Fix any $(\hat{B}, \hat{w})$ that satisfy the three limiting conditions of Theorem 1,*

$$B_{\star,\perp}^\top \hat{B} = 0,$$
$$\hat{B}\hat{w} = 0,$$
$$B_\star^\top \hat{B}\hat{B}^\top B_\star = \Lambda_\star.$$

*Then, $(\hat{B}, \hat{w})$ is the minimizer of the Equation (32), i.e.,*

$$(\hat{B}, \hat{w}) \in \underset{B,w}{\arg\min}\, \mathcal{L}_{\text{ANIL}}(B, w; m_{\text{in}}).$$

**Proof.** The strategy of proof is to iteratively show that modifying points to satisfy these three limits reduce the ANIL loss. Lemmas 18 to 20 demonstrates how to modify each point such that the resulting point obeys a particular limit and has better generalization.

For any $(B, w)$, define the following points,

$$(B_1, w_1) = \left(B - B_{\star,\perp}^\top B_{\star,\perp}^\top B, w\right),$$
$$(B_2, w_2) = \left(B_1, w_1 - B_1^\top\left(B_1 B_1^\top\right)^{-1}B_1 w_1\right).$$

Then, Lemmas 18 to 20 show that

$$\mathcal{L}_{\text{ANIL}}(B, w; m_{\text{in}}) \geq \mathcal{L}_{\text{ANIL}}(B_1, w_1; m_{\text{in}}) \geq \mathcal{L}_{\text{ANIL}}(B_2, w_2; m_{\text{in}}) \geq \mathcal{L}_{\text{ANIL}}(\hat{B}, \hat{w}; m_{\text{in}}).$$

Since $(B, w)$ is arbitrary,

$$(\hat{B}, \hat{w}) \in \underset{B,w}{\arg\min}\, \mathcal{L}_{\text{ANIL}}(B, w; m_{\text{in}}).$$

**Lemma 18.** *Consider any parameters $(B, w) \in \mathbb{R}^{d \times k'} \times \mathbb{R}^{k'}$. Let $B' = B - B_{\star,\perp} B_{\star,\perp}^\top B$. Then, for any $m > 0$, we have*
$$\mathcal{L}_{\text{ANIL}}(B, w; m) \geq \mathcal{L}_{\text{ANIL}}(B', w; m).$$

**Proof.** Decomposing the loss into two orthogonal terms yields the desired result,

$$\begin{aligned}
\mathcal{L}_{\text{ANIL}}(B, w; m) &= \frac{1}{2}\mathbb{E}_{w_{\star,i}, X_i, y_i}\left[\left\|B_\star^\top B\tilde{w} - w_{\star,i}\right\|^2\right] + \frac{1}{2}\mathbb{E}_{X_i, y_i}\left[\left\|B_{\star,\perp}^\top B\tilde{w}\right\|^2\right] \\
&\leq \frac{1}{2}\mathbb{E}_{w_{\star,i}, X_i, y_i}\left[\left\|B_\star^\top B\tilde{w} - w_{\star,i}\right\|^2\right] \\
&= \mathcal{L}_{\text{ANIL}}(B', w; m).
\end{aligned}$$

**Lemma 19.** *Consider any parameters $(B, w) \in \mathbb{R}^{d \times k'} \times \mathbb{R}^{k'}$ such that $B_{\star,\perp}^\top B = 0$. Let $w' = w - B^\top\left(BB^\top\right)^{-1}Bw$. Then, for any $m > 0$, we have*

$$\mathcal{L}_{\text{ANIL}}(B, w; m) \geq \mathcal{L}_{\text{ANIL}}(B, w'; m),$$

**Proof.** Expanding the square,

$$\mathcal{L}_{\text{ANIL}}(B, w; m) - \mathcal{L}_{\text{ANIL}}(B, w'; m) = \frac{1}{2} \mathbb{E}_{w_{\star,i}, X_i, y_i} \left[ \left\| B_{\star}^{\top} B \tilde{w}(w) - w_{\star,i} \right\|^2 - \left\| B_{\star}^{\top} B \tilde{w}(w') - w_{\star,i} \right\|^2 \right]$$

$$= \frac{1}{2} \underbrace{\mathbb{E}_{w_{\star,i}, X_i, y_i} \left[ \left\| B \tilde{w}(w) \right\|^2 - \left\| B \tilde{w}(w') \right\|^2 \right]}_{(A)} - \underbrace{\mathbb{E}_{w_{\star,i}, X_i, y_i} \left[ \langle B_{\star} w_{\star,i}, B \tilde{w}(w) - B \tilde{w}(w') \rangle \right]}_{(B)} .$$

First, expanding $\tilde{w}(w)$ and $\tilde{w}(w')$ by Equation (33),

$$B \tilde{w}(w) = \left( \mathbf{I}_d - \frac{\alpha}{m} B B^{\top} X_i^{\top} X_i \right) B w + \frac{\alpha}{m} B B^{\top} X_i^{\top} y_i, \quad B \tilde{w}(w') = \frac{\alpha}{m} B B^{\top} X_i^{\top} y_i.$$

For the first term,

$$(A) = \mathbb{E}_{X_i} \left[ \left\| \left( \mathbf{I}_d - \frac{\alpha}{m} B B^{\top} X_i^{\top} X_i \right) B w \right\|^2 \right] + \frac{2\alpha}{m} \mathbb{E}_{w_{\star,i}, X_i} \left[ \left\langle \left( \mathbf{I}_d - \frac{\alpha}{m} B B^{\top} X_i^{\top} X_i \right) B w, B B^{\top} X_i^{\top} X_i B_{\star} w_{\star} \right\rangle \right]$$

$$= \mathbb{E}_{X_i} \left[ \left\| \left( \mathbf{I}_d - \frac{\alpha}{m} B B^{\top} X_i^{\top} X_i \right) B w \right\|^2 \right] \geq 0,$$

where we have used that the tasks and the noise are centered around $0$. For the second term,

$$(B) = \left\langle \mathbb{E}_{w_{\star,i}} \left[ B_{\star} w_{\star,i} \right], \mathbb{E}_{X_i} \left[ \left( \mathbf{I}_d - \frac{\alpha}{m_{\text{in}}} B B^{\top} X_i^{\top} X_i \right) B w \right] \right\rangle = 0,$$

where we have again used that the tasks are centered around $0$. Putting two results together yields Lemma 19.

**Lemma 20.** *Consider any parameters* $(B, w) \in \mathbb{R}^{d \times k'} \times \mathbb{R}^{k'}$ *such that* $B_{\star,\perp}^{\top} B = 0, B w = 0$. *Let* $(B', w') \in \mathbb{R}^{d \times k'} \times \mathbb{R}^{k'}$ *such that* $B_{\star,\perp}^{\top} B' = 0, B' w' = 0$ *and* $B_{\star}^{\top} B' B'^{\top} B_{\star} = \Lambda_{\star}$. *Then, we have*

$$\mathcal{L}_{\text{ANIL}}(B, w; m_{\text{in}}) \geq \mathcal{L}_{\text{ANIL}}(B', w'; m_{\text{in}}).$$

**Proof.** Let $\Lambda := B_{\star}^{\top} B B^{\top} B_{\star}$ in this proof. Using $B_{\star,\perp}^{\top} B = 0$, we have

$$\mathcal{L}_{\text{ANIL}}(B, w; m_{\text{in}}) = \frac{1}{2} \mathbb{E}_{w_{\star,i}, X_i, y_i} \left[ \left\| B_{\star}^{\top} B \tilde{w} - w_{\star,i} \right\|^2 \right].$$

Plugging in the definition of $\tilde{w}$,

$$\mathcal{L}_{\text{ANIL}}(B, w; m_{\text{in}}) = \frac{\alpha^2}{2} \frac{1}{m_{\text{in}}^2} \underbrace{\mathbb{E}_{w_{\star,i}, X_i, y_i} \left[ \left\| B_{\star}^{\top} B B^{\top} X_i^{\top} y_i \right\|^2 \right]}_{(A)}$$

$$- \alpha \frac{1}{m_{\text{in}}} \underbrace{\mathbb{E}_{w_{\star,i}, X_i, y_i} \left[ \langle w_{\star,i}, B_{\star}^{\top} B B^{\top} X_i^{\top} y_i \rangle \right]}_{(B)} + \frac{1}{2} \text{tr} \left( \Sigma_{\star} \right).$$

Using that the label noise is centered,

$$(A) = \underbrace{\mathbb{E}_{w_{\star,i}, X_i} \left[ \left\| B_{\star}^{\top} B B^{\top} \Sigma_i B_{\star} w_{\star} \right\|^2 \right]}_{(C)} + \underbrace{\mathbb{E}_{X_i, z_i} \left[ \left\| B_{\star}^{\top} B B^{\top} X_i^{\top} z_i \right\|^2 \right]}_{(D)},$$

where $\Sigma_i := \frac{1}{m_{\text{in}}} X_i^{\top} X_i$. By the independence of $w_{\star,i}, X_i$ and Lemma 15,

$$(C) = \text{tr} \left( B_{\star}^{\top} B B^{\top} \mathbb{E}_{w_{\star,i}, X_i} \left[ \Sigma_i B_{\star} w_{\star} w_{\star}^{\top} B_{\star}^{\top} \Sigma_i \right] B B^{\top} B_{\star} \right)$$

$$= \text{tr} \left( B_{\star}^{\top} B B^{\top} \mathbb{E}_{X_i} \left[ \Sigma_i B_{\star} \Sigma_{\star} B_{\star}^{\top} \Sigma_i \right] B B^{\top} B_{\star} \right)$$

$$= \frac{m_{\text{in}} + 1}{m_{\text{in}}} \text{tr} \left( B_{\star}^{\top} B B^{\top} B_{\star} \Sigma_{\star} B_{\star}^{\top} B B^{\top} B_{\star} \right) + \frac{1}{m_{\text{in}}} \text{tr} \left( B_{\star}^{\top} B B^{\top} B B^{\top} B_{\star} \right)$$

$$= \frac{m_{\text{in}} + 1}{m_{\text{in}}} \text{tr} \left( \Lambda \Sigma_{\star} \Lambda \right) + \frac{1}{m_{\text{in}}} \text{tr} \left( \Sigma_{\star} \right) \text{tr} \left( \Lambda^2 \right).$$

For the term (D), we have

$$
\begin{aligned}
(D) &= \frac{1}{m_{\mathrm{in}}} \mathrm{tr}\left(B_\star^\top BB^\top \mathbb{E}_{X_i, z_i}\left[X_i^\top z_i z_i^\top X_i\right] BB^\top B_\star\right) \\
&= \sigma^2 \mathrm{tr}\left(B_\star^\top BB^\top \mathbb{E}_{X_i}\left[\Sigma_i\right] BB^\top B_\star\right) \\
&= \sigma^2 \mathrm{tr}\left(B_\star^\top BB^\top BB^\top B_\star\right) \\
&= \sigma^2 \mathrm{tr}\left(\Lambda^2\right).
\end{aligned}
$$

Lastly, for the term (B), we have

$$
\begin{aligned}
(B) &= \frac{1}{m_{\mathrm{in}}} \mathbb{E}_{w_{\star,i}, X_i}\left[\langle w_{\star,i}, B_\star^\top BB^\top X_i^\top X_i B_\star w_{\star,i}\rangle\right] \\
&= \mathbb{E}_{w_{\star,i}}\left[\langle w_{\star,i}, B_\star^\top BB^\top B_\star w_{\star,i}\rangle\right] \\
&= \mathrm{tr}\left(\Lambda \Sigma_\star\right).
\end{aligned}
$$

Putting everything together using $\Sigma_\star$ is scaled identity,

$$
\begin{aligned}
\mathcal{L}_{\mathrm{ANIL}}(B, w; m_{\mathrm{in}}) &= \frac{\alpha^2}{2m_{\mathrm{in}}}\left((m_{\mathrm{in}}+1)\,\mathrm{tr}\left(\Lambda \Sigma_\star \Lambda\right) + \mathrm{tr}\left(\Sigma_\star\right)\mathrm{tr}\left(\Lambda^2\right)\right) - \alpha \mathrm{tr}\left(\Lambda \Sigma_\star\right) + \frac{1}{2}\mathrm{tr}\left(\Sigma_\star\right) \\
&= \frac{\alpha^2}{2m_{\mathrm{in}}}\left((m_{\mathrm{in}}+1)\,\|\Sigma_\star\|_2 + \bar{\sigma}^2\right)\mathrm{tr}\left(\Lambda^2\right) - \alpha\|\Sigma_\star\|_2\mathrm{tr}(\Lambda) + \frac{1}{2}\mathrm{tr}\left(\Sigma_\star\right).
\end{aligned}
$$

Hence, the loss depends on $B$ only through $\Lambda := B_\star^\top BB^\top B_\star$ for all $(B, w)$ such that $B_{\star,\perp}^\top B = 0, Bw = 0$. Taking the derivative w.r.t. $\Lambda$ yields that $\Lambda$ is a minimizer if and only if

$$
\frac{\alpha}{m_{\mathrm{in}}}\left((m_{\mathrm{in}}+1)\,\|\Sigma_\star\|_2 + \bar{\sigma}^2\right)\Lambda - \lambda_{\max}\left(\Sigma_\star\right) I = 0.
$$

This quantity is minimized for $\Lambda_\star$ as

$$
\alpha \frac{m_{\mathrm{in}}+1}{m_{\mathrm{in}}}\Lambda_\star\left(\Sigma_\star + \frac{\bar{\sigma}^2}{m_{\mathrm{in}}+1}\Lambda_\star\right) = \Sigma_\star.
$$

## G  EXTENDING COLLINS ET AL. (2022) ANALYSIS TO THE MISSPECIFIED SETTING

We show that the dynamics for infinite samples in the misspecified setting $k < k' \leq d$ is reducible to a well-specified case studied in Collins et al. (2022). The idea is to show that the dynamics is restricted to a $k$-dimensional subspace via a time-independent bijection between misspecified and well-specified iterates.

In the infinite samples limit, $m_{\text{in}} = \infty, m_{\text{out}} = \infty$, the outer loop updates of Equation (3) simplify with Assumption 1 to

$$
\begin{aligned}
w_{t+1} &= w_t - \beta \Delta_t B_t^\top \left( B_t w_t - B_\star \mu_\star \right), \\
B_{t+1} &= B_t - \beta B_t \Delta_t w_t \left( \Delta_t w_t + \alpha B_t^\top B_\star \mu_\star \right)^\top \\
&\quad + \beta \left( \mathbf{I}_d - \alpha B_t B_t^\top \right) B_\star \left( \mu_\star \left( \Delta_t w_t \right)^\top + \alpha \Sigma_\star B_\star^\top B_t \right).
\end{aligned}
\tag{34}
$$

where $\mu_\star$ and $\Sigma_\star$ respectively are the empirical task mean and covariance, and $\Delta_t := \mathbf{I}_{k'} - \alpha B_t^\top B_t$. This leads to following updates on $C_t := B_\star^\top B_t$,

$$
\begin{aligned}
C_{t+1} &= \left( \mathbf{I}_k + \alpha \beta \left( \mathbf{I}_k - C_t C_t^\top \right) \Sigma_\star \right) C_t - \beta C_t \Delta_t w_t \left( \Delta_t w_t + \alpha C_t^\top \mu_\star \right)^\top \\
&\quad + \beta \left( \mathbf{I}_k - \alpha C_t C_t^\top \right) \mu_\star \left( \Delta_t w_t \right)^\top.
\end{aligned}
$$

A key observation of this recursion is that all the terms end with $C_t$ or $\Delta_t$. This observation is sufficient to deduce that $C_t$ is fixed in its row space.

Assume that $B_0$ is initialized such that

$$
\ker(C_0) \subseteq \ker(\Delta_0).
$$

This condition is always satisfiable by a choice of $B_0$ that guarantees $B_0^\top B_0 = \alpha \mathbf{I}_{k'}$, similarly to Collins et al. (2022). With this assumption, there is no dynamics in the kernel space of $C_0$. More precisely, we show that for all time $t$, $\ker(C_0) \subseteq \ker(C_t) \cap \ker(\Delta_t)$. Then, it is easy to conclude that $B_t$ has simplified rank-deficient dynamics.

Assume the following inductive hypothesis at time $t$,

$$
\ker(C_0) \subseteq \ker(C_t) \cap \ker(\Delta_t).
$$

For time step $t + 1$, we have for all $v \in \ker(C_t) \cap \ker(\Delta_t)$, $C_{t+1} v = 0$. As a result, the next step contains the kernel space of the previous step, i.e., $\ker(C_0) \subseteq \ker(C_t) \cap \ker(\Delta_t) \subseteq \ker(C_{t+1})$. Similarly, inspecting the expression for $\Delta_{t+1}$, we have for all $v \in \ker(C_t) \cap \ker(\Delta_t)$, $\Delta_{t+1} v = 0$ and $\ker(C_0) \subseteq \ker(C_t) \cap \ker(\Delta_t) \subseteq \ker(\Delta_{t+1})$. Therefore, the induction hypothesis at time step $t + 1$ holds.

Now, using that $\ker(C_t) = \mathrm{col}(C_t^\top)^\perp$, row spaces of $C_t$ are confined in the same $k$-dimensional subspace, $\mathrm{col}(C_t^\top) \supseteq \mathrm{col}(C_0^\top)$. Let $R \in \mathbb{R}^{k \times k'}$ and $R_\perp \in \mathbb{R}^{(k'-k) \times k'}$ be two orthogonal matrices that span $\mathrm{col}(C_0^\top)$ and $\mathrm{col}(C_0^\top)^\perp$, respectively. That is, $R$ and $R_\perp$ satisfy $R R^\top = \mathbf{I}_k$, $\mathrm{col}(R) = \mathrm{col}(C_0^\top)$ and $R_\perp R_\perp^\top = \mathbf{I}_{k'-k}$, $\mathrm{col}(R_\perp) = \mathrm{col}(C_0^\top)^\perp$. It is easy to show that updates to $B_t$ and $w_t$ are orthogonal to $\mathrm{col}(R_\perp)$, i.e.,

$$
B_t R_\perp^\top = B_0 R_\perp^\top, \quad \text{and} \quad R_\perp w_t = R_\perp w_0.
$$

With this result, we can prove that there is a $k$-dimensional parametrization of the misspecified dynamics. Let $\hat{w}_0 \in \mathbb{R}^k, \hat{B}_0 \in \mathbb{R}^{d \times k}$ defined as

$$
\hat{B}_0 := B_0 R^\top, \quad \hat{w}_0 := R w_0.
$$

Running FO-ANIL in the infinite samples limit, initialized with $\hat{B}_0$ and $\hat{w}_0$, mirrors the dynamics of the original misspecified iterations, i.e., $\hat{B}_t$ and $\hat{w}_t$ satisfy,

$$
\hat{B}_t = B_t R^\top, \quad \hat{w}_t = R w_t, \quad \hat{B}_t \hat{w}_t = B_t w_t - B_0 R_\perp^\top R_\perp w_0.
$$

This given bijection proves that iterates are fixed throughout training on the $k' - k$-dimensional subspace $\mathrm{col}(R_\perp)$. Hence, as argued in Section 4, the infinite samples dynamics do not capture unlearning behavior observed in Section 5. In contrast, the infinite tasks idealisation exhibits both learning and unlearning dynamics.

## H    CONVERGENCE RATE FOR UNLEARNING

In Proposition 4, we derive the rate $\|B_{\star,\perp}^\top B_t\|^2 = \mathcal{O}\big(\frac{m_{\text{in}}}{\alpha^2\beta\bar{\sigma}^2 t}\big)$.

**Proposition 4.** *Under the conditions of Theorem 2,*

$$\left\|B_{\star,\perp}^\top B_t\right\|_2^2 \leq \frac{1}{\alpha^2\beta\frac{\bar{\sigma}^2}{m_{\text{in}}}t + \frac{1}{\left\|B_{\star,\perp}^\top B_0\right\|_2^2}}, \tag{35}$$

*for any time $t \geq 0$.*

**Proof.**  Recall that Lemma 2 holds for all time steps by Theorem 2. That is, for all $t > 0$,

$$\left\|B_{\star,\perp}^\top B_{t+1}\right\|_2^2 \leq \left(1 - \kappa\left\|B_{\star,\perp}^\top B_t\right\|_2^2\right)\left\|B_{\star,\perp}^\top B_t\right\|_2^2, \tag{36}$$

where $\kappa := \frac{\alpha^2\beta}{m_{\text{in}}}\bar{\sigma}^2$ for brevity.  Now, assume the inductive hypothesis in Equation (35) holds for time $t$. Observe that the function $x \mapsto (1 - \kappa x)x$ is increasing on $[0, \frac{1}{2\kappa}]$ and

$$\|B_{\star,\perp}^\top B_t\|_2^2 \leq \|B_{\star,\perp}^\top B_0\|_2^2 \leq \frac{1}{\alpha}\frac{1}{m_{\text{in}}+1} \leq \frac{1}{2\kappa},$$

by the assumptions of Theorem 2. Then, by Equation (36) and monotonicity of $x \mapsto (1-\kappa x)x$,

$$\|B_{\star,\perp}^\top B_{t+1}\|_2^2 \leq \left(1 - \frac{\kappa}{\kappa t + \frac{1}{\|B_{\star,\perp}B_0\|_2^2}}\right)\frac{1}{\kappa t + \frac{1}{\|B_{\star,\perp}B_0\|_2^2}} = \frac{\kappa(t-1) + \frac{1}{\|B_{\star,\perp}^\top B_0\|_2^2}}{\left(\kappa t + \frac{1}{\|B_{\star,\perp}B_0\|_2^2}\right)^2}.$$

Using the inequality of arithmetic and geometric means,

$$\|B_{\star,\perp}^\top B_{t+1}\|_2^2 \leq \frac{\kappa(t-1) + \frac{1}{\|B_{\star,\perp}^\top B_0\|_2^2}}{\left(\kappa t + \frac{1}{\|B_{\star,\perp}B_0\|_2^2}\right)^2} \cdot \frac{\kappa(t+1) + \frac{1}{\|B_{\star,\perp}B_0\|_2^2}}{\kappa(t+1) + \frac{1}{\|B_{\star,\perp}B_0\|_2^2}}$$

$$\leq \frac{1}{\kappa(t+1) + \frac{1}{\|B_{\star,\perp}B_0\|_2^2}}.$$

Hence, the induction hypothesis at time step $t + 1$ holds.

# I  ADDITIONAL MATERIAL ON EXPERIMENTS

## I.1  EXPERIMENTAL DETAILS

In the experiments considered in Section 5, samples are split into two subsets with $m_{\text{in}} = 20$ and $m_{\text{out}} = 10$ for model-agnostic methods. The task parameters $w_{\star,i}$ are drawn i.i.d. from $\mathcal{N}(0, \Sigma_\star)$, where $\Sigma_\star = c\text{diag}(1, \ldots, k)$ and $c$ is a constant chosen so that $\|\Sigma_\star\|_F = \sqrt{k}$. Moreover, the features are drawn i.i.d. following a standard Gaussian distribution. All the curves are averaged over 10 training runs.

Model-agnostic methods are all trained using step sizes $\alpha = \beta = 0.025$. For the infinite tasks model, the iterates are computed using the close form formulas given by Equations (5) and (6) for $m_{\text{in}} = 20$. For the infinite samples model, it is computed using the closed form formula of Collins et al. (2022, Equation (3)) with $N = 5000$ tasks. The matrix $B_0$ is initialized randomly as an orthogonal matrix such that $B_0^\top B_0 = \frac{1}{4\alpha}\mathbf{I}_{k'}$. The vector $w_0$ is initialized uniformly at random on the $k'$-dimensional sphere with squared radius $0.01k'\alpha$.

For training Burer-Monteiro method, we initialize $B_0$ is initialized randomly as an orthogonal matrix such that $B_0^\top B_0 = \frac{1}{100}\mathbf{I}_{k'}$ and each column of $W$ is initialized uniformly at random on the $k'$-dimensional sphere with squared radius $0.01k'\alpha$. [2] Also, similarly to Tripuraneni et al. (2021), we add a $\frac{1}{8}\|B_t^\top B_t - W_t W_t^\top\|_F^2$ regularizing term to the training loss to ensure training stability. The matrices $B_t$ and $W_t$ are simultaneously trained with LBFGS using the default parameters of `scipy`.

For Table 1, we consider ridge regression for each learned representation. For example, if we learned the representation given by the matrix $\hat{B} \in \mathbb{R}^{d \times k'}$, the `Ridge` estimator is given by

$$\underset{w \in \mathbb{R}^{k'}}{\arg\min} \hat{\mathcal{L}}_{\text{test}}(\hat{B}w; X, y) + \lambda\|w\|_2^2.$$

The regularization parameter $\lambda$ is tuned for each method using a grid search over multiple values.

## I.2  GENERAL TASK DISTRIBUTIONS

In this section, we run similar experiments to Section 5, but with a more difficult task distribution and 3 training runs per method. In particular the task parameters are now generated as $w_{\star,i} \sim \mathcal{N}(\mu_\star, \Sigma_\star)$, where $\mu_\star$ is chosen uniformly at random on the $k$-sphere of radius $\sqrt{k}$. Also, $\Sigma_\star$ is chosen proportional to $\text{diag}(e^1, \ldots, e^k)$, so that its Frobenius-norm is $2\sqrt{k}$ and its condition number is $e^{k-1}$.

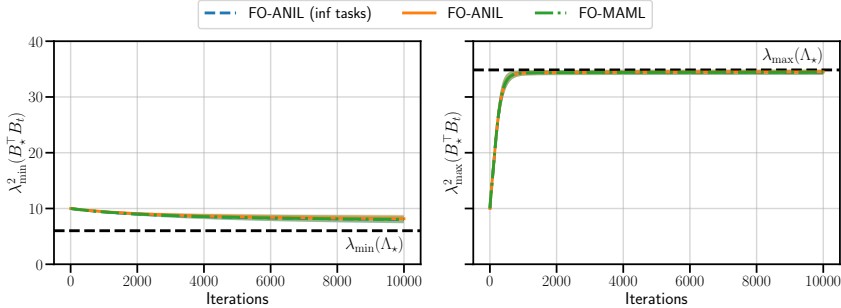

Figure 4:  Evolution of largest (*left*) and smallest (*right*) squared singular values of $B_\star^\top B_t$ during training. The shaded area represents the standard deviation observed over 3 runs.

Similarly to Section 5, Figures 4 and 5 show the evolution of the squared singular values on the good subspace and its orthogonal component during the training. Similarly to the well-behaved case of Section 5, model-agnostic methods seem to correctly learn the good subspace and unlearn its orthogonal complement, still at a very slow rate. The main difference is that the matrix towards which $B_\star^\top B_t B_t^\top B_\star$ converges does not exactly correspond to the $\Lambda_\star$ matrix defined in Theorem 1. We believe this is due to an additional term that should appear in the presence of a non-zero task mean. We yet do not fully understand what this term should be.

---

[2]We choose a small initialization regime for Burer-Monteiro to be in the good implicit bias regime. Note that Burer-Monteiro yields worse performance when using a larger initialization scale.

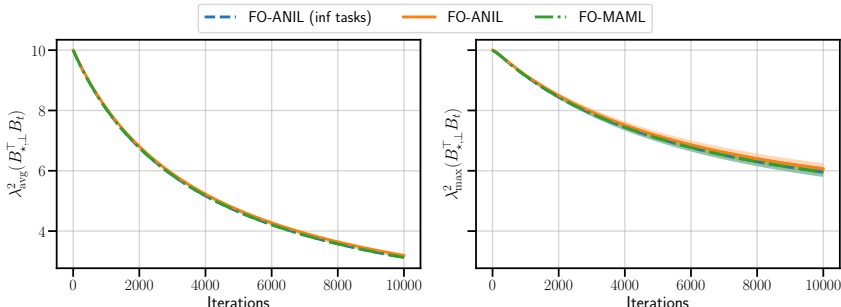

Figure 5: Evolution of average (*left*) and largest (*right*) squared singular value of $B_{\star,\perp}^{\top} B_t$ during training. The shaded area represents the standard deviation observed over 3 runs.

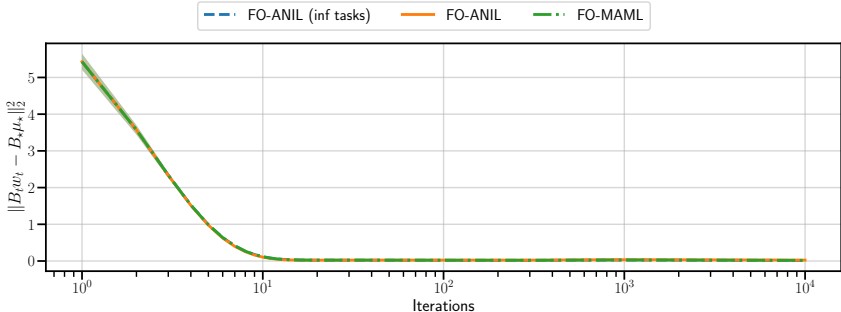

Figure 6: Evolution of $\|B_t w_t - B_\star \mu_\star\|_2^2$ during training. The shaded area represents the standard deviation observed over 3 runs.

Figure 6 on the other hand shows the evolution of $\|B_t w_t - B_\star \mu_\star\|$ while training. This value quickly decreases to 0. This decay implies that model-agnostic methods learn not only the low-dimensional space on which the task parameters lie, but also their mean value. It then chooses this mean value as the initial point, and consequentially, the task adaptation happens quickly at test time. Overall, the experiments in this section suggest that model-agnostic methods still learn a good representation when facing more general task distributions.

### I.3 NUMBER OF GRADIENT STEPS AT TEST TIME

This section studies what should be done at test time for the different methods. Figure 7 illustrates how the excess risk evolves when running gradient descent over the head parameters $w$, for the methods trained in Section 5. For all results, gradient descent is run with step size $0.01$, which is actually smaller than the $\alpha$ used while training FO-ANIL.

Keeping the step size equal to $\alpha$ leads to optimization complications when running gradient descent: the objective loss diverges, since the step size is chosen too large. This divergence is due to the fact that FO-ANIL chooses a large scale $B_t$ while training: this ensures a quick adaptation after a single gradient step but also leads to divergence of gradient descent after many steps.

The excess risk first decreases for all the methods while running gradient descent. However, after some critical threshold, it increases again for all methods except the Oracle. It is due to the fact that at some point in the task adaptation, the methods start overfitting the noise using components along the orthogonal complement of the ground-truth space. Even though the representation learned by FO-ANIL is nearly rank-deficient, it is still full rank. As can be seen in the difference between FO-ANIL and Oracle, this tiny difference between rank-deficient and full rank actually leads to a huge performance gap when running gradient descent until convergence.

Additionally, Figure 7 nicely illustrates how early stopping plays some regularizing role here. Overall, this suggests it is far from obvious how the methods should adapt at test time, despite having learned a good representation.

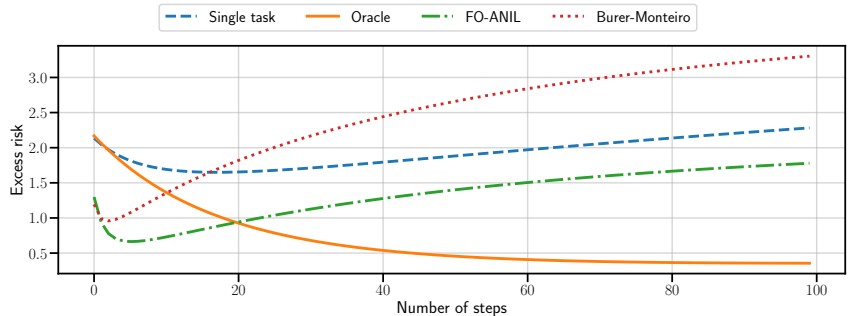

Figure 7: Evolution of the excess risk (evaluated on 1000 tasks with $m_{\text{test}} = 30$) with respect to the number of gradient descent steps processed, averaged over 10 training runs.

### I.4 IMPACT OF NOISE AND NUMBER OF SAMPLES IN INNER UPDATES

In this section, we run additional experiments to illustrate the impact of label noise and the number of samples on the decay of the orthogonal complement of the ground-truth subspace. The experimental setup is the same as Section 5 for FO-ANIL with finite tasks, except for the changes in the number of samples per task and the variance of label noise.

Figure 8 illustrates the decay of squared singular value of $B_{\star,\perp}^\top B_t$ during training. As predicted by Appendix H, the unlearning is fastest when $m_{\text{in}} = 10$ and slowest when $m_{\text{in}} = 30$. Figure 9 plots the decay with respect to different noise levels. The rate derived for the infinite tasks model suggests that the decay is faster for larger noise. However, experimental evidence with a finite number of tasks is more nuanced. The decay is indeed fastest for $\sigma^2 = 4$ and slowest for $\sigma^2 = 0$ on average. However, the decay of the largest singular value slows down for $\sigma^2 = 4$ in a second time, while the decay still goes on with $\sigma^2 = 0$, and the largest singular value eventually becomes smaller than in the $\sigma^2 = 4$ case. This observation might indicate the intricate dynamics of FO-ANIL with finite tasks.

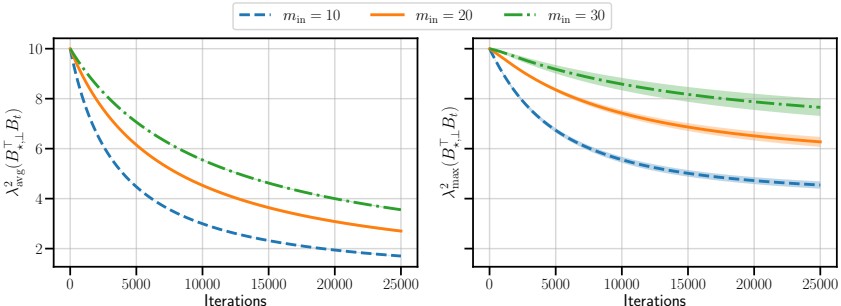

Figure 8: Evolution of average (*left*) and largest (*right*) squared singular value of $B_{\star,\perp}^\top B_t$ during training. The shaded area represents the standard deviation observed over 5 runs.

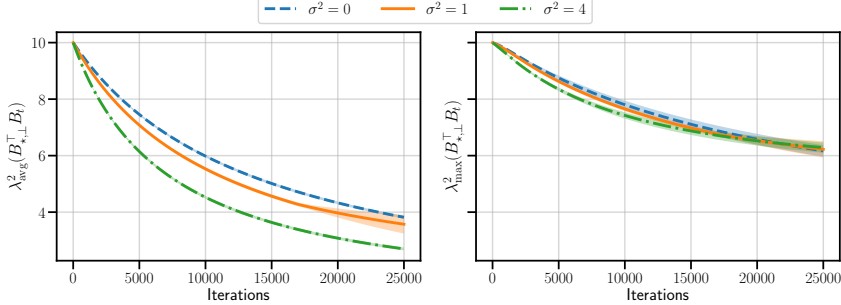

Figure 9: Evolution of average (*left*) and largest (*right*) squared singular value of $B_{\star,\perp}^\top B_t$ during training. The shaded area represents the standard deviation observed over 5 runs.

### I.5 SCALING LAWS IN PROPOSITION 1

In this subsection, we study the scaling laws predicted by the upper bound in Proposition 1. We compute excess risk and estimation errors and compare them with the predictions from Proposition 1. All errors are computed by sampling 1000 test-time tasks with 1000 test samples each.

In order to show that there is no dependency on $d$ after pretraining with FO-ANIL, we run experiments with varying $d = k'$ and $k$, in the same experimental setup as described in Appendix I.1. To mimic few-shot and high-sample regimes, we select $m_{\text{test}} = 20$ and $m_{\text{test}} = 1000$. Our results are shown in Figure 10. The excess risk does not scale with the ambient dimension $d$ but with the hidden low-rank dimension $k$.

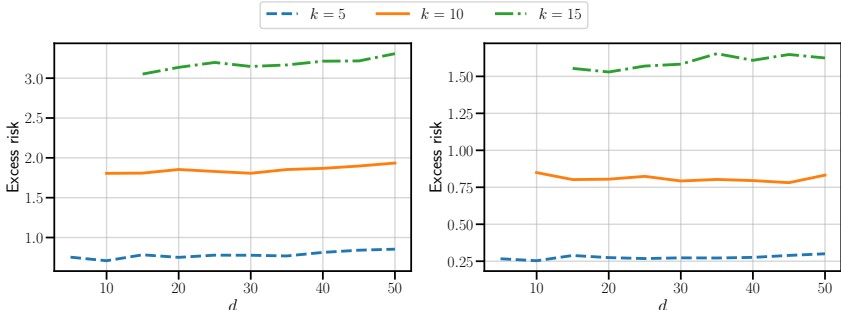

Figure 10: Excess risk for varying $d$ and $k$ for FO-ANIL with test samples $m_{\text{test}} = 10$ (*left*) and $m_{\text{test}} = 1000$ (*right*).

Next, we run a series of experiments to evaluate scaling laws predicted by Proposition 3. Similarly, we follow the experimental setting detailed in Appendix I.1 with an identity $\Sigma_\star$. In order to provide a clean comparison, $\Sigma_\star$ is scaled in test time such that $\mathbb{E}[\|w_\star\|] = 1$ for all $k$ values. This allows us to isolate the impact of $k, m_{\text{in}}$ and $m_{\text{test}}$ on the generalization error after adaptation. We also set $d = k' = 25$ and $N = 25000$ for the rest of this subsection.

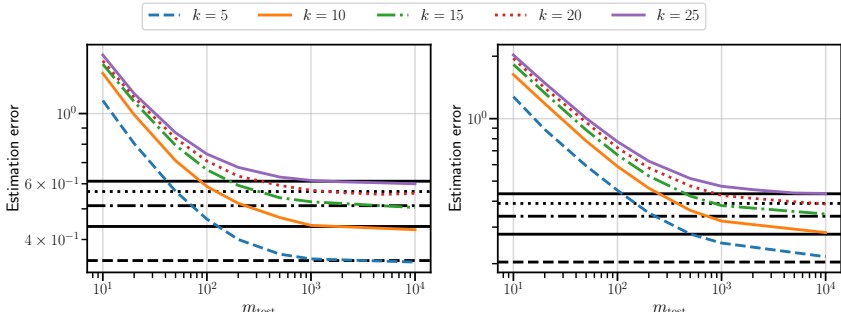

Figure 11: Estimation error of learned FO-ANIL representations for $k = 5, 10, 15$ and $m_{\text{in}} = 20$ (*left*) and $m_{\text{in}} = 40$ (*right*) with $\sigma^2 = 2$. Horizontal lines are bounds used in Proposition 1 for the first term that scales independently from $m_{\text{test}}$, i.e., , bounds for $m_{\text{test}} = \infty$.

Figure 11 shows the scaling of the loss with respect to $k$ and different choices of $m_{\text{test}}$ for $m_{\text{in}} = 20$ and $m_{\text{in}} = 40$, together with predictions made from Proposition 1. Black horizontal lines are bounds for the first term that does not scale with $m_{\text{test}}$, i.e., , bound to the generalization error when $m_{\text{test}} = \infty$. We observe that the bound used in Proposition 1 is tight. The dependency on $k$ is through the term $\overline{\sigma^2}/\sigma_{\min}(\Sigma_\star)$ which is equal to $\text{tr}(\Sigma)/\sigma_{\min}(\Sigma_\star) = k$ when $\sigma^2 = 0$.

In order to evaluate the other two terms in Proposition 1, we subtract the dashed black lines from $\|Bw_{\text{test}} - B_\star w_\star\|_2$ and plot it with respect to $\sqrt{k/m_{\text{test}}}$. Figure 12 shows that this excess estimation error is linear in $\sqrt{k/m_{\text{test}}}$. Black horizontal lines are $(1 + \sigma^2)\sqrt{k/m_{\text{test}}}$ that serves as upper bound to the two last terms in Proposition 1. Overall, our results indicate the scaling given by Proposition 1 is tight.

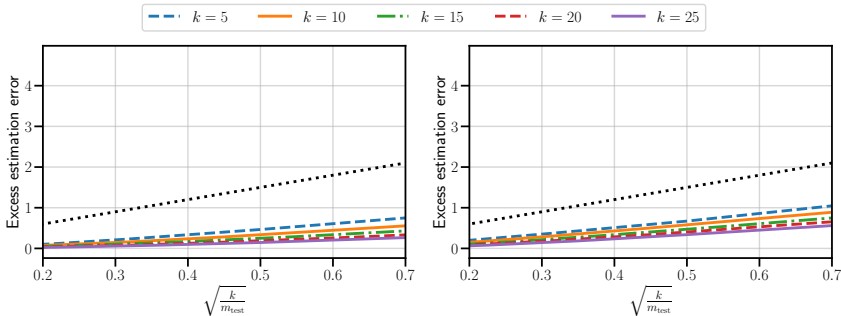

Figure 12: Excess estimation error of learned FO-ANIL representations for $k = 5, 10, 15, 20, 25$ and $m_{\text{in}} = 20$ (*left*) and $m_{\text{in}} = 40$ (*right*) with $\sigma^2 = 2$. Horizontal lines are bounds used in Proposition 1 for the last two terms that scale with $m_{\text{test}}$.

### I.6 IMPACT OF NONLINEARITY AND MULTIPLE LAYERS

We train two-layer and three-layer ReLU networks and study the scaling of test loss with the dimension $d$ and hidden dimension $k$. All the hidden layers have $d$ units. The experimental setting is the same as in Appendix I.1 except that tasks are processed in batches of size 100 out of a pool of 25000 for a faster training. In the case of two-layer ReLU networks, we set $\alpha = \beta = 0.025$, while for three-layer ReLU networks, we adjust the values to $\alpha = \beta = 0.01$.

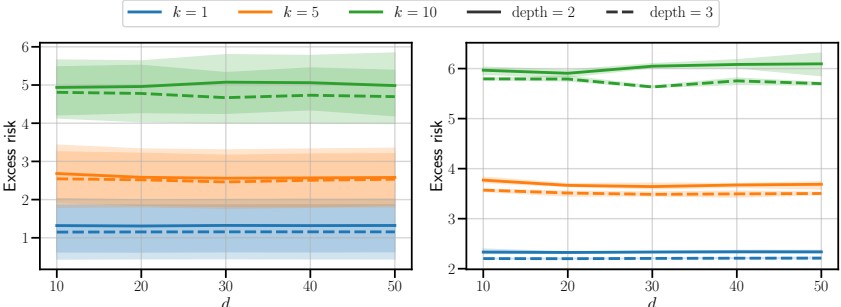

Figure 13: Excess risk for two-layer and three-layer ReLU networks pretrained with FO-ANIL and varying $d, k$ and $\sigma^2 = 0$ (*left*) and $\sigma^2 = 4$ (*right*). The shaded area represents the standard deviation observed over 3 runs.

Figure 13 shows that the excess risk does not scale with the ambient data dimensionality $d$ but with the hidden problem dimension $k$. This is evidence that suggests the adaptability of model-agnostic meta-learning pretraining extends to more general networks.

Next, we study representation learning in ReLU networks. Let $f(\cdot) : \mathbb{R}^d \to \mathbb{R}^d$ represent the network function that sends data to intermediate representations before the last layer. Then, we compute the best linear approximation of $f$ as follows: Let $X \in \mathbb{R}^{N \times d}$ be a matrix with each row is sampled from the $d$-dimensional isotropic Gaussian distribution. Solve the following minimization problem over all $B \in \mathbb{R}^{d \times d}$:

$$\underset{B \in \mathbb{R}^{d \times d}}{\arg\min} \|XB - f(X)\|_2^2,$$

where $f(X) \in \mathbb{R}^N$ is the output of the network applied to each row separately. Applying this to each time step $t$ with $N = 1000$, we obtain $B_t$ that approximates the ReLU network throughout its trajectory. Finally, we repeat the experiments on singular values to check learning in the good feature space and unlearning in the complement space with the sequence $B_t$.

The feature learning behavior in two-layer and three-layer ReLU networks, as illustrated in Figures 14 and 16, closely mirrors that of the linear case. Notably, both ANIL, MAML, and their first-order counterparts exhibit increasing singular values in the good feature space. We observe a swifter learning with second-order methods. Moreover, there is a difference in the scale of singular values between first-order and second-order approaches in two-layer networks. In the context of three-layer networks, ANIL and MAML exhibit distinct scales.

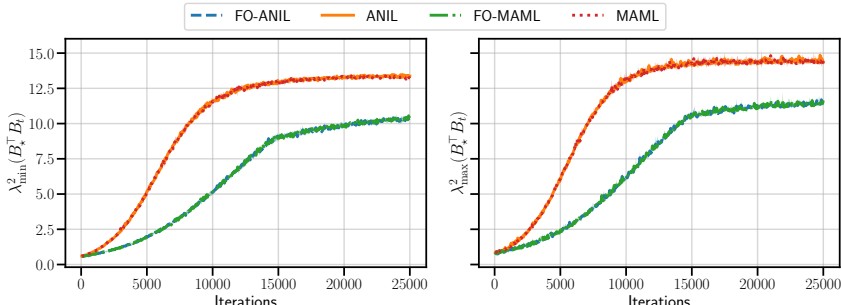

Figure 14: Evolution of average (*left*) and largest (*right*) squared singular value of $B_\star^\top B_t$ during FO-ANIL pretraining with two-layer ReLU networks. The shaded area represents the standard deviation observed over 3 runs.

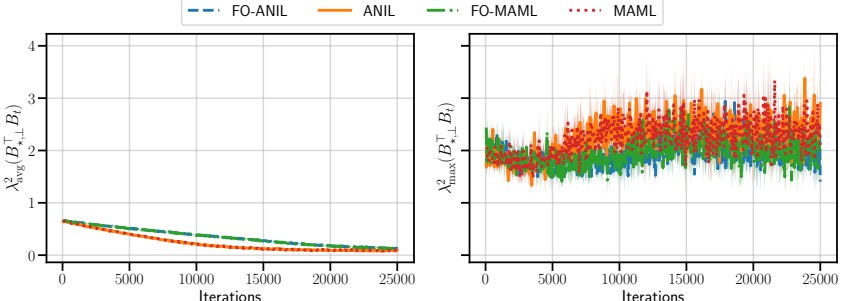

Figure 15: Evolution of average (*left*) and largest (*right*) squared singular value of $B_{\star,\perp}^\top B_t$ during FO-ANIL pretraining with two-layer ReLU networks. The shaded area represents the standard deviation observed over 3 runs.

In Figures 15 and 17, the dynamics in the complement feature space for two-layer and three-layer ReLU networks are depicted. While the average singular value exhibits a decaying trend, contrary to our experiments with two-layer linear networks, the maximal singular value does not show a similar decay. Note, however, that the scale of the initialization is smaller than in the linear case and singular values in all complement directions remain small when compared to good feature directions in the pretraining phase. This smaller initialization is due to nonlinearities in ReLU networks; the linear approximation yield smaller singular values than the weight matrices of the ReLU network which is initialized at the same scale as experiments with linear networks (for three-layer networks, their product is of the same scale). As elaborated further in Appendix A, this behavior in the complement space is also observable in two-layer linear networks under small initializations and is influenced by the finite number of tasks as opposed to the infinite tasks considered in our Theorem 1.

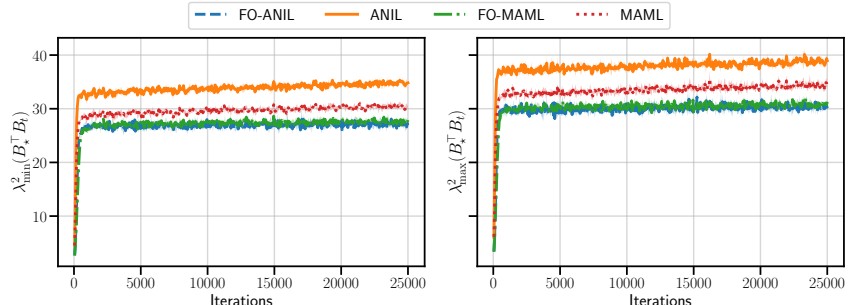

Figure 16: Evolution of average (*left*) and largest (*right*) squared singular value of $B_\star^\top B_t$ during FO-ANIL pretraining with three-layer ReLU networks. The shaded area represents the standard deviation observed over 3 runs.

Overall, experiments with two-layer and three-layer ReLU networks show that they learn the $k$-dimensional shared structure with a higher magnitude than the rest of the complement directions. This implies the learning of a shared task structure and good generalization under adaptation with

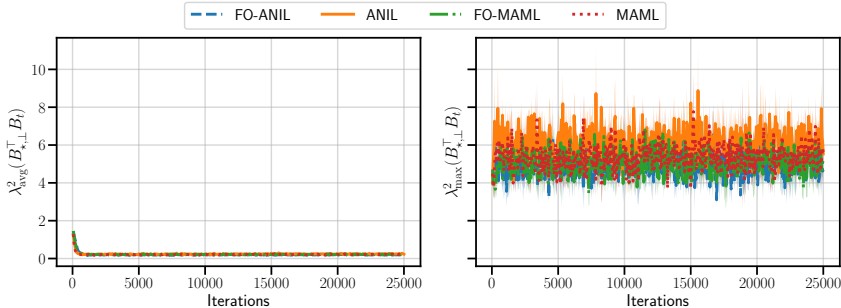

Figure 17: Evolution of average (*left*) and largest (*right*) squared singular value of $B_{\star,\perp}^{\top} B_t$ during FO-ANIL pretraining with three-layer ReLU networks. The shaded area represents the standard deviation observed over 3 runs.

few samples, and might indicate that Theorem 1 and Proposition 1 could be extended to nonlinear networks. The regularization effect of model-agnostic meta-learning on complement directions could be better seen with initializations that result in a linear map with high singular values in every direction. We leave detailed exploration of the unlearning process to future work.

