# OpenReview forum: "First-order ANIL provably learns representations despite overparametrisation"
_ICLR.cc/2024/Conference — ICLR 2024 poster_

### Official Review · Reviewer_pdse · 2023-10-23

**Soundness:** 4 excellent
**Presentation:** 3 good
**Contribution:** 3 good
**Rating:** 6
**Confidence:** 3

**Summary:**

This paper analyzes the representations learned by first-order ANIL (Almost-No-Inner-Loop), a representative gradient-based meta-learning algorithm. The analysis is performed in the context of two-layer linear networks in an overparameterized multi-task regression setting, where the width of the network is larger than the dimension of the ground-truth parameter subspace. The results generalize prior results from Collins et al. where it is assumed that the width of the network is well-specified and the number of samples in each task is sufficiently large.

**Strengths:**

-	The analysis is technically solid and in general well-presented.
-	The discussion and comparisons with prior results are clear.

**Weaknesses:**

-	As suggested by the paper, the main conceptual message, as well as the key distinction induced by the network overparameterization, is that meta-learning not only learns the desired subspace spanned by the columns of $B_*$, but also “unlearns” the orthogonal complement of this subspace. However, the analysis is based on the fact that the number of inner-loop training samples $m_\mathrm{in}$ is insufficient so the inner-loop gradient descent is prone to overfitting, which is then penalized by the outer-loop meta-training process. What if we use other techniques to mitigate overfitting such as weight decay as commonly used in practice?
-	In the high level, the theoretical results in this paper show that overparameterization/more task-specific data provably _hurts_ generalization in their simplified meta-learning setting. I am not sure if this is consistent with the growing empirical evidence that overparameterization/more data usually _helps_ generalization in practical deep learning. Does this mismatch suggest that the linear model considered in this paper is somewhat over-simplified?

**Questions:**

-	Will it change the learned representations if we add l2 weight decay regularization to the meta-training objective to avoid overfitting (with finite/infinite samples per task)?
-	Is there any empirical evidence suggesting that having more samples in each task can be harmful to gradient-based meta-learning approaches in practice, beyond your numerical experiments?

---

> ### Author Response · Authors · 2023-11-22
>
> We appreciate the reviewer's insightful comments and questions. Our responses are provided below
>
> > As suggested by the paper, the main conceptual message, as well as the key distinction induced by the network overparameterization, is that meta-learning not only learns the desired subspace spanned by the columns of $B_\star$, but also “unlearns” the orthogonal complement of this subspace. However, the analysis is based on the fact that the number of inner-loop training samples $m_{\mathrm{in}}$ is insufficient so the inner-loop gradient descent is prone to overfitting, which is then penalized by the outer-loop meta-training process. What if we use other techniques to mitigate overfitting such as weight decay as commonly used in practice?
>
> Following the reviewer's query, we have examined the impact of regularization. We must confirm this rigorously, but our preliminary understanding is that L2 weight decay regularization (applied in the outer loop) accelerates training and enhances the regularization effect. The adaptation performance remains similarly reliant on $k$. Depending on the magnitude of the L2 coefficient and the particular regime of interest—whether $m_{\mathrm{test}} > k$ or $m_{\mathrm{test}} < k$—the regularization could either hinder or aid generalization.
>
> We reiterate the main focus in this work is to capture the learning dynamics of FO-ANIL in a well-understood, simple theoretical setting to improve our understanding of model-agnostic meta-learning. This has shown the good “implicit bias” of FO-ANIL despite overparameterisation.
>
> > In the high level, the theoretical results in this paper show that overparameterization/more task-specific data provably hurts generalization in their simplified meta-learning setting. I am not sure if this is consistent with the growing empirical evidence that overparameterization/more data usually helps generalization in practical deep learning. Does this mismatch suggest that the linear model considered in this paper is somewhat over-simplified?
>
> The negative impact of using more task-specific data occurs when there are fewer samples during testing. This situation often leads to overfitting when the features lack a regularized, low-rank structure. The intuition is that the network, having been trained on tasks with a larger number of samples, may become overconfident in its adaptability. Note that this setting is somewhat atypical. Typically, if pretraining involves tasks with a large number of samples, one would expect the test tasks to also have a large number of samples. Moreover, it is important to note that this issue arises only with a single-step gradient descent at test time. As detailed in Appendix F, FO-ANIL has optimal performance in this setting when the number of test-time samples matches the number of training samples.
>
> In more complex fine-tuning scenarios, the asymptotic low-rank behaviour that is beneficial for generalisation applies to any finite sample size. However, whether a larger number of samples is advantageous remains unclear. This question is theoretically challenging, even in the context of this simple setting.
>
> On the other hand, a slowdown caused by overparameterisation has been proven in supervised learning [1]. Our findings seem to support the ubiquity of this type of slowdown. However, our perspective on overparameterisation is more optimistic: feature learning through meta-learning pretraining is effective, regardless of the size of the hidden dimension. This eliminates the need to estimate the correct dimension, which can be hard in real-world scenarios, and allows the use of overparameterised networks regardless of the hidden dimension. A more general result on adaptability is relevant as it supports the use of the most expressive model feasible for solving a problem. And as the reviewer already suggested, the experimental evidence from practice seems to support the idea that overparametrisation usually helps. Our work represents an initial step towards a theoretical understanding of such adaptive behaviour in the context of model-agnostic meta-learning.
>
> [1] Xu, Weihang, and Simon Du. "Over-parameterization exponentially slows down gradient descent for learning a single neuron." The Thirty Sixth Annual Conference on Learning Theory. PMLR, 2023.

---

### Official Review · Reviewer_aoqC · 2023-10-30

**Soundness:** 3 good
**Presentation:** 2 fair
**Contribution:** 3 good
**Rating:** 5
**Confidence:** 4

**Summary:**

This work theoretically studies the ability of first-order ANIL to find useful generalisation points for a network trained on one subsequent gradient descent step. Firstly the authors show that ANIL is able to align to the same feature space as the ground-truth data generating model and also completely ignores the orthogonal complement of this feature space. This implies that ANIL is learning an appropriate representation space for the set of possible tasks, essentially affording the model an initialisation with useful features. The benefit due to this useful initialisation is then characterised by bounding the network's risk for a new task after a single gradient step. This is shown to have better scaling when the network hidden layer is smaller than the original input feature space. Experiments demonstrating the alignment of the features space learned by  ANIL to ground-truth are shown and some results showing the benefit of ANIL over ridge regression are also shown.

**Strengths:**

## Originality
This work uses a teacher-student style of analysis where a ground-truth model generates data for the subsequent trained model. This is a well-established method of analysis but I have not seen it applied to ANIL or meta-learning. Thus, this work presents a sufficiently original setting for theory. It is also made clear how this work departs from other similar analysis in recent literature such as [1].

## Quality
Assumptions of this work are clearly stated and the limitations made explicit. The setup is also appropriate for studying the desired phenomena. The results and conclusions drawn from the theory are accurate and clear. Similarly, the interpretation of experimental results is also clear and direct from what is shown. The experimental design showing the approach of the meta-learned feature space singular values is also appropriate and supports and complements the theoretical results. The logical progression of the arguments made in Sections 2 and 3 is also intuitive and follows a clear structure which is appreciated.

## Clarity
The paper is well written and figures are clear and neat. Notation is used appropriately and is intuitive. Of particular note is how the mathematics is discussed. The language is clear and precise. This results in the mathematics and notation supporting the written discussion very well. I don't feel like it is always the case that both forms of exposition are of a high quality and it does help this work greatly. The Introduction is also particularly clear and establishes the setting of the paper well.

## Significance
The significance of this work is a particular strong points. The results shown here are practical and insightful. The fact that ANIL provably learns features is an important result and I can a lot of future work tying into these results - even just by analogy or inspiration, for example in work on the lottery ticket hypothesis and continual learning. Additionally, the result of a convergence rate provides a helpful degree of nuance to the results and one which I can see guiding practical intuition and future work which aims to converge quicker.

[1] Liam Collins, Aryan Mokhtari, Sewoong Oh, and Sanjay Shakkottai. Maml and anil provably learn representations. arXiv preprint arXiv:2202.03483, 2022.

**Weaknesses:**

## Quality
There are unfortunately some weaker points to the quality of this work as well. I do not think enough is done to thoroughly guide a reader through the actual theory of this work. While I appreciate that presenting theory in a page limit is difficult - and I do not propose for full derivations or anything of the sort be moved out of the appendix - there is in general little discussion or proof sketches presented. As a result the theorems and propositions are stated and it is not clear how the statement is derived from the premise. This is better for Proposition 1 than Theorem 1, where Proposition 1 at least has some discussion in the paragraph following immediately where it is stated that the error is decomposed into three terms and then bounded by concentration inequalities. What is two lines of writing then lead to deeper insight and more of this is needed. I see in Appendix A that a proof sketch of Theorem 1 is given and it is quite long. But as far as I can tell sufficient space could be made for a condensed version in the main paper and this should be prioritised as Theorem is the main result of this work which even Proposition 2 relies on. As a result, some points which I think could be very impactful fall flat, such as those at the top of page 5.

On the point of making space for a proof sketch, I appreciate the in-depth discussion of this work. However I am not certain that it should be prioritised over more technical detail. Particularly the long comparison to [1] seems excessive for the main paper and could be deferred to the appendix along with much of this section. Another example would be the statement "Theorem 1 answers the conjecture of Saunshi et al. (2020)." but the conjecture is not stated and so this does not benefit the reader. I would also caution against phrases like "More importantly, the infinite samples idealisation does not reflect the initial motivation of meta-learning, which is to learn tasks with a few samples. Interesting phenomena are thus not observed in this simplified setting.". Once again, this is quite conversational and speculative for the main body of work. It also comes across as fairly combative and seems to belittle prior work. Thus, I think it would be best to compress the discussion, keeping mainly the limitations component and the component which introduces Burer-Monteiro since it is used in the experiments. This would make significance space for Appendix A in the main paper.

Finally, I am concerned that the experiment summarised in Table 1 does not quite assess or demonstrate the truth of Proposition 1. Since a scaling law is given in Proposition 1, is it not possible to experimentally show how tight of a bound this is? Rather than just demonstrating that meta-learning can out-perform ridge regression on the input features but not ridge regression on the ground-truth representation?

## Clarity
I re-iterate that I think clarity is a strong point of this work. If possible, I would suggest a figure summarising the setup be presented. It would certainly be helpful and could significantly aid clarity. Once again space would need to be made, but I think the Introduction could also be compressed to make this possible. This is a minor point and I do think the setting is clear just with the notation. If such a figure could not fit in the main paper then one in the appendix would still be helpful.

Given the above. I would be inclined to increase my score to a 6 or 7 if the restructuring of Appendix A and Section 4 occurs and is done so clearly. I would increase my score further if experiments supporting Proposition 1 are shown and the figure summarising the setup included.

**Questions:**

1. Can Equation 7 not be simplified further since $\Sigma_*$ is proportional to the identity matrix and $B_*$ is orthogonal?
2. The scaling law on the generalisation error of FO-ANIL is based on $k$ while linear regression is based on $d$. This is used to justify that FO-ANIL outperforms regression, but does this only hold when ground truth representation space is smaller than the input space $k < d$? What if $k>d$?
3. On the first line of the subsection titled "Initialisation Regime" the bounded initialisation for Theorem 1 is mentioned. This seems different to what is stated in the theorem. Where exactly does this need for bounded initialisation appear in the theorem or its assumptions?

## Minor Points
1. "The goal of meta agnostic methods" is a typo.
2. Why is the highlighting misaligned for the pdf? There may be a rendering issue.

---

> ### Author Response · Authors · 2023-11-22
>
> We thank the reviewer for the detailed and helpful review. We respond to each of the points mentioned below.
>
> > There are unfortunately some weaker points to the quality of this work as well. I do not think enough is done to thoroughly guide a reader through the actual theory of this work. While I appreciate that presenting theory in a page limit is difficult - and I do not propose for full derivations or anything of the sort be moved out of the appendix - there is in general little discussion or proof sketches presented. As a result the theorems and propositions are stated and it is not clear how the statement is derived from the premise...
>
> Our presentation of Theorem 1 was indeed brief and lacked necessary details, largely due to space limitations. We have now added a paragraph that details the proof strategy and explains how specific assumptions are utilised in the proof.
>
> > On the point of making space for a proof sketch, I appreciate the in-depth discussion of this work. However I am not certain that it should be prioritised over more technical detail. Particularly the long comparison to [1] seems excessive for the main paper and could be deferred to the appendix along with much of this section. Another example would be the statement "Theorem 1 answers the conjecture of Saunshi et al. (2020)." but the conjecture is not stated and so this does not benefit the reader. I would also caution against phrases like "More importantly, the infinite samples idealisation does not reflect the initial motivation of meta-learning, which is to learn tasks with a few samples. Interesting phenomena are thus not observed in this simplified setting.". Once again, this is quite conversational and speculative for the main body of work. It also comes across as fairly combative and seems to belittle prior work. Thus, I think it would be best to compress the discussion, keeping mainly the limitations component and the component which introduces Burer-Monteiro since it is used in the experiments. This would make significance space for Appendix A in the main paper.
>
> We have done our best to condense the discussion. The first three paragraphs are retained as each highlights a unique aspect of our work. The “Infinite tasks models” paragraph, to which the reviewer referred, remains since it emphasises an essential modelling decision which contrasts with prior works. However, we have revised the language to reflect the reviewer's suggestions. Lastly, most of the technical comparisons with [1] have been moved to the Appendix.
>
> > Finally, I am concerned that the experiment summarised in Table 1 does not quite assess or demonstrate the truth of Proposition 1. Since a scaling law is given in Proposition 1, is it not possible to experimentally show how tight of a bound this is? Rather than just demonstrating that meta-learning can out-perform ridge regression on the input features but not ridge regression on the ground-truth representation?
>
> We have included additional experiments in the appendix I.5. to verify the scaling described in Proposition 1. Specifically, we have conducted controlled experiments to separately assess the impact of the first term (that does not depend on $m_{\mathrm{test}}$) and the last two terms (that scale with $\sqrt{k/m_{\mathrm{test}}}$) in the bound and show that the scaling in Proposition 1 is tight.
>
> > I re-iterate that I think clarity is a strong point of this work. If possible, I would suggest a figure summarising the setup be presented. It would certainly be helpful and could significantly aid clarity. Once again space would need to be made, but I think the Introduction could also be compressed to make this possible. This is a minor point and I do think the setting is clear just with the notation. If such a figure could not fit in the main paper then one in the appendix would still be helpful.
>
> As suggested by the reviewer, we have added a figure that summarises the meta-learning task considered in the paper, and highlights the task distribution, its decomposition into two-layer teachers and the misspecification/overparameterisation in the student network.
>
> > Given the above. I would be inclined to increase my score to a 6 or 7 if the restructuring of Appendix A and Section 4 occurs and is done so clearly. I would increase my score further if experiments supporting Proposition 1 are shown and the figure summarising the setup included.
>
> We hope these revisions meet the reviewer’s expectations, and we remain open to any further comments.

---

> ### Author Response · Authors · 2023-11-22
>
> > Can Equation 7 not be simplified further since $\Sigma_\star$ is proportional to the identity matrix and $B_\star$ is orthogonal?
>
> The covariance $\Sigma_\star$ could be replaced by identity, but $B_\star B_\star^\top \in \mathbb{R}^{d \times d}$ is a rank $k$ projection that cannot be further simplified. We have kept $\Sigma_\star$ in the presentation to enhance clarity, ensuring that Equation 7 is preserved with arbitrary covariances.
>
> > The scaling law on the generalisation error of FO-ANIL is based on $k$ while linear regression is based on $d$. This is used to justify that FO-ANIL outperforms regression, but does this only hold when ground truth representation space is smaller than the input space $k < d$? What if  $k > d$?
>
> In the linear regression setting considered, $k > d$ is not possible: assume a task distribution $\theta_{\star, i}$ over $\mathbb{R}^d$ that is induced by the matrix $B_{\star} \in \mathbb{R}^{d \times k}$ and tasks $w_{\star, i} \in \mathbb{R}^k$. As $B_\star \in \mathbb{R}^{d \times k}$ is at most rank $d$, this could be written as $B_{\star}' \in \mathbb{R}^{d \times d}$ and $w_{\star, i}' \in
> \mathbb{R}^{d}$ where $w_{\star, i}'$ is the projection of the $w_{\star, i}$ to the row space of $B_\star$. So, it is equivalent to a $k=d$ case.
>
> > On the first line of the subsection titled "Initialisation Regime" the bounded initialisation for Theorem 1 is mentioned. This seems different to what is stated in the theorem. Where exactly does this need for bounded initialisation appear in the theorem or its assumptions?
>
> We use the term "bounded" to describe the initialisation regime of Theorem 1. Indeed, the parameters $B$ and $w$ need to be initialised small compared to some problem parameters, as detailed in the first equation in Theorem 1. We use “bounded” to characterise this “small compared to” property.

---

### Official Review · Reviewer_FVcv · 2023-10-31

**Soundness:** 3 good
**Presentation:** 4 excellent
**Contribution:** 3 good
**Rating:** 8
**Confidence:** 3

**Summary:**

The paper studies the shared representation learned by first-order ANIL in a linear multi-task model. It theoretically shows that in the case of infinite tasks and overparameterized width, FO-ANIL succeeds in extracting the low-rank ground-truth shared structure while learning to ignore orthogonal directions allowing for fast adaptation. It empirically validates these theoretical findings contrasting the low-rank FO-ANIL solutions with those found via multi-task learning.

**Strengths:**

The paper is well motivated, presents its results clearly and is well embedded in existing literature.
The findings it presents are interesting and novel.

**Weaknesses:**

The paper clearly motivates the need to study a simplified linear, first-order, single gradient-step version of ANIL. Nevertheless, a small-scale empirical verification to the extent by which violating some of these assumptions affect the results could have further strengthened the paper.

While proofs for all theoretical results are provided in the appendix, the code for reproducing the small empirical section was missing upon submission to the best of my knowledge. Providing it would increase my confidence in the reproducibility of the paper.

**Questions:**

1. In your theoretical analysis you assume matching depth in the model architecture. Since this is a linear model, is it possible to say something about the case where we have a mismatch in terms of the number of layers? (given that in the linear case this does not change the expressiveness of the model)
2. Have you tried adding a nonlinearity to your model in the empirical experiments? I wonder to what extent this affects the main conclusion of learning the right subspace and unlearning the orthogonal one.

---

> ### Author Response · Authors · 2023-11-22
>
> We thank the reviewer for the review and questions.
>
> > In your theoretical analysis you assume matching depth in the model architecture. Since this is a linear model, is it possible to say something about the case where we have a mismatch in terms of the number of layers? (given that in the linear case this does not change the expressiveness of the model)
>
> Thanks for this great question. Firstly, in the linear case, a teacher with multiple layers is effectively equivalent to a two-layer formulation. Secondly, with students that have multiple layers, controlling the weight dynamics could be significantly more challenging. As discussed in Appendix F, one key insight persists: the global minima maintains the form $(B, w)$ form, where $B$ is now the product of initial layers up to the last layer. This structural property reveals that despite its nonconvexity, the problem has good global minima with rank deficiencies. It may suggest that our insights (regarding the low-rank regularised asymptotic limit and the dependency on $k$ during adaptation) from our paper could extend to multi-layer linear students. We leave such explorations for future work.
>
> > The paper clearly motivates the need to study a simplified linear, first-order, single gradient-step version of ANIL. Nevertheless, a small-scale empirical verification to the extent by which violating some of these assumptions affect the results could have further strengthened the paper.
>
> We have provided additional experiments in Appendix I.6. with 2 and 3-layer ReLU networks and checked the scaling of generalisation as $d$ varied. Our experiments show that the adaptation performance of FO-ANIL pretraining with ReLU networks does not scale with $d$. This is evidence supporting our main insight in ReLU networks: model-agnostic meta-learning pretraining adapts to the implicit problem complexity despite overparametrisation.
>
> > Have you tried adding a nonlinearity to your model in the empirical experiments? I wonder to what extent this affects the main conclusion of learning the right subspace and unlearning the orthogonal one
>
> We provide an additional experiment in Appendix I.6. with ReLU networks, where we first approximate the network representation as a linear map and then check learning and unlearning behaviour with this linear proxy. Our experiments show that, qualitatively, the picture is similar: the correct subspace is learned while the complement space stays small.

---

### Official Review · Reviewer_4g2e · 2023-11-06

**Soundness:** 4 excellent
**Presentation:** 3 good
**Contribution:** 2 fair
**Rating:** 5
**Confidence:** 4

**Summary:**

This paper studies the ANIL meta-learning algorithm, and proves that the meta-learned feature extractor converges towards a fixed point.
This result is obtained under the following assumptions:

- The learned architecture is a two-layer linear network, possibly over-parameterized.
- The algorithm is ANIL with first-order gradient (ie, FO-ANIL where the adaptation loop is not differentiated).
- There are infinite tasks but finite samples per tasks.
- Each task is generated by with a ground-truth linear regressor and additive noise.

Additionally, it shows that such fixed point yields a good performance after only a single step of gradient descent adaptation. The authors provide an extensive (and insightful!) discussion of their results, and validate them empirically under similar assumptions as above.

**Strengths:**

- Strong and correct theoretical analysis: the authors obtain non-trivial results for FO-ANIL and — to the best of my assessment — their analysis is correct. I especially like the two-steps approach, which first shows the existence of a fixed-point representation (Theorem 1), and then showing that this fixed-point achieves low excess risk (Proposition 1, Appendix C).
- In-depth discussion of results: my favorite part of the paper is Section 4, where the authors discuss the implications of their results. For example, they clearly contrast their result with prior work and argue that, in the over-parameterized regime (large width), model-agnostic methods such as ANIL can outperform traditional multi-task methods. (See “Superiority of agnostic methods” in Section 4.) They also show that this advantage does not appear with infinite samples thus motivating future work to pay attention to this regime. They also bring attention to the importance of a full-rank initializations, which is a generally understudied area in gradient-based meta-learning, potentially responsible for the many instabilities we observe in practice.

**Weaknesses:**

- Significance: while the analysis is strong, I wonder if it is relevant. FO-ANIL is almost never used in practice (ANIL is cheap enough), and definitely not with (two) linear layers. Thus, could the authors bring forward the insights that may apply to more realistic settings? I know others also consider this setting (eg, Tripuraneni et al., 2020) but they tend to include insights beyond parameter convergence (eg, effect of task diversity).
- Limited scope of empirical results: since this paper studies a highly idealized setting, I would hope the authors could provide results echoing their theorems in settings where the assumptions are relaxed. For example, what about multiple non-linear layers? Or 2nd-order ANIL (or even MAML)? Or a cross-entropy loss, which would be relevant to classification problems? Or, better, could we validate these insights on benchmarks widely used in the literature for real-world algorithms? ANIL is easy and cheap enough to implement that these asks are not unrealistic and would strongly support the authors’ results.
- Novelty: the authors emphasize that their analysis is in the finite-sample regime, which they successfully argue is significantly more relevant than infinite-sample one. However they are not the first ones to study this setting: as they mention, Collins et al., 2022 already consider it, but Ji et al., 2022 (Theoretical Convergence of Multi-Step Model-Agnostic Meta-Learning) and Fallah et al., 2021 (Generalization of Model-Agnostic Meta-Learning Algorithms: Recurring and Unseen Tasks) also do, and provide convergence guarantees and generalization bounds for MAML, a more generic algorithm than ANIL. Neither of these works are mentioned in the paper, so I’d encourage the authors to discuss the novel insights they bring to the meta-learning community which are not already available in these works.

**Questions:**

- While ANIL is typically motivated as an approximation to MAML, it much more closely resembles metric-based few-shot learning such as prototypical networks, metaoptnet, or R2D2 where the main difference is the algorithm to solve the linear head. How difficult would it be to extend your results for ANIL to these methods?
- In section 4, you discuss overfitting at the end of the “infinite task models” paragraphs. It isn’t obvious to me how unlearning the orthogonal space reduces overfitting and why there’s no need to unlearn it with infinite samples — could you expand on what is meant? I’m trying to understand how this observation may translate to the other methods mentioned above.

---

> ### Author Response · Authors · 2023-11-22
>
> > Significance: while the analysis is strong, I wonder if it is relevant. FO-ANIL is almost never used in practice (ANIL is cheap enough), and definitely not with (two) linear layers. Thus, could the authors bring forward the insights that may apply to more realistic settings?...
>
> Thanks for the insightful comment. Our work answers two important questions on the behavior of FO-ANIL for two-layer linear networks: (a) how come these architectures can be successful despite misspecification? and (b) how does FO-ANIL learn the shared task representation while not being designed to do so? These questions extend beyond this simple setting and are relevant in practice of model-agnostic meta-learning: (a) architectural misspecifications are common in practice as practitioners tend to favour the largest, feasible model and (b) as pointed out by Raghu et al. (2020) (Rapid Learning or Feature Reuse? Towards Understanding the Effectiveness of MAML), model-agnostic meta-learning methods learn shared representations even though they are not designed explicitly for representation learning.
>
> Moreover, we think insights derived in our theory-friendly setting, such as regularisation and dependency on the shared complexity of tasks instead of the ambient dimension, also apply to practical settings. We have provided additional experiments with ReLU networks that show promising results in line with this hypothesis. Lastly, we also show that FO-ANIL recovers the global minima of the ANIL (Appendix F). This could indicate that FO-ANIL and ANIL are closely related in the two-layer linear setting.
>
> > Limited scope of empirical results: since this paper studies a highly idealized setting, I would hope the authors could provide results echoing their theorems in settings where the assumptions are relaxed. For example, what about multiple non-linear layers? Or 2nd-order ANIL (or even MAML)?...
>
> We have added additional experiments with 2 and 3-layer ReLU networks in Appendix I.6. We first approximate the network representation as a linear map and then check singular values of this linear proxy. Our experiments indicate that our insights generalise to FO-ANIL pretraining with ReLU networks. In particular, the singular values associated with good directions are increasing whereas the rest of the singular values stay small. In addition, we provide experimentations with ANIL, FO-MAML and MAML. The results are quantitatively the same for all up to the scaling of the singular values.
>
> > Novelty: the authors emphasize that their analysis is in the finite-sample regime, which they successfully argue is significantly more relevant than infinite-sample one. However they are not the first ones to study this setting...
>
> Our work departs from the studies by Ji et al. (2022) and Fallah et al. (2021), which analyse the convergence of MAML for general loss functions without addressing representation learning. Ji et al. (2022) show convergence to a local minimum, and Fallah et al. (2021) study meta-learning under a strong convexity assumption (which does not apply to networks with 2 or more layers, even with linear activation). In contrast, our work focuses on feature learning, drawing inspiration from the insights of Raghu et al. (2020) (Rapid Learning or Feature Reuse? Towards Understanding the Effectiveness of MAML). In the same vein as Collins et al. (2022), we explore feature learning within a specific teacher-student framework. We highlight that, despite these methods not being explicitly designed for learning shared representations, such learning occurs if the tasks share underlying structures. We have included a new paragraph in Appendix A to further explain the connection between our work and these two prior works. We will expand on this brief literature review in order to better contrast our work with other orthogonal theoretical works on meta-learning.
>
> The novelties of our work compared to the setting of Collins et al. (2022) are detailed thoroughly in the discussed section. Most notably, we consider a misspecified setting where the hidden layer is overparametrised. This is an important aspect that is important to capture in theory, and our work shows successful representational learning irrespective of the degree of overparametrisation. Although Collins et al. (2022) address the finite task and sample regime, it should be noted that the analysis is based on the concentration of the finite sample update around the infinite sample case. This regime, as argued in our work, does not capture the additional unlearning behaviour that is explained by our analysis. Another key improvement in our work is the initialisation, which does not require any weak alignment. In $k \ll d$ settings, the usual full rank initialisation assumed by Collins et al. (2022) fails, even when $k' = k$, indicating that their analysis might only describe the latter stages of the training process (i.e., training phase after the initial harder alignment phase is completed).

---

> > ### Author Response · Authors · 2023-11-22
> >
> > > While ANIL is typically motivated as an approximation to MAML, it much more closely resembles metric-based few-shot learning such as prototypical networks, metaoptnet, or R2D2 where the main difference is the algorithm to solve the linear head. How difficult would it be to extend your results for ANIL to these methods?
> >
> > Thank you for raising this excellent question. We have also identified a related question for future investigation: what would happen if the inner loop involved running multiple steps of gradient descent? Providing a precise answer to these questions is challenging without studying a teacher-student setting under a form of idealisation, such as infinite tasks, individually for each meta-learning algorithm. For algorithms like R2D2 and MetaOptNet, the multi-task framework could be used (one could take linearly separable classification tasks for SVM and logistic regression).
> >
> > More broadly, we expect to see regularisation in FO-ANIL from the few-shot inner step to transfer to such methods with more intricate inner finetuning steps. This regularisation would then be beneficial to obtain a simple structure that is better at adaptation with fewer samples at hand. A priori, it is harder to comment on learning dynamics. One of our interesting findings is that FO-ANIL does not require any form of alignment at initialisation. An interesting related question is if this applies to algorithms with more sophisticated finetuning.
> >
> > > In section 4, you discuss overfitting at the end of the “infinite task models” paragraphs. It isn’t obvious to me how unlearning the orthogonal space reduces overfitting and why there’s no need to unlearn it with infinite samples — could you expand on what is meant? I’m trying to understand how this observation may translate to the other methods mentioned above.
> >
> > In the presence of a finite number of samples, the excess risk of the corresponding linear regression is known to scale in the square root of the dimension. Even worse can happen when the number of samples is smaller than the dimension: in that case, the linear regression can overfit the training set and yields a very bad performance on new, unobserved, test data.
> >
> > Unlearning the orthogonal space here means that we decrease the initial representation dimension from $k’$ to $k$, which first improves the scaling of the excess risk, but also avoids this potential bad overfitting regime when the number of samples ($m_{\mathrm{test}}$) is between $k$ and $k’$.
> >
> > On the other hand, when having an infinite number of samples in linear regression, the excess risk will be exactly $0$ (i.e., we will recover exactly the model parameters). As a consequence, there is no benefit from reducing the representation dimension in that case, which is why it is not done by the algorithm.
> >
> > On the pretraining dynamics, this translates in the outer step. The outer step indeed plays a regularisation role, as it uses validation data (with $m_{\mathrm{out}}$ samples per task): if the inner step tends to overfit, this overfitting, that will happen on the orthogonal subspace, will be mitigated during the outer loop. This mitigation will be done by unlearning, i.e., reducing the weight given to this orthogonal subspace.
> > All of this does not happen with $m_{\mathrm{in}}=\infty$, as there is no overfitting to mitigate.

---

### Meta-Review · Area_Chair_yfaP · 2023-12-05

**Metareview:**

This submission provides a theoretical analysis of ANIL, a first-order optimization-based meta-learning algorithm empirically argued to learn shared representations during pretraining. The authors show that, in the case of infinite number of tasks, ANIL indeed successfully learns linear shared representations, even in the case of architectural misspecifications (e.g., overparametrization).

Reviewers commented positively on the correctness, clarity, and strength of the theoretical analysis (pdse, 4g2e, FVcv), although the comments by reviewer aoqC suggest this could be further improved.  A possible weakness of this submission is the idealized theoretical setting (two-layer MLPs) and the limited use of FO-ANIL (Reviewer 4g2e), although the authors try to address this point by providing additional experiments for different architectures to increase the case for practical relevance. While reviewer FVcv argues that code availability would improve their confidence, this is not a requirement for an ICLR submission, especially since the emphasis of this work is of a more theoretical nature.

Overall, while relevance to the practical use cases of Meta-Learning (including diverse architectures and algorithms) are perhaps not clearly established, I would not consider this a case strong enough for rejection, given the often necessary simplifications required to make the theoretical process and the authors' attempts to provide a complementary empirical perspective.

**Justification For Why Not Higher Score:**

While Reviewer FVcv provides a relatively high score, I would have liked to see a more thorough review for this review to count enough for a higher score. In addition, this would have also required a more enthusiastic response by at least one more reviewer.

**Justification For Why Not Lower Score:**

The authors appear to have done their best in responding to criticism through the rebuttal, responding convincingly to several questions and comments.  I also feel that theoretical work in meta-learning is perhaps underrepresented as a proportion of work in this field, making the argument for the relative contribution of this submission stronger.

---

### Decision · Program_Chairs · 2024-01-16

Accept (poster)